# Do we need to estimate the variance in robust mean estimation?

## Abstract

In this paper, we propose self-tuned robust estimators for estimating the mean of heavy-tailed distributions, where heavy-tailed distributions refer to distributions with only finite variances. Our method involves introducing a new loss function that considers both the mean parameter and a robustification parameter. By simultaneously optimizing the empirical loss function with respect to both parameters, the resulting estimator for the robustification parameter can automatically adapt to the unknown data variance and can achieve near-optimal finite-sample performance. Our approach outperforms previous methods in terms of both computational and asymptotic efficiency. Specifically, it does not require cross-validation or Lepski's method to tune the robustification parameter, and the variance of our estimator achieves the Cramér-Rao lower bound.

## 1 Introduction

The success of numerous statistical and learning methods heavily relies on the assumption of light-tailed or sub-Gaussian errors (Wainwright, 2019). A random variable $Z$ is considered to have sub-Gaussian tails if there exist constants $c_1$ and $c_2$ such that $\mathbb{P}(|Z - \mathbb{E}Z| > t) \leq c_1 \exp(-c_2 t^2)$ for any $t \geq 0$. However, in many practical applications, data are often collected with a high degree of noise. For instance, in the context of gene expression data analysis, it has been observed that certain gene expression levels exhibit kurtoses much larger than 3, regardless of the normalization method used (Wang et al., 2015). Furthermore, a recent study on functional magnetic resonance imaging (Eklund et al., 2016) demonstrates that the principal cause of invalid functional magnetic resonance imaging inferences is that the data do not follow the assumed Gaussian shape. It is therefore important to develop robust statistical methods with desirable statistical performance in the presence of heavy-tailed data.

In this paper, we focus on robust mean estimation problems, which serves as the foundation for tackling more general problems. Specifically, we consider a generative model where data, $\{y_i, 1 \leq i \leq n\}$, are generated according to

$$y_i = \mu^* + \varepsilon_i, \ 1 \leq i \leq n, \tag{1.1}$$

where the random errors $\varepsilon_i \in \mathbb{R}$ are independent and identically distributed (i.i.d.) samples from $\varepsilon$, which follows the law $F_0$. We assume that $\varepsilon$ is mean zero and has a finite variance only, with $\mathbb{E}_{\varepsilon \sim F_0} \varepsilon = 0$ and $\mathbb{E}_{\varepsilon \sim F_0} \varepsilon^2 = \sigma^2$, where the expectation $\mathbb{E}_{\varepsilon \sim F_0} \varepsilon$ represents the expected value of $\varepsilon$ when $\varepsilon$ follows the distribution $F_0$.

When estimating the mean $\mu^*$, the sample mean estimator $\sum_{i=1}^{n} y_i / n$ is known to achieve, at best, a polynomial-type nonasymptotic confidence width (Catoni, 2012), when the errors have only finite variances. This means that there exists a distribution $F = F_{n,\delta}$ for $\varepsilon$ with a mean of 0 and variance of $\sigma^2$, such that

the followings hold simultaneously:

$$\mathbb{P}\left(\left|\sum_{i=1}^{n}\frac{y_i}{n}-\mu^*\right|\leq\sigma\sqrt{\frac{1}{2n}\cdot\frac{1}{\delta}}\right)\geq 1-2\delta, \quad \forall\ \delta\in\left(0,1/2\right);$$

$$\mathbb{P}\left(\left|\sum_{i=1}^{n}\frac{y_i}{n}-\mu^*\right|\leq\sigma\sqrt{\frac{1}{2n}\cdot\frac{1}{\delta}}\left(1-\frac{2e\delta}{n}\right)^{\frac{n-1}{2}}\right)\leq 1-2\delta, \quad \forall\ \delta\in\left(0,(2e)^{-1}\right).$$

In simpler terms, this indicates that the sample mean does not converge quickly enough to the true mean when the errors have only finite variances.

Catoni (2012) made an important step towards estimating the mean with faster concentration. Specifically, Catoni introduced a robust mean estimator $\widehat{\mu}(\tau)$, which depends on a tuning parameter $\tau$, and deviates from the true mean $\mu^*$ logarithmically in $1/\delta$. Specifically, with $\tau$ properly tuned, $\widehat{\mu}(\tau)$ satisfies the following concentration inequality:

$$\mathbb{P}\left(|\widehat{\mu}(\tau)-\mu^*|\leq c\sigma\sqrt{\frac{1}{n}\cdot\log\left(\frac{1}{\delta}\right)}\right)\geq 1-2\delta, \tag{1.2}$$

where $c$ is some constant. We refer to estimators that satisfy this deviation property as sub-Gaussian mean estimators because they achieve the same performance as if the data were sub-Gaussian. Following Catoni's work, there has been a surge of research on sub-Gaussian estimators using the empirical risk minimization approach in various settings; see Brownlees et al. (2015); Hsu & Sabato (2016); Fan et al. (2017); Avella-Medina et al. (2018); Lugosi & Mendelson (2019b); Lecué & Lerasle (2020); Wang et al. (2021) and Sun et al. (2020), among others. For a recent review, we refer readers to Ke et al. (2019).

To implement Catoni's estimator (Catoni, 2012), there is a tuning parameter $\tau=\tau(\sigma)$ that depends on the unknown variance $\sigma^2$ and needs to be carefully tuned. However, in practice, this often involves computationally expensive methods such as cross-validation or Lepski's method (Catoni, 2012). For instance, when using the adaptive Huber estimator (Sun et al., 2020; Avella-Medina et al., 2018) to estimate a $d\times d$ covariance matrix entrywise, as many as $d^2$ tuning parameters can be involved. If cross-validation or Lepski's method were employed, the computational burden would increase significantly as $d$ grows. Therefore, it is natural to ask the following question:

> *Is it possible to develop computationally efficient robust mean estimators for distributions with finite but unknown variances?*

This paper tackles the aforementioned challenge by proposing a self-tuned robust mean estimator for distributions with only two moments. We propose an empirical risk minimization (ERM) approach based on a novel loss function. The proposed loss function is smooth with respect to both the mean parameter and the robustification parameter. By jointly optimizing both parameters, we prove that the resulting robustification parameter can automatically adapt to the unknowns, and the resulting mean estimator can achieve sub-Gaussian-like performance as if the data were Gaussian, up to logarithmic terms. Therefore, compared to previous methods, our approach eliminates the need to use cross-validation or Lepski's method to tune the robustification parameters. This significantly boosts the computational efficiency of data analysis in practical applications. Furthermore, from an asymptotic standpoint, we establish that our proposed estimator is asymptotically efficient, meaning it achieves the Cramér-Rao lower bound asymptotically (Van der Vaart, 2000).

**Related work** In addition to the empirical risk minimization (ERM) based methods, median-of-means (MoM) techniques (Devroye et al., 2016; Lugosi & Mendelson, 2019b; Lecué & Lerasle, 2020) are often used to construct robust estimators in the presence of heavy-tailed distributions. A recent survey on median-of-means can be found in the work by Lugosi & Mendelson (2019a). The MoM technique involves randomly splitting the full data into $k$ subsamples and computing the mean for each subsample. The MoM estimator is then obtained as the median of these local estimators. The number of subsamples $k$ is the only user-defined

parameter in MoM, and it can be chosen to be independent of the unknowns, making it tuning-free. However, in our experience, MoM often exhibits undesirable numerical performance when compared to ERM based estimators. To understand this phenomenon, we take an asymptotic viewpoint and compare the asymptotic efficiencies of our estimator and the MoM estimator. We show that the relative efficiency of the MoM estimator with respect to ours is only $2/\pi \approx 0.64$.

**Paper overview** Section 2 introduces a novel loss function and presents the empirical risk minimization (ERM) approach. The nonasymptotic theory is presented in Section 3. In Section 4, we compare our proposed estimator with popular alternatives in terms of asymptotic performance. Section 5 provides numerical experiments. Finally, we conclude in Section 6. The supplementary material contains basic calculations, algorithms, a comparison with Lepski's method, proofs of the main results, supporting lemmas, and additional results.

**Notation** We summarize here the notation that will be used throughout the paper. We use $c$ and $C$ to denote generic constants which may change from line to line. For two sequences of real numbers $\{a_n, n \geq 1\}$ and $\{b_n, n \geq 1\}$, $a_n \lesssim b_n$ or $a_n = O(b_0)$ denotes $a_n \leq Cb_n$ for some constant $C > 0$, and $a_n \gtrsim b_n$ if $b_n \lesssim a_n$. We use $a_n \propto b_n$ to denote that $a_n \gtrsim b_n$ and $a_n \lesssim b_n$. $\widetilde{O}$ hides logarithmic terms. The log operator is understood with respect to the base $e$. For a function $f(x, y)$, we use $\nabla_x f(x, y)$ or $\frac{\partial}{\partial x} f(x, y)$ to denote its partial derivative of $f(x, y)$ with respect to $x$. $\nabla f(x, y)$ denotes the gradient of $f(x, y)$. For a vector $x \in \mathbb{R}^d$, let $\|x\|_2$ denote its Euclidean norm. For a symmetric positive semi-definite matrix $\Sigma$, $\lambda_{\max}(\Sigma)$ denotes its largest eigenvalue.

## 2 A new loss function for self-tuning

This section introduces a new loss function to robustly estimate the mean of distributions with only finite variances while automatically tuning the robustification parameter.

We start with the pseudo-Huber loss (Hastie et al., 2009)

$$\ell_\tau(x) = \tau\sqrt{\tau^2 + x^2} - \tau^2 = \tau^2\sqrt{1 + x^2/\tau^2} - \tau^2, \tag{2.1}$$

where $\tau$ serves as a tuning parameter. It exhibits behavior similar to the Huber loss (Huber, 1964), approximating $x^2/2$ when $|x|$ is small and resembling a straight line with slope $\tau$ for large values of $|x|$. To see this, some algebra yields

$$\begin{cases} \frac{\epsilon^2 - 2(1+\epsilon)}{2\epsilon^2} x^2 \leq \ell_\tau(x) \leq \frac{x^2}{2}, & \text{if } x^2/\tau^2 \leq 4(1+\epsilon)/\epsilon^2, \\ \frac{\tau|x|}{1+\epsilon} \leq \ell_\tau(x) \leq \tau|x|, & \text{if } x^2/\tau^2 > 4(1+\epsilon)/\epsilon^2. \end{cases}$$

We refer to $\tau$ as the robustification parameter because it controls the trade-off between the quadratic loss and the least absolute deviation loss, where the latter induces robustness. In practice, $\tau$ is typically tuned using computationally expensive methods such as Lepski's method (Catoni, 2012) or cross-validation (Sun et al., 2020).

To avoid these computationally expensive procedures, our goal is to propose a new loss function of both the mean parameter $\mu$ and the robustification parameter $\tau$ (or its equivalent) so that optimizing over them jointly yields an automatically tuned robustification parameter $\widehat{\tau}$ and thus the correspondingly self-tuned mean estimator $\widehat{\mu}(\widehat{\tau})$. Unlike the Huber loss (Sun et al., 2020), the pseudo-Huber loss is a smooth function of $\tau$, making optimization with respect to $\tau$ possible. To motivate the new loss function, let us first consider the estimator $\widehat{\mu}(\tau)$ with $\tau$ fixed *a priori*:

$$\widehat{\mu}(\tau) = \operatorname*{argmin}_\mu \left\{ \frac{1}{n} \sum_{i=1}^n \ell_\tau(y_i - \mu) \right\}. \tag{2.2}$$

We provide an informal result below, with its rigorous version presented as Theorem 3.1 in subsequent sections.

**Theorem 2.1** (Informal result)**.** Take $\tau = \sigma\sqrt{n}/z$ with $z = \sqrt{\log(1/\delta)}$, and assume $n$ is sufficiently large. Then for any $0 < \delta < 1$, with probability at least $1 - \delta$, we have

$$|\widehat{\mu}(\tau) - \mu^*| \lesssim \sigma\sqrt{\frac{\log(2/\delta)}{n}}.$$

The above result indicates that when $\tau = \sigma\sqrt{n}/z$ with $z = \sqrt{\log(1/\delta)}$, the estimator $\widehat{\mu}(\tau)$ achieves the desired sub-Gaussian performance. The only unknown parameter in $\tau$ is the standard deviation $\sigma$. In view of this, we treat $\sigma$ as an unknown parameter $v$, substitute $\tau = \sqrt{n}v/z$ into (2.1), and obtain

$$\ell(x, v) := \ell_\tau(x) = \frac{nv^2}{z^2}\left(\sqrt{1 + \frac{x^2 z^2}{nv^2}} - 1\right), \tag{2.3}$$

where $z$ is a confidence parameter because it depends on $\delta$ as in the theorem above.

Instead of searching for the optimal $\tau$, we will search for the optimal $v$, which is expected to be close to the underlying standard deviation $\sigma$ intuitively. We will use the term "robustification parameter" interchangeably for both $\tau$ and $v$, as they are equivalent up to a multiplier. However, directly minimizing $\ell(x, v)$ with respect to $v$ leads to meaningless solutions, specifically $v = 0$ and $v = +\infty$. To avoid these trivialities, we consider a new loss by dividing $\ell(x, v)$ by $v$ and then adding a linear penalty function $av$. This will be referred to as the penalized pseudo-Huber loss, formally defined below.

**Definition 2.2** (Penalized pseudo-Huber loss)**.** The penalized pseudo-Huber loss $\ell^P(x, v)$ is defined as follows:

$$\ell^P(x, v) := \frac{\ell(x, v) + av^2}{v} = \frac{nv}{z^2}\left(\sqrt{1 + \frac{x^2 z^2}{nv^2}} - 1\right) + av, \tag{2.4}$$

where $n$ is the sample size, $z$ is a confidence parameter, and $a$ is an adjustment factor.

We thus propose to optimize jointly over $\mu$ and $v$ by solving the following ERM problem:

$$\{\widehat{\mu}, \widehat{v}\} = \underset{\mu, v}{\operatorname{argmin}}\left\{L_n(\mu, v) := \frac{1}{n}\sum_{i=1}^{n}\ell^P(y_i - \mu, v)\right\}$$

$$= \underset{\mu, v}{\operatorname{argmin}}\frac{1}{n}\sum_{i=1}^{n}\left\{\frac{nv}{z^2}\left(\sqrt{1 + \frac{(y_i - \mu)^2 z^2}{nv^2}} - 1\right) + av\right\}. \tag{2.5}$$

When $v$ is fixed *a priori*, solving the optimization problem above with respect to $\mu$ is equivalent to directly minimizing the empirical pseudo-Huber loss in (2.2) with $\tau = v\sqrt{n}/z$.

To gain insight into the loss function $L_n(\mu, v)$, let us consider its population version:

$$L(\mu, v) = \mathbb{E}L_n(\mu, v) = \frac{nv}{z^2}\mathbb{E}\left(\sqrt{1 + \frac{(y - \mu)^2 z^2}{nv^2}} - 1\right) + av.$$

Define the population oracle $v_*$ as the minimizer of $L(\mu^*, v)$ with the true mean $\mu^*$ given *a priori*, that is

$$v_* = \underset{\tau}{\operatorname{argmin}}\, L(\mu^*, v) = \underset{v}{\operatorname{argmin}}\left\{\frac{nv}{z^2}\mathbb{E}\left(\sqrt{1 + \frac{(y - \mu)^2 z^2}{nv^2}} - 1\right) + av\right\},$$

or equivalently,

$$\nabla_v L(\mu^*, v)\big|_{v=v_*} = \left\{\frac{n}{z^2}\left(\nabla_v\mathbb{E}\sqrt{v^2 + \frac{\varepsilon^2 z^2}{n}} - 1\right) + a\right\}\bigg|_{v=v^*} = 0.$$

By interchanging the derivative with the expectation, we obtain

$$\mathbb{E}\frac{v_*}{\sqrt{v_*^2 + z^2\varepsilon^2/n}} = 1 - \frac{az^2}{n}. \qquad (2.6)$$

Let $\sigma_{x^2}^2 := \mathbb{E}\{\varepsilon^2 1(\varepsilon^2 \leq x^2)\}$, where $1(A)$ is the indicator function of the set $A$. Our first key result utilizes the above characterization of $v_*$ to derive how $v_*$ is able to automatically adapt to the standard deviation $\sigma$, promising the effectiveness of our procedure.

**Theorem 2.3** (Self-tuning property of $v_*$)**.** Suppose $n \geq az^2$. Then, for any $\gamma \in [0, 1)$, we have

$$\frac{(1-\gamma)\sigma_{\varphi\tau_*^2}^2}{2a} \leq v_*^2 \leq \frac{\sigma^2}{2a},$$

where $\varphi = \gamma/(1-\gamma)$ and $\tau_* = v_*\sqrt{n}/z$. Moreover, we have $\lim_{n\to\infty} v_*^2 = \sigma^2/(2a)$.

The above result indicates that for any finite sample size $n \geq az^2$, the oracle $v_*^2$ can automatically adapt to the (truncated) variance. It is bounded between the scaled truncated variance $\sigma_{\varphi\tau_*}^2/(2a)$ and the scaled variance $\sigma^2/(2a)$. Since the second moment exists, we have $\sigma_{\varphi\tau^2}^2 \to \sigma^2$ as $\varphi\tau_*^2 \to \infty$ by the dominant convergence theorem. For a large sample size $n$, $\sigma_{\varphi\tau^2}^2$ is close to $\sigma^2$, and therefore $v_*^2$ is approximately between $(1-\gamma)\sigma^2/(2a)$ and $\sigma^2/(2a)$. Furthermore, an asymptotic analysis reveals that $\lim_{n\to\infty} v_*^2 = \sigma^2/(2a)$. Taking $a = 1/2$ yields $\lim_{n\to\infty} v_*^2 = \sigma^2$, indicating that the oracle $v_*^2$ with $a = 1/2$ should approximate the true variance in the large sample limit. This also suggests the optimality of choosing $a = 1/2$, which is assumed throughout the rest of the paper.

Our next result shows that the proposed empirical loss function is jointly convex in both $\mu$ and $v$. This convexity property allows us to employ standard first-order optimization algorithms to compute the global optima efficiently.

**Proposition 2.4** (Joint convexity)**.** The empirical loss function $L_n(\mu, v)$ in (2.5) is jointly convex in both $\mu$ and $v$. Furthermore, if there exist at least two distinct data points, the empirical loss function is strictly convex in both $\mu$ and $v$ provided that $v > 0$.

Lastly, it was brought to our attention that our formulation (2.5) bears a resemblance to the concomitant estimator by Ronchetti & Huber (2009):

$$\operatorname*{argmin}_{\mu,v}\left\{\frac{1}{n}\sum_{i=1}^{n}\rho\left(\frac{y_i - \mu}{v}\right)v + av\right\},$$

where $\rho$ is any loss function, and $a$ is a user-specified constant. Notably, the selection of the appropriate constant $a$ is scarcely addressed within the existing literature. Our motivation stems from a different perspective. We aim to develop robust mean estimators that exhibit improved finite-sample properties in the presence of heavy-tailed data. The empirical loss function $L_n$ that we propose can be perceived as an intricately adapted variant of theirs. Specifically, we leverage the smooth pseudo-Huber loss, in which we set the robustification parameter $\tau$ as $\tau = v\sqrt{n}/z$ to ensure the sub-Gaussian-like performance for the robust mean estimator. Here $z$ serves as a judiciously chosen confidence parameter. Concurrently, we identify the optimal adjustment factor as $a = 1/2$. To the best of our knowledge, all of these findings are the first among the literature.

## 3 Finite-sample theory

This section presents the self-tuning property for estimated robustification parameter and then the finite-sample property of the self-tuned mean estimator. Recall $a = 1/2$.

### 3.1 Estimation with a fixed $v$

With a light abuse of notation, we use $\widehat{\mu}(v)$ to denote the minimizer of the penalized pseudo-Huber loss in (2.5) with $v$ fixed. Recall that we have used $\widehat{\mu}(\tau)$ to denote the minimizer of the pseudo-Huber loss in (2.2), and the $\widehat{\mu}(v)$ equivalent to $\widehat{\mu}(\tau)$ with $\tau = v\sqrt{n}/z$. We start with the theoretical properties for $\widehat{\mu}(v)$. We need the following locally strong convexity assumption, which will be verified later in this subsection.

**Assumption 1** (Locally strong convexity in $\mu$)**.** The empirical Hessian matrix is locally strongly convex with respect to $\mu$ such that, for any $\mu \in \mathbb{B}_r(\mu^*) := \{\mu : |\mu - \mu^*| \leq r\}$,

$$\inf_{\mu \in \mathbb{B}_r(\mu^*)} \frac{\langle \nabla_\mu L_n(\mu, v) - \nabla_\mu L_n(\mu^*, v), \mu - \mu^* \rangle}{|\mu - \mu^*|^2} \geq \kappa_\ell > 0$$

where $r > 0$ is a local radius parameter.

**Theorem 3.1.** For any $0 < \delta < 1$, suppose $v > 0$ is fixed and let $z^2 = \log(1/\delta)$. Assume Assumption 1 holds with any $r \geq r_0 := \kappa_\ell^{-1} \left( \sigma/(\sqrt{2}v) + 1 \right)^2 \sqrt{\log(2/\delta)/n}$. Then, with probability at least $1 - \delta$, we have

$$|\widehat{\mu}(v) - \mu^*| < \frac{1}{\kappa_\ell} \left( \frac{\sigma}{\sqrt{2}v} + 1 \right)^2 \sqrt{\frac{\log(2/\delta)}{n}} = \frac{C}{\kappa_\ell} \sqrt{\frac{\log(2/\delta)}{n}},$$

where $C = (\sigma/(\sqrt{2}v) + 1)^2$ only depends on $v$ and $\sigma$.

The above theorem states that under the assumption of local strong convexity, $\widehat{\mu}(v)$ achieves a sub-Gaussian deviation bound when the data have only bounded variances. In particular, if we choose $v = \sigma$ and apply the theorem, we obtain:

$$|\widehat{\mu}(\sigma) - \mu^*| \leq \frac{1}{\kappa_\ell} \left( \frac{\sigma}{\sigma} + 1 \right)^2 \sqrt{\frac{\log(2/\delta)}{n}} \leq \frac{4}{\kappa_\ell} \sqrt{\frac{\log(2/\delta)}{n}}.$$

Assumption 1 requires the loss function to exhibit curvature in a small neighborhood $\mathbb{B}_r(\mu^*)$, while the penalized loss (2.4) transitions from a quadratic function to a linear function roughly at $|x| = \tau \propto \sqrt{n}$. Quadratic functions always have curvature, so intuitively, Assumption 1 holds as long as

$$\sqrt{n} \gtrsim r \geq r_0 \propto \sqrt{1/n}.$$

The latter condition is automatically guaranteed when $n$ is sufficiently large. Taking $r$ to be the smallest $r_0$ results in Assumption 2 being at its weakest. In other words, in this scenatrio, the empirical loss function only needs to exhibit curvature in a diminishing neighborhood of $\mu^*$, approximatley with radius of $\sqrt{1/n}$. The following lemma rigorously proves this claim.

**Lemma 3.2.** Suppose $v \geq v_0$. For any $0 < \delta < 1$, suppose $n \geq C \max \left\{ z^2(\sigma^2 + r^2)/v_0^2, \log(1/\delta) \right\}$ for some absolute constant $C$. Then, with probability at least $1 - \delta$, Assumption 1 with $\kappa_\ell = 1/(2v)$ holds uniformly over $v \geq v_0 > 0$.

The first sample complexity condition that $n \geq Cz^2(\sigma^2 + r^2)/v_0^2$ comes from requirement that $\tau_{v_0}^2 := v_0^2 n/z^2 \geq C(\sigma^2 + r^2)$ in the proof of Lemma 3.2. Recall that the robustification parameter $\tau_{v_0}^2 := v_0^2 n/z^2$ determines the size of the quadratic region. Intuitively, this requirement is minimal in the sense that Assumption 2 can only hold when $\tau_{v_0}^2$ is larger than $r^2$ plus the noise variance $\sigma^2$ (due to stochasticity). As argued before, Assumption 2 holds with any $r$ such that $\sqrt{n} \gtrsim r \gtrsim \sqrt{1/n}$. Thus we can take $r$ to be a constant, and this will not make the sample complexity condition worse. Finally, by combing Lemma 3.2 and Theorem 3.1, we obtain the following result.

**Corollary 3.3.** Suppose $v \geq v_0$. For any $0 < \delta < 1$, suppose $n \geq C \max \left\{ (r^2 + \sigma^2)/v_0^2, 1 \right\} \log(1/\delta)$ for some universal constant $C$, and let $z^2 = \log(1/\delta)$. For any $v \geq v_0$, we have with probability at least $1 - \delta$ that

$$|\widehat{\mu}(v) - \mu^*| \leq 2v \left( \frac{\sigma}{\sqrt{2}v} + 1 \right)^2 \sqrt{\frac{\log(4/\delta)}{n}} \lesssim v \sqrt{\frac{1 + \log(1/\delta)}{n}}.$$

### 3.2 Self-tuned mean estimators

We proceed to characterize the theoretical property of the self-tuned mean estimator. We need the additional constraint that $v_0 \leq v \leq V_0$, and consider the constrained empirical risk minimization problem

$$\{\widehat{\mu}, \widehat{v}\} = \underset{\mu,\, v_0 \leq v \leq V_0}{\operatorname{argmin}} \left\{ L_n(\mu, v) := \frac{1}{n} \sum_{i=1}^{n} \ell^{\mathrm{P}}(y_i - \mu, v) \right\}. \tag{3.1}$$

Indeed, when $v$ is either $0$ or $\infty$, the loss function is no longer smooth or it becomes trivial respectively. In other words, the loss function is not strongly convex in $\mu$ in either case, and the strong convexity is essential for our analysis. Let $\tau_{v_0} = v_0 \sqrt{n}/z$.

**Theorem 3.4** (Self-tuning property). Assume that $n$ is sufficiently large. Let $c_0$ and $C_0$ be some constants, and suppose $v_0 < c_0 \sigma_{\tau_{v_0}^2/2-1} \leq C_0 \sigma < V_0$. For any $0 < \delta < 1$, take $z^2 \geq \log(5/\delta)$. Then, with probability at least $1 - \delta$, we have

$$c_0 \sigma_{\tau_{v_0}^2/2-1} \leq \widehat{v} \leq C_0 \sigma.$$

The above theorem suggests that $\widehat{v}$ automatically adapts to the standard deviation, aka $\widehat{v}$ approximates $\sigma$, if $\sigma_{\tau_{v_0}^2/2-1}$ approximates $\sigma$ which is expected to hold for a large sample size by the dominated convergence theorem. Of course $\sigma_{\tau_{v_0}^2}$ can not be close to $\sigma$ at any predictable rate under the weak assumption that the data have bounded variances only. We proceed to characterize the finite-sample property of the self-tuned mean estimator $\widehat{\mu}(\widehat{v})$.

**Theorem 3.5.** Assume that $n$ is sufficiently large. Let $c_0$ and $C_0$ be some constants, and suppose $v_0 < c_0 \sigma_{\tau_{v_0}^2/2-1} \leq C_0 \sigma < V_0$. For any $0 < \delta < 1$, take $z^2 = \log(n/\delta)$. Then, with probability at least $1 - \delta$, we have

$$|\widehat{\mu}(\widehat{v}) - \mu^*| \leq C \cdot \sigma \sqrt{\frac{\log(n/\delta)}{n}}$$

where $C$ is some constant.

The above result indicates that the mean estimator $\widehat{\mu} = \widehat{\mu}(\widehat{v})$ with a self-tuned robustification parameter $\widehat{v}$ achieves the optimal deviation property up to a logarithmic factor. In practical applications, we recommend setting $\delta = 0.05$, which corresponds to a failure probability of $0.05$ or a confidence level of $0.95$.

## 4 Comparing with alternatives

Other than the ERM based approach, the median-of-means technique (Lugosi & Mendelson, 2019a) is another method to construct robust estimators under heavy-tailed distributions. The MoM mean estimator works as follows:

1. Partition $[n] = \{1, \ldots, n\}$ into $k$ blocks $\mathcal{B}_1, \ldots, \mathcal{B}_k$, each of size $|\mathcal{B}_i| \geq \lfloor n/k \rfloor \geq 2$.

2. Compute the sample mean in each block $z_j = \frac{1}{|B_j|} \sum_{i \in B_j} x_i$.

3. Take the median of $z_j$'s as the the MoM mean estimator $\widehat{\mu}^{\mathrm{MoM}} = \mathrm{med}(z_1, \ldots, z_k)$, where $\mathrm{med}(\cdot)$ is the median operator.

The following theorem is taken from Lugosi & Mendelson (2019a). Without loss of generality and for simplicity, we shall assume throughout this section that $n$ is divisible by $k$ so that each block has $m = n/k$ elements.

**Theorem 4.1** (Theorem 2 by Lugosi & Mendelson (2019a) )**.** For any $\delta \in (0, 1)$, if $k = \lceil 8 \log(1/\delta) \rceil$, then, with probability at least $1 - \delta$,

$$\left| \widehat{\mu}^{\text{MoM}} - \mu^* \right| \le \sigma \sqrt{\frac{32 \log(1/\delta)}{n}}.$$

The theorem above indicates that in order to obtain a sub-Gaussian mean estimator, we only need to choose $k = \lceil 8 \log(1/\delta) \rceil$ when constructing the MoM mean estimator. Thus, the MoM estimator is naturally tuning-free. However, in our numerical experiments, we have observed that the MoM estimator often has inferior numerical performance compared to our proposed estimator. To shed light on this observation, we compare the asymptotic efficiencies of $\widehat{\mu}^{\text{MoM}}$ and our estimator $\widehat{\mu}(\widehat{\tau})$ in the following two theorems.

**Theorem 4.2** (Asymptotic inefficiency of $\widehat{\mu}^{\text{MoM}}$)**.** Fix any $0 < \iota \le 1$. Assume $\mathbb{E}|y_i - \mu^*|^{2+\iota} < \infty$. Suppose $k \to \infty$ and $k = o\big(n^{\iota/(1+\iota)}\big)$, then

$$\sqrt{n} \left( \widehat{\mu}^{\text{MoM}} - \mu^* \right) \rightsquigarrow \mathcal{N} \left( 0, \frac{\pi}{2} \sigma^2 \right).$$

**Theorem 4.3** (Asymptotic efficiency of our estimator)**.** Fix any $0 < \iota \le 1$. Assume $\mathbb{E}\varepsilon_i^{2+\iota} < \infty$ and the same assumptions as in Theorem 3.4. Take $z^2 = 2 \log(n)$. Then

$$\sqrt{n} \left( \widehat{\mu}(\widehat{v}) - \mu^* \right) \rightsquigarrow \mathcal{N} \left( 0, \sigma^2 \right).$$

We emphasize that the MoM mean estimator shares the same asymptotic property as the median estimator (Van der Vaart, 2000) due to taking the median operation in the third step, and thus is asymptotically inefficient. In sharp contrast, our proposed estimator achieves full asymptotic efficiency. The relative efficiency $e_{\text{r}}$ of MoM with respect to our estimator is

$$e_{\text{r}} \left( \widehat{\mu}^{\text{MoM}}, \widehat{\mu}(\widehat{v}) \right) = \frac{2}{\pi} \approx 0.64.$$

This means that our proposed estimator is more efficient than the MoM estimator in terms of asymptotic performance, partly explaining the empirical success of our method; see Section 5 for details.

We explain intuitively why our self-tuned estimator can achieve (near) optimal performance in both the finite-sample regime and the asymptotic regime. Because our self-tuned estimator in (3.1) is a self-tuned version of the pseudo-Huber estimator in (2.2), thus we focus on the pseudo-Huber estimator $\widehat{\mu}(\tau)$. Theorem 2.1 suggests that taking $\tau = \sigma \sqrt{n/\log(1/\delta)}$ guarantees the sub-Gaussian performance of $\widehat{\mu}(\tau)$ for finite samples. Meanwhile, as $n \to \infty$, we have $\tau = \sigma \sqrt{n/\log(1/\delta)} \to \infty$. Thus the pseudo-Huber loss approaches to the least square loss which corresponds to the negative maximum likelihood of Gaussian distributions, which leads to the asymptotically efficient mean estimator.

For MoM estimators, the situation differs. On one hand, to attain robustness in the finite-sample regime, the number of blocks $k$ should be greater than or equal to $\lceil 8 \log(1/\delta) \rceil$, as demonstrate in the proof of Theorem 4.1 by Lugosi & Mendelson (2019a). On the other hand, to approach the sample mean estimator and achieve asymptotic efficiency in the large sample limit, the number of blocks should diminish to 1 as the sample size $n$ grows. Consequently, optimal finite-sample and asymptotic properties represent two contrasting characteristics for MoM estimators. In other words, the MoM estimator can not simultaneously adapt to both regimes (to perform optimally). This contrast seems to arise from the discontinuous nature of the MoM estimator which cannot smoothly transition from requiring at least $k = 3$ blocks (for defining the median) to functioning as an empirical mean estimator.

Another popular estimator is the trimmed mean estimator (Lugosi & Mendelson, 2021). The univariate trimmed mean estimator works as follows: (i) Split the data points into two subsamples with equal size, (ii) use the first subsample to determine the trimming parameters, and (iii) use the second subsample to construct the trimmed mean estimator. Due to this sample splitting scheme, the trimmed mean estimator lacks sample efficiency.

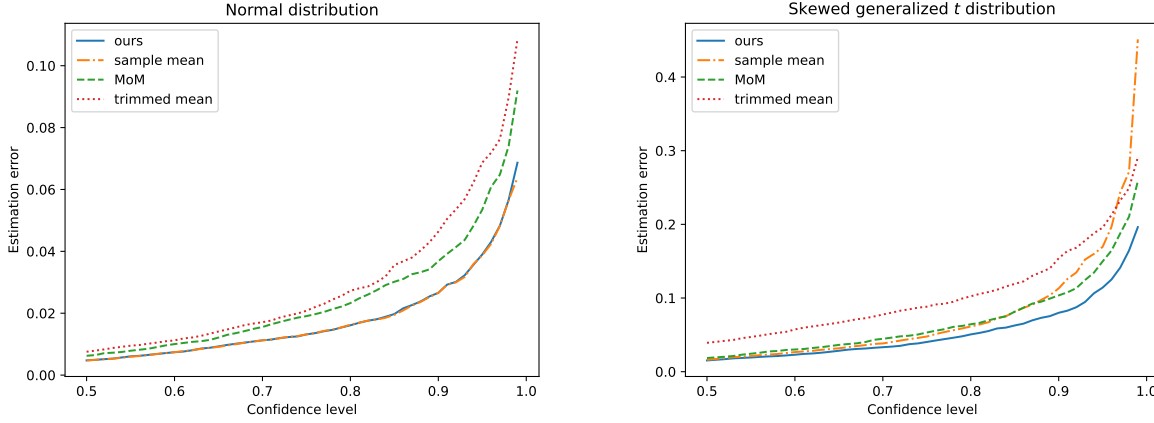

Figure 1: Estimation error versus confidence level for our estimator, the sample mean estimator, the MoM estimator and the trimmed mean estimator.

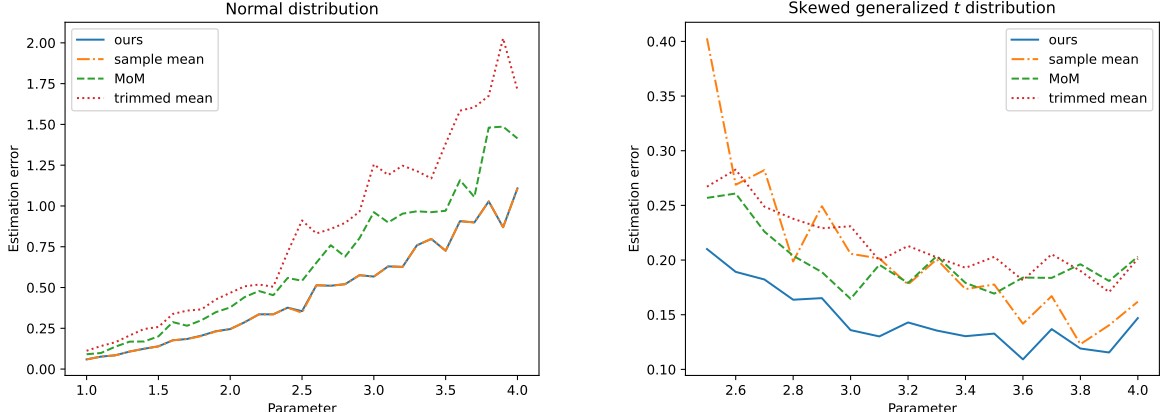

Figure 2: Empirical 99%-quantile of the estimation error versus parameter measuring tails and skewness for our estimator, the sample mean estimator, the MoM estimator and the trimmed mean estimator.

## 5 Numerical studies

This section examines numerically the finite sample performance of our proposed mean estimators in the presence of heavy-tailed data. In all of our numerical examples, we take $z = \sqrt{\log(n/\delta)}$ with $\delta = 0.05$ as suggested by Theorem 3.5. This choice ensures that our results hold with a probability of at least 0.95. We consider the following four distribution settings for the random data point $y$ in order to investigate the robustness and efficiency of the proposed estimator:

1. Normal distribution $\mathcal{N}(\mu, \sigma^2)$ with mean $\mu = 1$ and a sequence of variances $\sigma^2 \geq 1$;

2. Skewed generalized $t$ distribution $\mathsf{sgt}(\mu, \sigma^2, \lambda, p, q)$, where mean $\mu = 0$, a sequence of variances $\sigma^2 = q/(q-2)$ with $q > 2$, shape $p = 2$ and skewness $\lambda = 0.75$.

For each setting, we generate an independent sample of size $n = 100$ and compute four mean estimators: our proposed estimator (ours), the sample mean estimator, the MoM mean estimator, and the trimmed mean estimator.

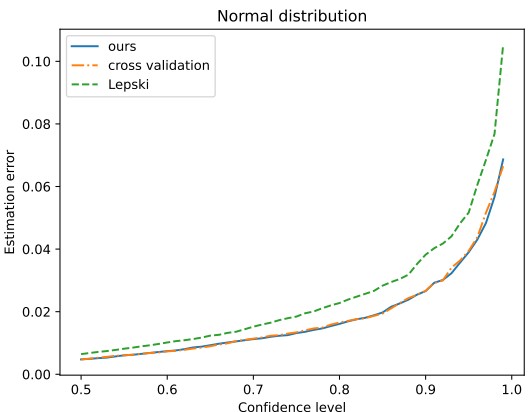
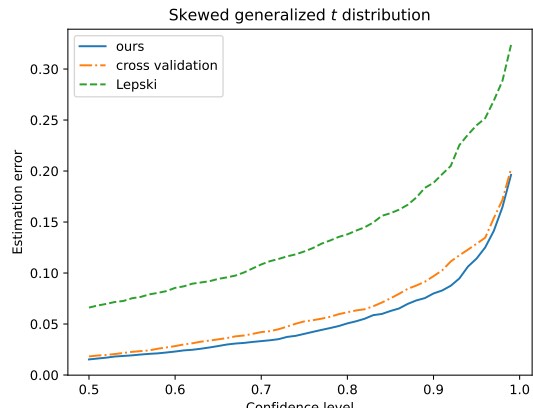

Figure 3: Estimation error versus confidence level for our estimator, cross validation and Lepski's method.

Figure 1 displays the $\alpha$-quantile of the squared estimation error, with $\alpha$ ranging from 0.5 to 1, based on 1000 simulations for both scenarios. For Gaussian distributed data, our estimator performs almost identically to the sample mean estimator, both of which outperform the MoM mean estimator and the trimmed mean estimator. Since the sample mean estimator is the optimal mean estimator for Gaussian distributed data, this suggests that our estimator does not sacrifice statistical efficiency when the data is Gaussian. In the case of heavy-tailed skewed generalized $t$ distributions, the deviation of the sample mean from the population mean grows rapidly with the confidence level. This is in contrast to the three robust estimators: our estimator, the MoM mean estimator, and the trimmed mean estimator. Our estimator is the only one that consistently outperforms the others in both scenarios.

Figure 2 examines the 99%-quantile of the estimation error versus a distribution parameter that measures the tail behavior and skewness, based on 1000 simulations. Specifically, for Gaussian data, we let $\sigma$ vary between 1 and 4. For skewed generalized $t$ distributions, we increase the shape parameter $q$ from 2.5 to 4. For Gaussian data, our estimator performs identically to the sample mean estimator, both of which outperform the MoM mean estimator and the trimmed mean estimator. For skewed generalized $t$ distributions with $q \leq 3$, all three robust mean estimators outperform, or are as competitive as, the sample mean estimator. When $q > 3$, the sample mean estimator starts to performs better than both MoM and the trimmed mean estimator. Our proposed estimator, on the other hand, consistently outperforms all other methods across the entire range of parameter values.

We also compare the performances of our proposed method, pseudo-Huber loss + cross validation and pseudo-Huber loss + Lepski's method. For cross validation, we choose the best $\tau$, which is equivalent to choosing the best $v$, from a list of candidates $\{1, 2, \ldots, 100\}$ using 10-fold cross validation. For Lepski's method, we follow the appendix and pick $V = 2$, $\rho = 1.2$ and $s = 50$. We run 1000 simulations for the mean estimation problem in Setting 1 with $\sigma^2 = 1$ and sample size $n = 100$. All studies are performed on a Macbook Pro with Apple M1 Max and 64 GB memory. The execution times are summarized in Table 1. Our proposed method is about $90\times$ faster than cross validation and about $10\times$ faster than Lepski's method. The run time for sample mean, MoM, and trimmed mean in the same scenario is 0.018, 0.111, and 0.057 seconds, respectively. Lastly we compare the run time for our estimator with increasing sample sizes. Specifically, for $n = 100, 1000, 10,000, 100,000$, the run time is 1.54, 1.58, 3.02, 25.04 seconds, respectively.

Finally we compare their statistical performances in both settings with various parameters. The results are summarized in Figure 3 and Figure 4. In both figures, the our method and the cross validation have similar performances although our method is slightly better while Lepski's method does not perform well. We suspect this is because Lepski's method depends on three more hyper parameters $V, \rho$ and $s$ and our choice are perhaps not optimally tuned. This perhaps shows that the Lepski's method does not achieve great empirical performances in general.

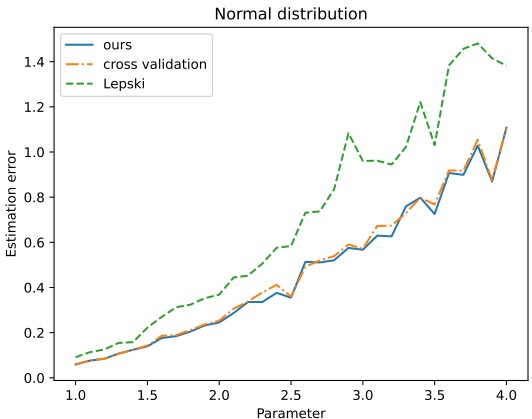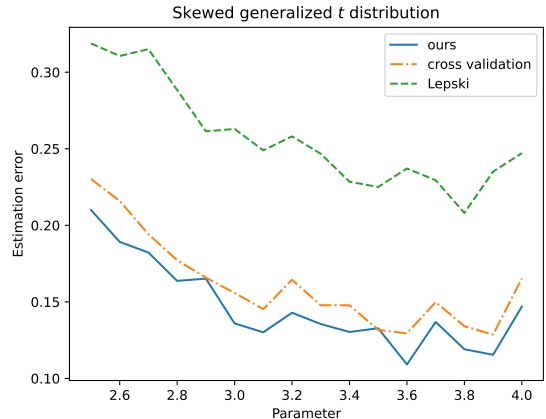

Figure 4: Empirical 99%-quantile of the estimation error versus parameter measuring tails and skewness for our estimator, cross validation and Lepski's method.

Table 1: Comparing different tuning methods: Run time (in seconds) for 1000 simulations in Setting 1 with $\sigma^2 = 1$ and $n = 100$.

| ours | Lepski's method | cross validation |
|------|-----------------|------------------|
| 1.5  | 16.7            | 133.5            |

In summary, the most attractive feature of our method is its self-tuning property: (i) It is as efficient as the sample mean estimator for normal distributions and is more robust for asymmetric and/or heavy-tailed distributions; (ii) It incurs much less computational cost than cross validation and Lepski's method. The latter property is particularly important for large-scale inference with a myriad of parameters to be estimated simultaneously.

## 6    Conclusions

**Summary**   This paper investigates robust mean estimation for distributions with finite variances only. We introduce a novel loss function that depends on both the mean parameter and a robustification parameter. By jointly optimizing these parameters, we demonstrate that, even under only second moment conditions, the resulting robustification parameter can automatically adapt to the variance. As a result, our mean estimator achieves nearly optimal performance in finite samples, akin to the case of sub-Gaussian data. This distinguishes our approach from previous methods that require cross-validation or Lepski's method to tune the robustification parameter.

**Adaptivity**   In our experience, the performance of MoM estimators is often subpar[1], and our proposed estimator consistently outperforms MoM estimators. As discussed previously, we believe this is because our estimator can perform (near) optimally in both the finite-sample and large-sample regimes. We shall refer to this ability as "adaptivity to different regimes". The MoM estimator does not naturally enjoy this adaptivity due to its discontinuous nature.

**The multidimensional case**   We briefly discuss how to extend the proposed estimator to the multivariate case. Assume model (1.1) but with $y_i, \mu^*$, and $\varepsilon_i \in \mathbb{R}^d$ being i.i.d. such that $\mathbb{E}\varepsilon_i = 0$ and $\text{cov}(\varepsilon_i) = \Sigma$. A simple strategy, as recommended by one of the referees, is to apply the univariate estimator coordinate-wise and then combine them to form our final estimator $\widehat{\mu}$. Then the following proposition holds. Let $\sigma_{kk}^2$ be the $k$-th diagonal term of $\Sigma$, and $\sigma_{kk,x^2}^2 = \mathbb{E}[\varepsilon_{ik}^2 1(\varepsilon_{ik}^2 \le x^2)]$, where $\varepsilon_{ik}$ is the $k$-th coordinate of $\varepsilon_i$.

---

[1]We had similar empirical observations in our earlier studies.

**Proposition 6.1.** Assume that $n$ is sufficiently large. Let $c_0$ and $C_0$ be some constants, and suppose $v_0 < c_0 \sigma_{kk, \tau_{v_0}^2 - 1} \leq C_0 \sigma_{kk} < V_0$. For any $0 < \delta < 1$, taking $z^2 = \log(n/\delta)$, with probability at least $1 - \delta$, we have

$$\|\widehat{\mu} - \mu^*\|_2 \leq C \sqrt{\frac{\mathrm{tr}(\Sigma) \, \log(nd/\delta)}{n}},$$

where $C$ is some constant.

We also have the following asymptotic result which states that the multivariate mean estimator also achieves asymptotic efficiency.

**Proposition 6.2** (Asymptotic efficiency for the multivariate mean estimator)**.** Fix any $0 < \iota \leq 1$. Assume $\max_{1 \leq k \leq d} \mathbb{E} \varepsilon_{ik}^{2+\iota} < \infty$ and the same assumptions as in Proposition 6.1. Take $z^2 = 2 \log(n)$. Then

$$\sqrt{n} \left( \widehat{\mu}(\widehat{v}) - \mu^* \right) \rightsquigarrow \mathcal{N} \left( 0, \Sigma \right).$$

**Limitation**  One limitation is that the finite-sample performance of our self-tuned estimator depends on unknown constants, which means that the sample complexity cannot be computed in advance for fixed error. Moreover, the proposed estimator is only optimal up to a logrithmic term. It remains unclear whether this logrithmic factor can be removed. Another limitation is the scope of the study. This paper focuses on robust mean estimators since this is the simplest case and the proof is already complicated. However, it is possible to extend current work to more general problems, such as regression and matrix estimation problems. We have extended the estimator to the multivariate case in the above but such an extension is not optimal; see Lugosi & Mendelson (2019a) for the optimal finite-sample bound. It would also be interesting to study the asymptotic properties of the multivariate median-of-mean estimators.

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

# Appendix

## Table of Contents

## A  Basics

This section collects the basic facts such as first-order derivatives and the Hessian matrix for the loss function. Let $\tau = v\sqrt{n}/z$ throughout the appendix. Recall that our loss function is

$$L_n(\mu, v) = \frac{1}{n}\sum_{i=1}^{n} \ell(y_i - \mu, v) = \frac{1}{n}\sum_{i=1}^{n}\left\{\frac{\sqrt{n}}{z}\sqrt{\frac{nv^2}{z^2} + (y_i - \mu)^2} - \left(\frac{n}{z^2} - a\right)v\right\}$$

$$= \frac{1}{n}\sum_{i=1}^{n}\left\{\frac{\sqrt{n}}{z}\left(\sqrt{\tau^2 + (y_i - \mu)^2} - \tau\right) + a\cdot\frac{\tau}{z\sqrt{n}}\right\}.$$

The first-order, second-order derivatives of $L_n(\mu, v)$ are

$$\nabla_\mu L_n(\mu, v) = -\frac{1}{n}\sum_{i=1}^n \frac{y_i - \mu}{v\sqrt{1 + z^2(y_i - \mu)^2/(nv^2)}} = -\frac{\sqrt{n}}{z} \cdot \frac{1}{n}\sum_{i=1}^n \frac{y_i - \mu}{\sqrt{\tau^2 + (y_i - \mu)^2}},$$

$$\nabla_v L_n(\mu, v) = \frac{1}{n}\sum_{i=1}^n \frac{n/z^2}{\sqrt{1 + z^2(y_i - \mu)^2/(nv^2)}} - \left(\frac{n}{z^2} - a\right) = \frac{n}{z^2} \cdot \frac{1}{n}\sum_{i=1}^n \left(\frac{\tau}{\sqrt{\tau^2 + (y_i - \mu)^2}} - 1\right) + a$$

where $a = 1/2$. The Hessian matrix is

$$H(\mu, v) = \begin{bmatrix} \frac{\sqrt{n}}{z}\frac{1}{n}\sum_{i=1}^n \frac{\tau^2}{\left(\tau^2 + (y_i - \mu)^2\right)^{3/2}} & \frac{n}{z^2}\frac{1}{n}\sum_{i=1}^n \frac{\tau(y_i - \mu)}{(\tau^2 + (y_i - \mu)^2)^{3/2}} \\ \frac{n}{z^2}\frac{1}{n}\sum_{i=1}^n \frac{\tau(y_i - \mu)}{(\tau^2 + (y_i - \mu)^2)^{3/2}} & \frac{n^{3/2}}{z^3}\frac{1}{n}\sum_{i=1}^n \frac{(y_i - \mu)^2}{(\tau^2 + (y_i - \mu)^2)^{3/2}} \end{bmatrix}.$$

## B  Population bias

Let $\mu^*(v)$ be the underlying pseudo-Huber regression coefficient with $v$ fixed *a priori*

$$\mu^*(v) = \operatorname*{argmin}_\mu \mathbb{E}L_n(\mu, v).$$

Recall that $\tau = v\sqrt{n}/z$. Let

$$\psi_v(x) := \nabla_x \ell^{\mathrm{P}}(x, v) = \frac{\sqrt{n}x}{z\sqrt{\tau^2 + x^2}},$$

$$h_v(x) := \nabla_x^2 \ell^{\mathrm{P}}(x, v) = \frac{\sqrt{n}}{z\sqrt{\tau^2 + x^2}} - \frac{\sqrt{n}x^2}{z(\tau^2 + x^2)^{3/2}} = \frac{\sqrt{n}\tau^2}{z(\tau^2 + x^2)^{3/2}}.$$

**Assumption 2.** The second-order derivative of $L(\mu, v) = \mathbb{E}\ell^{\mathrm{P}}(y_i - \mu, v)$ satisfies that

$$0 < \kappa_\ell \leq \nabla_\mu^2 L(\mu, v)$$

for any $\mu \in \mathbb{B}(r, \mu^*) := \{\mu : |\mu - \mu^*| \leq r\}$, where we use the same $\kappa_\ell$ as in Assumption 1 without loss of generality.

Our next proposition shows that the population bias is at the order of $\sqrt{n}/\tau^2$.

**Proposition B.1** (Population bias). Assume Assumption 2 holds with $r > \sqrt{n}\sigma^2/(2\kappa_\ell\tau^2)$. We have

$$|\mu^*(v) - \mu^*| \leq \frac{\sigma^2}{2z\kappa_\ell} \cdot \frac{\sqrt{n}}{\tau^2} \lesssim \frac{\sqrt{n}}{\tau^2}.$$

*Proof of Proposition B.1.* Define the bias term $\Delta = \mu^* - \mu^*(v)$ and the function $h_v(\mu) = n^{-1}\sum_{i=1}^n \mathbb{E}\{\ell^{\mathrm{P}}(y_i - \mu, v)\}$. We first assume that $|\Delta| \leq r$. By the first order optimality of $\mu^*(v)$, we have $\nabla h_v(\mu^*(v)) = 0$, and thus

$$\langle \Delta, \nabla^2 h_v(\widetilde{\mu})\Delta \rangle = \langle \nabla h_v(\mu^*) - \nabla h_v(\mu^*(v)), \Delta \rangle = \langle \nabla h_v(\mu^*), \Delta \rangle = -\frac{1}{n}\sum_{i=1}^n \mathbb{E}\{\psi_v(\sigma\varepsilon_i)\}\Delta, \qquad \text{(B.1)}$$

where $\widetilde{\mu} = \lambda\mu^* + (1 - \lambda)\mu^*(v)$ for some $0 \leq \lambda \leq 1$.

Since $\mathbb{E}(\varepsilon_i) = 0$, we have

$$|\mathbb{E}\{-\psi_v(\varepsilon_i)\}| = \frac{\sqrt{n}}{z} \cdot \left|\mathbb{E}\left\{\frac{-\varepsilon_i/\tau}{\sqrt{1 + \varepsilon_i^2/\tau^2}}\right\}\right| = \frac{\sqrt{n}}{z} \cdot \left|\mathbb{E}\left\{\frac{\tau^{-1}\varepsilon_i\left(\sqrt{1 + \varepsilon_i^2/\tau^2} - 1\right)}{\sqrt{1 + \varepsilon_i^2/\tau^2}}\right\}\right|$$

$$\leq \frac{\sqrt{n}}{2z} \cdot \mathbb{E}\left|\frac{(\varepsilon_i)^3/\tau^3}{\sqrt{1 + \varepsilon_i^2/\tau^2}}\right| \leq \frac{\sqrt{n}\sigma^2}{2z\tau^2}, \qquad \text{(B.2)}$$

---

**Algorithm 1** An alternating gradient descent algorithm.

---
   **Input**: $\mu_{\text{init}}, v_{\text{init}}, v_0, V_0, \eta_1, \eta_2, (y_1, \ldots, y_n)$
   **for** $k = 1, 2, \ldots$ until convergence **do**
      $\mu_{k+1} = \mu_k - \eta_1 \nabla_\mu L_n(\mu_k, v_k)$
      $\widetilde{v}_{k+1} = v_k - \eta_2 \nabla_\tau L_n(\mu_{k+1}, v_k)$ and $v_{k+1} = \min\{\max\{\widetilde{v}_{k+1}, v_0\}, V_0\}$
   **end for**
   **Output**: $\widehat{\mu} = \mu_{k+1}, \widehat{v} = v_{k+1}$

---

where the first inequality uses the inequality $\sqrt{1 + x^2} \leq 1 + x^2/2$ and the last inequality uses the fact that $\sqrt{1 + \varepsilon_i^2/\tau^2} \geq 1 \vee |\varepsilon_i|/\tau$.

Using equality (B.1) together with Assumption 2 and inequality (B.2) and canceling one $|\Delta|$ term on both sides, we obtain

$$|\Delta| \leq \frac{\sqrt{n}\sigma^2}{2z\kappa_\ell\tau^2}.$$

We then show it must hold that $|\Delta| \leq r$. If not, then we shall construct an intermediate solution between $\mu^*$ and $\mu^*(v)$, denoted by $\mu_\eta^*(v) = \mu^* + \eta(\mu^*(v) - \mu^*)$, such that $|\mu_\eta^*(v) - \mu^*| = r$. Specifically, we can choose some $\eta \in (0, 1)$ so that $|\mu_\eta^*(v) - \mu^*| = r$. We then proceed the above calculation and would obtain

$$|\mu_{\tau,\eta} - \mu^*| \leq \frac{\sqrt{n}\sigma^2}{2z\rho_\ell\tau^2} < r.$$

This is a contradiction. $\qquad\square$

## C   An alternating gradient descent algorithm

This section derives algorithms to optimize (2.5) with the constraint $v_0 \leq v \leq V_0$. Starting with initialization $v = v_{\text{init}}$ and $\mu = \mu_{\text{init}}$, we use gradient descent to alternatively update the solution sequence $\{(\mu_k, v_k) : k \geq 1\}$ where $(\mu_1, v_1) = (\mu_{\text{init}}, v_{\text{init}})$. Specifically, at working solution $(\mu_k, v_k)$, the $(k+1)$-th iteration carries out the following two steps

1. $\mu_{k+1} = \mu_k - \eta_1 \nabla_\mu L_n(\mu_k, v_k)$,

2. $\widetilde{v}_{k+1} = v_k - \eta_2 \nabla_\tau L_n(\mu_{k+1}, v_k)$ and $v_{k+1} = \min\{\max\{\widetilde{v}_{k+1}, v_0\}, V_0\}$,

where $\eta_1$ and $\eta_2$ are the learning rates and

$$\nabla_\mu L_n(\mu, v) = -\frac{1}{n}\sum_{i=1}^{n}\frac{y_i - \mu}{v\sqrt{1 + z^2(y_i - \mu)^2/(nv^2)}},$$

$$\nabla_v L_n(\mu, v) = \frac{1}{n}\sum_{i=1}^{n}\frac{n/z^2}{\sqrt{1 + z^2(y_i - \mu)^2/(nv^2)}} - \left(\frac{n}{z^2} - a\right).$$

We then repeat the above two steps until convergence. We summarize the details in Algorithm 1. In practice, the learning rate $\eta_1$ and $\eta_2$ can be chosen adaptively. Specifically, in our experiments, we use alternating gradient descent with the Barzilai and Borwein method and backtracking line search.

## D   Comparing with Lepski's method

We compare our method with Lepski's method. The idea of Lepski's method is very simple: consider a sequence of confidence intervals obtained by assuming that the variance is bounded by a sequence of bounds $v_k$ and pick up as an estimator the middle of the smallest interval intersecting all the larger ones.

We will use Lepski's method to tune the robustification parameter $v$ and thus $\tau = v\sqrt{n}/z$ in the empirical pseudo-Huber loss:

$$L_n^{\mathrm{h}}(\mu, \tau) = \frac{1}{n} \sum_{i=1}^{n} \left( \tau \sqrt{\tau^2 + (y_i - \mu)^2} - \tau^2 \right).$$

Let $v_{\max}$ be an upper bound for $\sigma$, and $\tau_{\max} = v_{\max}\sqrt{n}/z$ with $z = \sqrt{\log(1/\delta)}$. Let $n$ be sufficiently large. Then with probability at least $1 - \delta$, we have

$$|\widetilde{\mu}(v_{\max}) - \mu^*| \le 6\, v_{\max} \sqrt{\frac{\log(4/\delta)}{n}} =: \epsilon(v_{\max}, \delta),$$

where $\widetilde{\mu}(v_{\max}) = \operatorname{argmin}_\mu L_n(\mu, \tau_{\max})$. Let $\epsilon(v_{\max}, 0) = +\infty$ by convention. Clearly, $\epsilon(v_{\max}, \delta)$ is homogeneous:

$$\epsilon(v_{\max}, \delta) = B(\delta) v_{\max}, \text{ with } B(\delta) = 6\sqrt{\frac{\log(4/\delta)}{n}}.$$

For some parameters $V \in \mathbb{R}$, $\rho > 1$, and $s \in \mathbb{N}$, choose for $\mathcal{V}$ the following distribution for $v_{\max}$

$$\mathcal{V}(v_{\max}) = \begin{cases} \frac{1}{2s+1}, & \text{if } v_{\max} = V\rho^k, \ k \in \mathbb{Z}, \ |k| \le s, \\ 0, & \text{otherwise.} \end{cases}$$

Consider for any $v_{\max}$ such that $\epsilon(v_{\max}, \delta v(v_{\max})) < \infty$ the confidence interval

$$I(v_{\max}) = \widetilde{\mu}(v_{\max}) + \epsilon(v_{\max}, \delta v(v_{\max})) \times [-1, 1],$$

where $\epsilon(v_{\max}, \delta v(v_{\max})) = 6v_{\max}\sqrt{\frac{\log(4/\delta) + \log(2s+1)}{n}}$. We set $I(v_{\max}) = \mathbb{R}$ when $\epsilon(v_{\max}, \delta v(v_{\max})) = +\infty$.

Let us consider the non-decreasing family of closed intervals

$$J(v_1) = \bigcap \left\{ I(v_{\max}) : v_{\max} \ge v_1 \right\}, \ v_1 \in \mathbb{R}_+.$$

Lepski's method picks the center point of the intersection

$$\bigcap \left\{ J(v_1) : v_1 \in \mathbb{R}_+, \ J(v_1) \ne \emptyset \right\}$$

to be the final estimator $\widehat{\mu}_{\mathrm{Lepski}}$. Then the following result holds.

**Proposition D.1.** Suppose $|\log(\sigma/V)| \le 2s\log(\rho)$. Then with probability at least $1 - \delta$

$$|\widehat{\mu}_{\mathrm{Lepski}} - \mu^*| \le 12\rho\sigma \sqrt{\frac{\log(4/\delta) + \log(2s+1)}{n}}.$$

If we take the grid fine enough such that $s = n$, then the deviation bound above reduces to

$$12\rho\sigma \sqrt{\frac{\log(4/\delta) + \log(2n+1)}{n}},$$

which agrees with deviation bound for our proposed estimator, up to a constant multiplier. Therefore, our proposed estimator is comparable to Lepski's method in terms of deviation bound. Computationally, our estimator is self-tuned and is thus computationally more efficient than Lepski's method.

# E   Proofs for Section 2

## E.1   Proofs for Theorem 2.3

*Proof of Theorem 2.3.* We prove first the finite-sample result and then the asymptotic result. Recall that $\tau_* = v_*\sqrt{n}/z$.

**Proving the finite-sample result.** On one side, if $v_* = 0$ and by the definition of $v_*$, $v_*$ satisfies

$$1 - \frac{az^2}{n} = \mathbb{E}\frac{\sqrt{n}v_*}{\sqrt{nv_*^2 + z^2\varepsilon^2}} = 0,$$

which is a contradiction. Thus $v_* > 0$. Using the convexity of $1/\sqrt{1+x}$ for $x > -1$ and Jensen's inequality acquires

$$1 - \frac{az^2}{n} = \mathbb{E}\frac{\sqrt{n}v_*}{\sqrt{nv_*^2 + z^2\varepsilon^2}} = \mathbb{E}\frac{1}{\sqrt{1 + z^2\varepsilon^2/(nv_*^2)}} \geq \frac{1}{\sqrt{1 + z^2\sigma^2/(nv_*^2)}} \geq 1 - \frac{z^2\sigma^2}{2nv_*^2},$$

where the last inequality uses the inequality $(1+x)^{-1/2} \geq 1 - x/2$, that is Lemma J.4 (i) with $r = -1/2$. This implies

$$v_*^2 \leq \frac{\sigma^2}{2a}.$$

On the other side, using the concavity of $\sqrt{x}$, we obtain, for any $\gamma \in [0,1)$, that

$$1 - \frac{az^2}{n} = \mathbb{E}\frac{\sqrt{n}v_*}{\sqrt{nv_*^2 + z^2\varepsilon^2}} = \mathbb{E}\frac{1}{\sqrt{1 + \sigma^2 z^2\varepsilon^2/(nv_*^2)}}$$

$$\leq \sqrt{\mathbb{E}\left(\frac{1}{1 + z^2\varepsilon^2/(nv_*^2)}\right)}$$

$$\leq \sqrt{\mathbb{E}\left\{\left(1 - (1-\gamma)\frac{z^2\varepsilon^2}{nv_*^2}\right)1\left(\frac{z^2\varepsilon^2}{nv_*^2} \leq \frac{\gamma}{1-\gamma}\right) + \frac{1}{1 + z^2\varepsilon^2/(nv_*^2)}1\left(\frac{z^2\varepsilon^2}{nv_*^2} > \frac{\gamma}{1-\gamma}\right)\right\}}$$

$$\leq \sqrt{1 - (1-\gamma)\mathbb{E}\left\{\frac{z^2\varepsilon^2}{nv_*^2}1\left(\frac{z^2\varepsilon^2}{nv_*^2} \leq \frac{\gamma}{1-\gamma}\right)\right\}}$$

$$\leq \sqrt{1 - (1-\gamma)\frac{\mathbb{E}\left\{\varepsilon^2 1\left(\varepsilon^2 \leq \gamma\tau_*^2/(1-\gamma)\right)\right\}}{nv_*^2/z^2}}, \tag{E.1}$$

where the second inequality uses Lemma E.1, that is,

$$(1+x)^{-1} \leq 1 - (1-\gamma)x, \text{ for any } x \in \left[0, \frac{\gamma}{1-\gamma}\right].$$

Taking square on both sides of (E.1) and using the fact that $n \geq az^2$ together with Lemma J.4 (i) with $r = 2$, aka $(1+x)^r \geq 1 + rx$ for $x \geq -1$ and $r \in \mathbb{R} \setminus (0,1)$, we obtain

$$1 - \frac{2az^2}{n} \leq \left(1 - \frac{az^2}{n}\right)^2 \leq 1 - (1-\gamma)\frac{\mathbb{E}\{\varepsilon^2 1(\varepsilon^2 \leq \gamma\tau_*^2/(1-\gamma))\}}{nv_*^2/z^2},$$

or equivalently

$$v_*^2 \geq \frac{\sigma_{\varphi\tau_*^2}^2}{2a},$$

where $\varphi = \gamma/(1-\gamma)$. Combining the upper bound and the lower bound for $v_*^2$ completes the proof for the finite-sample result.

**Proving the asymptotic result.** The above implies that $v_* < \infty$ for any $n \geq az^2$. By the definition of $v_*$, we have

$$\frac{az^2}{n} = 1 - \mathbb{E}\frac{1}{\sqrt{1 + z^2\varepsilon^2/(nv_*^2)}}. \tag{E.2}$$

We must have $nv_*^2/z^2 \to \infty$. Otherwise assume $\limsup_{n\to\infty} nv_*^2/z^2 \leq M < \infty$. Taking $n \to \infty$, the left hand side of the above equality goes to 0 while the right hand is lower bounded as

$$
\begin{aligned}
1 - \mathbb{E}\frac{1}{\sqrt{1+\varepsilon^2/M}} &\geq 1 - \sqrt{\mathbb{E}\left(\frac{1}{1+\varepsilon^2/M}\right)} \\
&\geq 1 - \sqrt{1 - \frac{\mathbb{E}\{\varepsilon^2 1(\varepsilon^2 \leq M)\}}{2M}} \\
&\geq 1 - \sqrt{\frac{1}{2}} > 0,
\end{aligned}
$$

where the first two inequalities follow from (E.1) with $\gamma = 1/2$ and the third inequality uses the fact that $\mathbb{E}\{\varepsilon^2 1(\varepsilon^2 \leq M)\} \leq M$. This is a contradiction. Thus $nv_*^2/z^2 \to \infty$. Multiplying both sides of the above equality by $n$, taking $n \to \infty$, and using the dominated convergence theorem, we obtain

$$
\begin{aligned}
az^2 &= \lim_{n\to\infty} \mathbb{E}\left(n \cdot \frac{\sqrt{1+z^2\varepsilon^2/(nv_*^2)}-1}{\sqrt{1+z^2\varepsilon^2/(nv_*^2)}}\right) \\
&= \lim_{n\to\infty} \mathbb{E}\left(n \cdot \frac{1}{\sqrt{1+z^2\varepsilon^2/(nv_*^2)}} \cdot \frac{\sqrt{1+z^2\varepsilon^2/(nv_*^2)}-1}{z^2\varepsilon^2/(2nv_*^2)} \cdot \frac{z^2\varepsilon^2}{2nv_*^2}\right) \\
&= \frac{\mathbb{E}z^2\varepsilon^2}{2\lim_{n\to\infty} v_*^2},
\end{aligned}
$$

and thus $\lim_{n\to\infty} v_*^2 = \sigma^2/(2a)$.

$\square$

## E.2 Proof of Proposition 2.4

*Proof of Proposition 2.4.* The convexity proof consists of two steps: (1) prove that $L_n(\mu, v)$ is jointly convex in $\mu$ and $v$; (2) prove that $L_n(\mu, v)$ is strictly convex, provided that there are at least two distinct data points. To show that $L_n(\mu, v) = n^{-1}\sum_{i=1}^n \ell^{\mathrm{P}}(y_i - \mu, v)$ in (2.5) is jointly convex in $\mu$ and $v$, it suffices to show that each $\ell^{\mathrm{P}}(y_i - \mu, v)$ is jointly convex in $\mu$ and $v$. Recall that $\tau = v\sqrt{n}/z$. The Hessian matrix of $\ell^{\mathrm{P}}(y_i - \mu, v)$ is

$$
H_i(\mu, v) = \frac{\sqrt{n}}{z} \cdot \frac{1}{\left(\tau^2 + (y_i - \mu)^2\right)^{3/2}} \begin{bmatrix} \tau^2 & (\sqrt{n}/z)\,\tau(y_i - \mu) \\ (\sqrt{n}/z)\,\tau(y_i - \mu) & (\sqrt{n}/z)^2\,(y_i - \mu)^2 \end{bmatrix} \succeq 0,
$$

which is positive semi-definite. Thus $L_n(\mu, v)$ is jointly convex in $\mu$ and $v$.

We proceed to show (2). Because the Hessian matrix $H(\mu, v)$ of $L_n(\mu, v)$ satisfies $H(\mu, v) = n^{-1}\sum_{i=1}^n H_i(\mu, v)$ and each $H_i(\mu, v)$ is positive semidefinite, we only need to show $H(\mu, v)$ is of full rank. Without generality, assume that $y_1 \neq y_2$. Then

$$
H_1(\mu, v) + H_2(\mu, v) = \frac{\sqrt{n}}{z} \cdot \sum_{i=1}^2 \frac{1}{\left(\tau^2 + (y_i - \mu)^2\right)^{3/2}} \begin{bmatrix} \tau^2 & (\sqrt{n}/z)\,\tau(y_i - \mu) \\ (\sqrt{n}/z)\,\tau(y_i - \mu) & (\sqrt{n}/z)^2\,(y_i - \mu)^2 \end{bmatrix}.
$$

Some algebra yields

$$
\det\left(H_1(\mu, v) + H_2(\mu, v)\right) = \frac{n^2\tau^2}{z^4} \cdot \frac{(y_1 - y_2)^2}{(\tau^2 + (y_1 - \mu)^2)^{3/2}(\tau^2 + (y_2 - \mu)^2)^{3/2}} \neq 0
$$

for any $v > 0$, and thus $\tau > 0$, and $\mu \in \mathbb{R}$, provided that $y_1 \neq y_2$. Therefore, $H_1(\mu, v) + H_2(\mu, v)$ is of full rank and thus is $H(\mu, \tau)$, provided $v > 0$, $\mu \in \mathbb{R}$, and $y_1 \neq y_2$.

$\square$

### E.3 Supporting lemmas

**Lemma E.1.** Let $0 \leq \gamma < 1$. For any $0 \leq x \leq \gamma/(1-\gamma)$, we have

$$(1+x)^{-1} \leq 1 - (1-\gamma)x.$$

*Proof of Lemma E.1.* To prove the lemma, it suffices to prove, for any $\gamma \in [0,1)$, that

$$1 \leq (1+x) - (1-\gamma)x(1+x), \quad \forall \, 0 \leq x \leq \frac{\gamma}{1-\gamma},$$

which is equivalently to

$$x\left(x - \frac{\gamma}{1-\gamma}\right) \leq 0, \quad \forall \, 0 \leq x \leq \frac{\gamma}{1-\gamma}.$$

The above inequality always holds, and this completes the proof.

$\square$

## F  Proofs for the fixed $v$ case

This section collects proofs for Theorem 3.1, Lemma 3.2, and Corollary 3.3. Recall that $\tau = v\sqrt{n}/z$, and the gradients with respect to $\mu$ and $v$ are

$$\nabla_\mu L_n(\mu, v) = -\frac{1}{n}\sum_{i=1}^{n}\frac{y_i - \mu}{v\sqrt{1 + z^2(y_i - \mu)^2/(nv^2)}} = -\frac{\sqrt{n}}{z} \cdot \frac{1}{n}\sum_{i=1}^{n}\frac{y_i - \mu}{\sqrt{\tau^2 + (y_i - \mu)^2}},$$

$$\nabla_v L_n(\mu, v) = \frac{1}{n}\sum_{i=1}^{n}\frac{n/z^2}{\sqrt{1 + z^2(y_i - \mu)^2/(nv^2)}} - \left(\frac{n}{z^2} - a\right) = \frac{n}{z^2} \cdot \frac{1}{n}\sum_{i=1}^{n}\left(\frac{\tau}{\sqrt{\tau^2 + (y_i - \mu)^2}} - 1\right) + a.$$

### F.1  Proof of Theorem 3.1

*Proof of Theorem 3.1.* Let $\tau = v\sqrt{n}/z$. Because $\widehat{\mu}(v)$ is the stationary point of $L_n(\mu, v)$, we have

$$\frac{\partial}{\partial\mu}L_n(\widehat{\mu}(v), v) = -\frac{1}{n}\sum_{i=1}^{n}\frac{y_i - \widehat{\mu}(v)}{v\sqrt{1 + z^2(y_i - \widehat{\mu}(v))^2/(nv^2)}} = -\frac{\sqrt{n}}{z} \cdot \frac{1}{n}\sum_{i=1}^{n}\frac{y_i - \widehat{\mu}(v)}{\sqrt{\tau^2 + (y_i - \widehat{\mu}(v))^2}} = 0.$$

Let $\Delta = \widehat{\mu}(v) - \mu$. We first assume that $|\Delta| := |\widehat{\mu}(v) - \mu^*| \leq r_0 \leq r$. Using Assumption 1 obtains

$$\kappa_\ell|\widehat{\mu}(v) - \mu^*|^2 \leq \left\langle \frac{\partial}{\partial\mu}L_n(\widehat{\mu}(v), v) - \frac{\partial}{\partial\mu}L_n(\mu^*, v), \widehat{\mu}(v) - \mu^* \right\rangle$$

$$\leq \left|\frac{1}{\sqrt{n}}\sum_{i=1}^{n}\frac{\varepsilon_i}{z\sqrt{\tau^2 + \varepsilon_i^2}}\right| |\widehat{\mu}(v) - \mu^*|,$$

or equivalently

$$\kappa_\ell|\widehat{\mu}(v) - \mu^*| \leq \left|\frac{1}{\sqrt{n}}\sum_{i=1}^{n}\frac{\varepsilon_i}{z\sqrt{\tau^2 + \varepsilon_i^2}}\right|.$$

Applying Lemma F.1 with the fact that $\left|\mathbb{E}\left(\tau\varepsilon_i/(\tau^2 + \varepsilon_i^2)^{1/2}\right)\right| \leq \sigma^2/(2\tau)$, we obtain with probability at least $1 - 2\delta$

$$\kappa_\ell|\widehat{\mu}(v) - \mu^*| \leq \left|\frac{\sqrt{n}}{\tau}\frac{1}{n}\sum_{i=1}^{n}\frac{\tau\varepsilon_i}{z\sqrt{\tau^2 + \varepsilon_i^2}}\right| \leq \frac{\sqrt{n}}{z\tau}\left(\sigma\sqrt{\frac{2\log(1/\delta)}{n}} + \frac{\tau\log(1/\delta)}{3n} + \frac{\sigma^2}{2\tau}\right),$$

or equivalently

$$\kappa_\ell |\widehat{\mu}(v) - \mu^*| \leq \sqrt{\frac{2\log(1/\delta)}{z^2\tau^2/\sigma^2}} + \frac{\log(1/\delta)}{3z\sqrt{n}} + \frac{\sqrt{n}\sigma^2}{2z\tau^2}.$$

Since $\tau = v\sqrt{n}/z$, we have

$$\kappa_\ell |\widehat{\mu}(v) - \mu^*| \leq \left( \frac{\sqrt{2}\sigma}{v} + \frac{\sqrt{\log(1/\delta)}}{3z} \right) \sqrt{\frac{\log(1/\delta)}{n}} + \frac{1}{2} \cdot \frac{\sigma^2}{v^2} \cdot \frac{z}{\sqrt{n}}.$$

Taking $z = \sqrt{\log(w/\delta)}$ then yields

$$\begin{aligned}
\kappa_\ell |\widehat{\mu}(\tau) - \mu^*| &\leq \left( \frac{\sqrt{2}\sigma}{v} + \frac{\sqrt{\log(1/\delta)}}{3\sqrt{\log(w/\delta)}} \right) \sqrt{\frac{\log(1/\delta)}{n}} + \frac{1}{2} \cdot \frac{\sigma^2}{v^2} \cdot \sqrt{\frac{\log(w/\delta)}{n}} \\
&\leq \left( \frac{\sqrt{2}\sigma}{v} + \frac{1}{3} + \frac{1}{2} \cdot \frac{\sigma^2}{v^2} \right) \sqrt{\frac{\log(w/\delta)}{n}} \\
&< \left( 1 + \frac{\sigma}{\sqrt{2}v} \right)^2 \sqrt{\frac{\log(w/\delta)}{n}} =: \kappa_\ell r_0 \leq \kappa_\ell r
\end{aligned}$$

for any $0 \leq \delta < 1$. Moving $\kappa_\ell$ to the right hand side obtains the desired bound.

We then show that $|\Delta| \leq r_0$ must hold. If not, we shall construct an intermediate solution between $\mu^*$ and $\widehat{\mu}(\tau)$, denoted by $\mu_\eta = \mu^* + \eta(\widehat{\mu}(\tau) - \mu^*)$, such that $|\mu_\eta - \mu^*| = r_0$. Specifically, we can choose some $\eta \in (0, 1)$ so that $|\mu_\eta - \mu^*| = r_0$. We then repeat the above calculation and obtain

$$\begin{aligned}
|\widehat{\mu}(\tau) - \mu^*| &\leq \frac{1}{\kappa_\ell} \cdot \left( \frac{\sqrt{2}\sigma}{v} + \frac{1}{3} + \frac{1}{2} \cdot \frac{\sigma^2}{v^2} \right) \sqrt{\frac{\log(w/\delta)}{n}} \\
&< r_0 = \frac{1}{\kappa_\ell} \cdot \left( 1 + \frac{\sigma}{\sqrt{2}v} \right)^2 \sqrt{\frac{\log(w/\delta)}{n}}
\end{aligned}$$

which is a contradiction. Thus it must hold that $|\Delta| \leq r_0$. Taking $w = 1$ and using a change of variable $2\delta \to \delta$ complete the proof.

$\square$

## F.2 Proof of Lemma 3.2

*Proof of Lemma 3.2.* We prove that, with probability at least $1 - \delta$, Assumption 1 with $\kappa_\ell = 1/(2v)$ holds uniformly over $v \geq v_0$. Recall that $\tau = v\sqrt{n}/z$. For notational simplicity, let $\Delta = \mu - \mu^*$ and $\tau_{v_0} = v_0\sqrt{n}/z$. It follows that

$$\begin{aligned}
\langle \nabla_\mu L_n(\mu, v) - \nabla_\mu L_n(\mu^*, v), \Delta \rangle &= \left\langle \frac{1}{\sqrt{n}} \sum_{i=1}^n \frac{\varepsilon_i}{z\sqrt{\tau^2 + \varepsilon_i^2}} - \frac{1}{\sqrt{n}} \sum_{i=1}^n \frac{y_i - \mu}{z\sqrt{\tau^2 + (y_i - \mu)^2}}, \Delta \right\rangle \\
&= \frac{1}{\sqrt{n}} \sum_{i=1}^n \frac{\tau^2}{z(\tau^2 + (y_i - \widetilde{\mu})^2)^{3/2}} \Delta^2,
\end{aligned}$$

where $\widetilde{\mu}$ is some convex combination of $\mu^*$ and $\mu$, that is $\widetilde{\mu} = (1 - \lambda)\mu^* + \lambda\mu$ for some $\lambda \in [0, 1]$. Obviously we have $|\widetilde{\mu} - \mu^*| = \lambda|\Delta| \leq |\Delta| \leq r_0$. Since $(y_i - \widetilde{\mu})^2 \leq 2\varepsilon_i^2 + 2\lambda^2\Delta^2 \leq 2\varepsilon_i^2 + 2\Delta^2 \leq 2\varepsilon_i^2 + 2r_0^2$ the above

displayed equality implies

$$
\inf_{\mu \in \mathbb{B}_r(\mu^*)} \frac{\langle \nabla_\mu L_n(\mu, v) - \nabla_\mu L_n(\mu^*, v), \mu - \mu^* \rangle}{|\mu - \mu^*|^2}
$$

$$
\geq \frac{\sqrt{n}}{z} \cdot \frac{1}{n} \sum_{i=1}^{n} \frac{\tau^2}{z(\tau^2 + 2r_0^2 + 2\varepsilon_i^2)^{3/2}}
$$

$$
= \frac{\sqrt{n}}{z} \cdot \frac{\tau^2}{(\tau^2 + 2r_0^2)^{3/2}} \cdot \frac{1}{n} \sum_{i=1}^{n} \frac{(\tau^2 + 2r_0^2)^{3/2}}{(\tau^2 + 2r_0^2 + 2\varepsilon_i^2)^{3/2}}
$$

$$
\geq \frac{\sqrt{n}}{z} \cdot \frac{\tau^2}{(\tau^2 + 2r_0^2)^{3/2}} \cdot \left( \mathbb{E} \frac{(\tau_{v_0}^2 + 2r_0^2)^{3/2}}{(\tau_{v_0}^2 + 2r_0^2 + 2\varepsilon_i^2)^{3/2}} - \sqrt{\frac{\log(1/\delta)}{2n}} \right)
$$

$$
= \frac{\sqrt{n}}{z} \cdot \frac{\tau^2}{(\tau^2 + 2r_0^2)^{3/2}} \cdot \left( \mathrm{I} - \sqrt{\frac{\log(1/\delta)}{2n}} \right), \tag{F.1}
$$

where the last inequality uses Lemma F.2.

It remains to lower bound I. Using the convexity of $1/(1+x)^{3/2}$ and Jensen's inequality, we obtain

$$
\frac{1}{n} \sum_{i=1}^{n} \mathbb{E} \frac{(\tau_{v_0}^2 + 2r^2)^{3/2}}{(\tau_{v_0}^2 + 2r^2 + 2\varepsilon_i^2)^{3/2}} = \mathbb{E} \frac{(\tau_{v_0}^2 + 2r^2)^{3/2}}{(\tau_{v_0}^2 + 2r^2 + 2\varepsilon_i^2)^{3/2}}
$$

$$
= \mathbb{E} \frac{1}{(1 + 2\varepsilon_i^2/(\tau_{v_0}^2 + 2r^2))^{3/2}}
$$

$$
\geq \frac{1}{(1 + 2\sigma^2/(\tau_{v_0}^2 + 2r^2))^{3/2}}
$$

$$
= \frac{(\tau_{v_0}^2 + 2r^2)^{3/2}}{(\tau_{v_0}^2 + 2r^2 + 2\sigma^2)^{3/2}}.
$$

Plugging the above lower bound into (F.1) and using the facts that

$$
\frac{\tau^3}{(\tau^2 + 2r^2)^{3/2}} \geq \frac{\tau_{v_0}^3}{(\tau_{v_0}^2 + 2r^2)^{3/2}} \quad \text{for } \tau_{v_0} \geq \tau \quad \text{and} \quad \frac{\tau^3}{(\tau^2 + 2r^2)^{3/2}} \leq 1,
$$

we obtain with probability at least $1 - \delta$

$$
\inf_{\mu \in \mathbb{B}_r(\mu^*)} \frac{\langle \nabla_\mu L_n(\mu) - \nabla_\mu L_n(\mu^*), \mu - \mu^* \rangle}{|\mu - \mu^*|^2}
$$

$$
\geq \frac{\sqrt{n}}{z} \cdot \frac{\tau^2}{(\tau^2 + 2r^2)^{3/2}} \cdot \left( \frac{(\tau_{v_0}^2 + 2r^2)^{3/2}}{(\tau_{v_0}^2 + 2r^2 + 2\sigma^2)^{3/2}} - \sqrt{\frac{\log(1/\delta)}{2n}} \right)
$$

$$
= \frac{\sqrt{n}}{z\tau} \left( \frac{\tau^3}{(\tau^2 + 2r^2)^{3/2}} \cdot \frac{(\tau_{v_0}^2 + 2r^2)^{3/2}}{(\tau_{v_0}^2 + 2r^2 + 2\sigma^2)^{3/2}} - \frac{\tau^3}{(\tau^2 + 2r^2)^{3/2}} \cdot \sqrt{\frac{\log(1/\delta)}{2n}} \right)
$$

$$
\geq \frac{\sqrt{n}}{z\tau} \left( \frac{1}{(1 + (2r^2 + 2\sigma^2)/\tau_{v_0}^2)^{3/2}} - \sqrt{\frac{\log(1/\delta)}{2n}} \right)
$$

$$
= \frac{1}{v} \left( \frac{1}{(1 + (2r^2 + 2\sigma^2)/\tau_{v_0}^2)^{3/2}} - \sqrt{\frac{\log(1/\delta)}{2n}} \right)
$$

$$
\geq \frac{1}{2v}
$$

provided $\tau_{v_0}^2 \geq 4r^2 + 4\sigma^2$ and $n \geq C \log(1/\delta)$ for some large enough absolute constant $C$.

$\square$

### F.3 Proof of Corollary 3.3

*Proof of Corollary 3.3.* Recall $z = \sqrt{\log(w/\delta)}$ and let

$$r = 2v \left( \frac{\sigma}{\sqrt{2}v} + 1 \right)^2 \sqrt{\frac{\log(2w/\delta)}{n}}.$$

If $n \geq C \max \left\{ (r^2 + \sigma^2)/v_0^2, 1 \right\} \log(1/\delta)$, which is guaranteed by the conditions of the corollary, then with probability at least $1 - \delta$, Assumption 1 holds with $\kappa_\ell = 1/(2v)$ uniformly over $v \geq v_0$. Denote this probability event by $\mathcal{E}$. If Assumption 1 holds, then by Theorem 3.1, we have

$$\mathbb{P} \left( |\widehat{\mu}(v) - \mu^*| \leq 2v \left( \frac{\sigma}{\sqrt{2}v} + 1 \right)^2 \sqrt{\frac{\log(2w/\delta)}{n}} \,\Big|\, \mathcal{E} \right) \geq 1 - \delta.$$

Thus

$$\mathbb{P} \left( |\widehat{\mu}(v) - \mu^*| > 2v \left( \frac{\sigma}{\sqrt{2}v} + 1 \right)^2 \sqrt{\frac{\log(2w/\delta)}{n}} \right)$$

$$= \mathbb{P} \left( |\widehat{\mu}(v) - \mu^*| > 2v \left( \frac{\sigma}{\sqrt{2}v} + 1 \right)^2 \sqrt{\frac{\log(2w/\delta)}{n}}, \mathcal{E} \right)$$

$$+ \mathbb{P} \left( |\widehat{\mu}(v) - \mu^*| > 2v \left( \frac{\sigma}{\sqrt{2}v} + 1 \right)^2 \sqrt{\frac{\log(2w/\delta)}{n}}, \mathcal{E}^c \right)$$

$$\leq \mathbb{P} \left( |\widehat{\mu}(v) - \mu^*| > 2v \left( \frac{\sigma}{\sqrt{2}v} + 1 \right)^2 \sqrt{\frac{\log(2w/\delta)}{n}} \,\Big|\, \mathcal{E} \right) + \mathbb{P} \left( \mathcal{E}^c \right)$$

$$\leq 2\delta.$$

Then with probability at least $1 - 2\delta$, we have

$$|\widehat{\mu}(v) - \mu^*| \leq 2v \left( \frac{\sigma}{\sqrt{2}v} + 1 \right)^2 \sqrt{\frac{\log(2w/\delta)}{n}}.$$

Using a change of variable $2\delta \to \delta$ finishes the proof. $\qquad\square$

### F.4 Supporting lemmas

This subsection collects two supporting lemmas that are used earlier in this section.

**Lemma F.1.** Let $\varepsilon_i$ be i.i.d. random variables such that $\mathbb{E}\varepsilon_i = 0$ and $\mathbb{E}\varepsilon_i^2 = 1$. For any $0 \leq \delta \leq 1$, with probability at least $1 - 2\delta$, we have

$$\left| \frac{1}{n} \sum_{i=1}^n \frac{\tau \varepsilon_i}{\sqrt{\tau^2 + \varepsilon_i^2}} - \mathbb{E} \frac{\tau \varepsilon_i}{\sqrt{\tau^2 + \varepsilon_i^2}} \right| \leq \sigma \sqrt{\frac{2\log(1/\delta)}{n}} + \frac{\tau \log(1/\delta)}{3n}.$$

*Proof of Lemma F.1.* The random variables $Z_i := \tau \psi_\tau(\varepsilon_i) = \tau \varepsilon_i / (\tau^2 + \varepsilon_i^2)^{1/2}$ with $\mu_z = \mathbb{E}Z_i$ and $\sigma_z^2 = \text{var}(Z_i)$ are bounded i.i.d. random variables such that

$$|Z_i| = \left| \tau \varepsilon_i / (\tau^2 + \varepsilon_i^2)^{1/2} \right| \leq |\varepsilon_i| \wedge \tau \leq \tau,$$

$$|\mu_z| = |\mathbb{E}Z_i| = \left| \mathbb{E} \left( \tau \varepsilon_i / (\tau^2 + \varepsilon_i^2)^{1/2} \right) \right| \leq \frac{\sigma^2}{2\tau},$$

$$\mathbb{E}Z_i^2 = \mathbb{E} \left( \frac{\tau^2 \varepsilon_i^2}{\tau^2 + \varepsilon_i^2} \right) \leq \sigma^2,$$

$$\sigma_z^2 := \text{var}(Z_i) = \mathbb{E} \left( \tau \varepsilon_i / (\tau^2 + \varepsilon_i^2)^{1/2} - \mu_z \right)^2$$

$$= \mathbb{E} \left( \frac{\tau^2 \varepsilon_i^2}{\tau^2 + \varepsilon_i^2} \right) - \mu_z^2 \leq \sigma^2.$$

For third and higher order absolute moments, we have

$$\mathbb{E}|Z_i|^k = \mathbb{E}\left|\frac{\tau \varepsilon_i}{\sqrt{\tau^2 + \varepsilon_i^2}}\right|^k \leq \sigma^2 \tau^{k-2} \leq \frac{k!}{2}\sigma^2(\tau/3)^{k-2}, \text{ for all integers } k \geq 3.$$

Using Lemma J.2 with $v = n\sigma^2$ and $c = \tau/3$, we have for any $t \geq 0$

$$\mathbb{P}\left(\left|\sum_{i=1}^n \frac{\tau \varepsilon_i}{\sqrt{\tau^2 + \varepsilon_i^2}} - \sum_{i=1}^n \mathbb{E}\frac{\tau \varepsilon_i}{\sqrt{\tau^2 + \varepsilon_i^2}}\right| \geq \sqrt{2n\sigma^2 t} + \frac{\tau t}{3}\right) \leq 2\exp(-t).$$

Taking $t = \log(1/\delta)$ acquires that for any $0 \leq \delta \leq 1$

$$\mathbb{P}\left(\left|\frac{1}{n}\sum_{i=1}^n \frac{\tau \varepsilon_i}{\sqrt{\tau^2 + \varepsilon_i^2}} - \frac{1}{n}\sum_{i=1}^n \mathbb{E}\frac{\tau \varepsilon_i}{\sqrt{\tau^2 + \varepsilon_i^2}}\right| \leq \sigma\sqrt{\frac{2\log(1/\delta)}{n}} + \frac{\tau \log(1/\delta)}{3n}\right) \geq 1 - 2\delta.$$

This completes the proof.

$\qquad\square$

**Lemma F.2.** For any $0 < \delta < 1$, with probability at least $1 - \delta$,

$$\frac{1}{n}\sum_{i=1}^n \frac{\tau^3}{(\tau^2 + \varepsilon_i^2)^{3/2}} - \mathbb{E}\frac{\tau^3}{(\tau^2 + \varepsilon_i^2)^{3/2}} \geq -\sqrt{\frac{\log(1/\delta)}{2n}}.$$

Moreover, with probability at least $1 - \delta$, it holds uniformly over $\tau \geq \tau_{v_0} \geq 0$ that

$$\frac{1}{n}\sum_{i=1}^n \frac{\tau^3}{(\tau^2 + \varepsilon_i^2)^{3/2}} \geq \mathbb{E}\frac{\tau_{v_0}^3}{(\tau_{v_0}^2 + \varepsilon_i^2)^{3/2}} - \sqrt{\frac{\log(1/\delta)}{2n}}.$$

*Proof of Lemma F.2.* The random variables $Z_i = Z_i(\tau) := \tau^3/(\tau^2 + \varepsilon_i^2)^{3/2}$ with $\mu_z = \mathbb{E}Z_i$ and $\sigma_z^2 = \text{var}(Z_i)$ are bounded i.i.d. random variables such that

$$0 \leq Z_i = \tau^3/(\tau^2 + \varepsilon_i^2)^{3/2} \leq 1.$$

Therefore, using Lemma J.1 with $v = n$ acquires that for any $t \geq 0$

$$\mathbb{P}\left(\sum_{i=1}^n \frac{\tau^3}{(\tau^2 + \varepsilon_i^2)^{3/2}} - \sum_{i=1}^n \mathbb{E}\left(\frac{\tau^3}{(\tau^2 + \varepsilon_i^2)^{3/2}}\right) \leq -\sqrt{\frac{nt}{2}}\right) \leq \exp(-t).$$

Taking $t = \log(1/\delta)$ acquires that for any $0 < \delta \leq 1$

$$\mathbb{P}\left(\frac{1}{n}\sum_{i=1}^n \frac{\tau^3}{(\tau^2 + \varepsilon_i^2)^{3/2}} - \frac{1}{n}\sum_{i=1}^n \mathbb{E}\left(\frac{\tau^3}{(\tau^2 + \varepsilon_i^2)^{3/2}}\right) > -\sqrt{\frac{\log(1/\delta)}{2n}}\right) > 1 - \delta.$$

The second result follows from the fact that $Z_i(\tau)$ is an increasing function of $\tau$. Specifically, we have with probability at least $1 - \delta$

$$\frac{1}{n}\sum_{i=1}^n \frac{\tau^3}{(\tau^2 + \varepsilon_i^2)^{3/2}} \geq \frac{1}{n}\sum_{i=1}^n \frac{\tau_{v_0}^3}{(\tau_{v_0}^2 + \varepsilon_i^2)^{3/2}}$$

$$\geq \mathbb{E}\left(\frac{\tau_{v_0}^3}{(\tau_{v_0}^2 + \varepsilon_i^2)^{3/2}}\right) + \frac{1}{n}\sum_{i=1}^n \frac{\tau_{v_0}^3}{(\tau_{v_0}^2 + \varepsilon_i^2)^{3/2}} - \mathbb{E}\left(\frac{\tau_{v_0}^3}{(\tau_{v_0}^2 + \varepsilon_i^2)^{3/2}}\right)$$

$$\geq \mathbb{E}\left(\frac{\tau_{v_0}^3}{(\tau_{v_0}^2 + \varepsilon_i^2)^{3/2}}\right) - \sqrt{\frac{\log(1/\delta)}{2n}}.$$

This finishes the proof.

$\qquad\square$

# G   Proofs for the self-tuned case

This section collects proofs for Theorem 3.4.

## G.1   Proof of Theorem of 3.4

*Proof of Theorem of 3.4.* Recall that $\tau = v\sqrt{n}/z$. For simplicity, let $\widehat{\tau} = \widehat{v}\sqrt{n}/z$. Define the profile loss $L_n^{\mathrm{pro}}(v)$ as

$$L_n^{\mathrm{pro}}(v) := L_n(\widehat{\mu}(v), v) = \min_\mu L_n(\mu, v).$$

Its first order gradient is

$$\nabla L_n^{\mathrm{pro}}(v) = \nabla L_n(\widehat{\mu}(v), v) = \frac{\partial}{\partial v}\widehat{\mu}(v) \cdot \frac{\partial}{\partial v}L_n(\mu, v)\Big|_{\mu=\widehat{\mu}(v)} + \frac{\partial}{\partial v}L_n(\mu, v)\Big|_{\mu=\widehat{\mu}(v)} = \frac{\partial}{\partial v}L_n(\widehat{\mu}(v), v), \quad (\text{G.1})$$

where we use the fact that $\partial/\partial\mu\, L_n(\mu, v)|_{\mu=\widehat{\mu}(v)} = 0$, implied by the stationarity of $\widehat{\mu}(v)$.

**Assuming that the constraint is inactive.**   We first assume that the constraint is not active for any stationary point $\widehat{v}$, that is, any stationary point $\widehat{v}$ is an interior point of $[v_0, V_0]$, aka $\widehat{v} \in (v_0, V_0)$. By the joint convexity of $L_n(\mu, v)$ and the convexity of $L_n^{\mathrm{pro}}(v)$, $(\widehat{\mu}(\widehat{v}), \widehat{v})$ and $\widehat{v}$ are stationary points of $L_n(\mu, v)$ and $L_n(\widehat{\mu}(v), v)$, respectively. Thus we have

$$\frac{\partial}{\partial\mu}L_n(\mu, v)\Big|_{(\mu,v)=(\widehat{\mu}(\widehat{v}),\widehat{v})} = -\frac{\sqrt{n}}{z} \cdot \frac{1}{n}\sum_{i=1}^n \frac{y_i - \widehat{\mu}(\widehat{v})}{\sqrt{\widehat{\tau}^2 + (y_i - \widehat{\mu}(\widehat{v}))^2}} = 0,$$

$$\frac{\partial}{\partial v}L_n(\mu, v)\Big|_{(\mu,v)=(\widehat{\mu}(\widehat{v}),\widehat{v})} = \frac{n}{z^2} \cdot \frac{1}{n}\sum_{i=1}^n \frac{\widehat{\tau}}{\sqrt{\widehat{\tau}^2 + (y_i - \widehat{\mu}(\widehat{v}))^2}} - \left(\frac{n}{z^2} - a\right) = 0,$$

$$\nabla L_n^{\mathrm{pro}}(v)\Big|_{v=\widehat{v}} = \nabla L_n(\widehat{\mu}(\widehat{v}), \widehat{v})\Big|_{v=\widehat{v}} = \frac{\partial}{\partial v}L_n(\widehat{\mu}(v), v)\Big|_{v=\widehat{v}} = \frac{\partial}{\partial v}L_n(\mu, v)\Big|_{(\mu,v)=(\widehat{\mu}(\widehat{v}),\widehat{v})} = 0,$$

where the first two equalities are on partial derivatives of $L_n(\mu, v)$ and the last one is on the derivative of the profile loss $L_n(\widehat{\mu}(v), v)$.

Recall that $\tau = \sqrt{n}v/z$. Let $f(\tau) = z^2\nabla L_n^{\mathrm{pro}}(v)/n$, that is

$$f(\tau) = \frac{1}{n}\sum_{i=1}^n \frac{\tau}{\sqrt{\tau^2 + (y_i - \widehat{\mu}(v))^2}} - \left(1 - \frac{az^2}{n}\right).$$

In other words, $\widehat{\tau} = \sqrt{n}\widehat{v}/z$ satisfies $f(\widehat{\tau}) = 0$. We now split the proof for the inactive constraint case into two steps.

**Step 1: Proving $\widehat{v} \le C_0\sigma$ for some universal constant $C_0$.**   We will employ the proof by contradiction argument. Assume there exists some $v$ such that $v > (1 + \epsilon)\sqrt{r^2 + \sigma^2}$ and $\nabla L_v^{\mathrm{pro}}(v) = 0$; or equivalently, there exists some $\tau$ such that

$$\tau > (1 + \epsilon)\sqrt{r^2 + \sigma^2}\sqrt{n}/z =: \bar{\tau} \quad \text{and} \quad f(\tau) = 0, \quad (\text{G.2})$$

where $\epsilon$ and $r$ are to be determined later. Let $\tau_{v_0} = v_0\sqrt{n}/z$. Then, provided that $n$ is large enough, Lemma 3.2 implies that Assumption 1 with $r$ and $\kappa_\ell = 1/(2v)$ holds uniformly over $v \ge v_0$ conditional on the following event

$$\mathcal{E}_1 := \left\{\frac{1}{n}\sum_{i=1}^n \frac{(\tau_{v_0}^2 + 2r^2)^{3/2}}{(\tau_{v_0}^2 + 2r^2 + 2\varepsilon_i^2)^{3/2}} - \frac{1}{n}\sum_{i=1}^n \mathbb{E}\frac{(\tau_{v_0}^2 + 2r^2)^{3/2}}{(\tau_{v_0}^2 + 2r^2 + 2\varepsilon_i^2)^{3/2}} \ge -\sqrt{\frac{\log(1/\delta)}{2n}}\right\}.$$

Conditional on the intersection of event $\mathcal{E}_1$ and the following event

$$\mathcal{E}_2 := \left\{ \sup_{v \in [v_0, V_0]} \left| \frac{1}{n} \sum_{i=1}^n \frac{\varepsilon_i}{\sqrt{\tau^2 + \varepsilon_i^2}} \right| \leq C \cdot \frac{V_0}{v_0} \cdot \frac{\log(n/\delta)}{n} \right\},$$

where $z \lesssim \sqrt{\log(n/\delta)}$ and $C$ is some constant, and following the proof of Theorem 3.1, for any fixed $v$ and thus $\tau$, we have

$$\kappa_\ell |\widehat{\mu}(v) - \mu^*| \leq \left| \frac{1}{\sqrt{n}} \sum_{i=1}^n \frac{\varepsilon_i}{z\sqrt{\tau^2 + \varepsilon_i^2}} \right|.$$

Thus, for any $v$ such that $v_0 \vee \bar{v}_0 := v_0 \vee (1+\epsilon)\sqrt{r^2 + \sigma^2} < v < V_0$, we have on $\mathcal{E}_2$ that

$$\sup_{v_0 \vee \bar{v}_0 < v < V_0} \kappa_\ell(v) |\widehat{\mu}(v) - \mu^*| \leq \sup_{v \in [v_0, V_0]} \kappa_\ell(v) |\widehat{\mu}(v) - \mu^*|$$

$$\leq \sup_{v \in [v_0, V_0]} \left| \frac{1}{\sqrt{n}} \sum_{i=1}^n \frac{\varepsilon_i}{z\sqrt{\tau^2 + \varepsilon_i^2}} \right|$$

$$\leq C \cdot \frac{V_0}{v_0} \cdot \frac{\log(n/\delta)}{z\sqrt{n}},$$

which, by Lemma 3.2, yields

$$\sup_{v \in [v_0, V_0]} |\widehat{\mu}(v) - \mu^*| \leq 2C \cdot \frac{V_0^2}{v_0} \cdot \frac{\log(n/\delta)}{z\sqrt{n}} =: r.$$

The above $r$ can be further refined by using the finer lower bound $\bar{v}_0$ of $v$ instead of $v_0$, but we use $v_0$ for simplicity. Let $\Delta = \mu^* - \widehat{\mu}(v)$, and we have $|\Delta| \leq r$. Let the event $\mathcal{E}_3$ be

$$\mathcal{E}_3 := \left\{ \frac{1}{n} \sum_{i=1}^n \frac{\sqrt{\bar{\tau}^2 + 2(r^2 + \varepsilon_i^2)} - \bar{\tau}}{\sqrt{\bar{\tau}^2 + 2(r^2 + \varepsilon_i^2)}} - \mathbb{E}\left( \frac{\sqrt{\bar{\tau}^2 + 2(r^2 + \varepsilon_i^2)} - \bar{\tau}}{\sqrt{\bar{\tau}^2 + 2(r^2 + \varepsilon_i^2)}} \right) \leq \sqrt{\frac{\log(1/\delta)2(r^2 + \sigma^2)}{n\bar{\tau}^2}} + \frac{\log(1/\delta)}{3n} \right\}.$$

Thus on the event $\mathcal{E}_1 \cap \mathcal{E}_2 \cap \mathcal{E}_3$ and using the fact that $1 - 1/\sqrt{1+x}$ is an increasing function, we have

$$f(\tau) = \frac{az^2}{n} - \frac{1}{n} \sum_{i=1}^n \frac{\sqrt{\tau^2 + (\Delta + \varepsilon_i)^2} - \tau}{\sqrt{\tau^2 + (\Delta + \varepsilon_i)^2}} \geq \frac{az^2}{n} - \frac{1}{n} \sum_{i=1}^n \frac{\sqrt{\tau^2 + 2(r^2 + \varepsilon_i^2)} - \tau}{\sqrt{\tau^2 + 2(r^2 + \varepsilon_i^2)}}$$

$$> \frac{az^2}{n} - \frac{1}{n} \sum_{i=1}^n \frac{\sqrt{\bar{\tau}^2 + 2(r^2 + \varepsilon_i^2)} - \bar{\tau}}{\sqrt{\bar{\tau}^2 + 2(r^2 + \varepsilon_i^2)}} \qquad (\tau < \bar{\tau})$$

$$\geq \frac{az^2}{n} - \left\{ \mathbb{E}\left( \frac{\sqrt{\bar{\tau}^2 + 2(r^2 + \varepsilon_i^2)} - \bar{\tau}}{\sqrt{\bar{\tau}^2 + 2(r^2 + \varepsilon_i^2)}} \right) + \frac{1}{n} \sum_{i=1}^n \frac{\sqrt{\bar{\tau}^2 + 2(r^2 + \varepsilon_i^2)} - \bar{\tau}}{\sqrt{\bar{\tau}^2 + 2(r^2 + \varepsilon_i^2)}} - \mathbb{E}\left( \frac{\sqrt{\bar{\tau}^2 + 2(r^2 + \varepsilon_i^2)} - \bar{\tau}}{\sqrt{\bar{\tau}^2 + 2(r^2 + \varepsilon_i^2)}} \right) \right\}$$

$$\geq \frac{az^2}{n} - \left( \frac{r^2 + \sigma^2}{\bar{\tau}^2} + \sqrt{\frac{\log(1/\delta) \cdot 2(r^2 + \sigma^2)}{n\bar{\tau}^2}} + \frac{\log(1/\delta)}{3n} \right)$$

$$= \frac{z^2}{n} \left( a - \frac{\log(1/\delta)}{3z^2} \right) - \left( \frac{r^2 + \sigma^2}{r^2 + \sigma^2} \frac{z^2}{(1+\epsilon)^2 n} + \sqrt{\frac{r^2 + \sigma^2}{r^2 + \sigma^2} \frac{2z^2 \log(1/\delta)}{(1+\epsilon)^2 n^2}} \right) \qquad \text{(Definition of } \bar{\tau})$$

$$\geq \frac{(a - 1/3)z^2}{n} - \left( \frac{r^2 + \sigma^2}{r^2 + \sigma^2} \frac{z^2}{(1+\epsilon)^2 n} + \sqrt{\frac{r^2 + \sigma^2}{r^2 + \sigma^2} \frac{2z^4}{(1+\epsilon)^2 n^2}} \right) \qquad (z^2 \geq \log(1/\delta))$$

$$\geq \frac{(a - 1/3)z^2}{n} - \frac{z^2}{n} \cdot \left( \frac{1}{(1+\epsilon)^2} + \sqrt{\frac{2}{(1+\epsilon)^2}} \right)$$

$$= \frac{z^2}{n} \left( a - \frac{1}{3} - \frac{1}{(1+\epsilon)^2} - \sqrt{\frac{2}{(1+\epsilon)^2}} \right)$$

$$\geq 0,$$

provided that

$$\frac{1}{1+\epsilon} \leq \frac{\sqrt{1+2(a-1/3)}-1}{\sqrt{2}},$$

or equivalently

$$\epsilon \geq \frac{\sqrt{4a+2/3}+2/3-\sqrt{2}-2a}{2(a-1/3)} =: \epsilon(a).$$

In other words, conditional on the event $\mathcal{E}_1 \cap \mathcal{E}_2 \cap \mathcal{E}_3$ and taking $\epsilon \geq \epsilon(a)$, $f(\tau) > 0$ for $\tau > \bar{\tau} := (1 + \epsilon)\sqrt{r^2 + \sigma^2}\sqrt{n}/z$. This contradicts with (G.2), and thus

$$\widehat{\tau} \leq (1+\epsilon)\sqrt{r^2+\sigma^2}\sqrt{n}/z.$$

If $a = 1/2$ and conditional on the same event, the above holds with

$$\epsilon = \frac{\sqrt{4a+2/3}+2/3-\sqrt{2}-2a}{2(a-1/3)} \geq 9.$$

If $n$ is large enough such that $12\sigma \geq 5\sqrt{r^2+\sigma^2}$, then conditional on the event $\mathcal{E}_1 \cap \mathcal{E}_2 \cap \mathcal{E}_3$, we have

$$v_0 \leq \widehat{v} \leq C_0\sigma,$$

where $C_0 = 12$.

**Step 2: Proving $\widehat{v} \geq c_0\sigma_{\tau_{v_0}^2-1}$ for some constant $c_0$.** We will again employ the proof by contradiction argument. Let

$$g(\tau) := \left(\frac{1}{n}\sum_{i=1}^{n}\frac{\tau^2}{\sqrt{\tau^2+(\Delta+\varepsilon_i)^2}}\right)^2 - \left(1 - \frac{az^2}{n}\right)^2.$$

Assume there exists some $v$ such that $v < c$ and $\frac{\partial}{\partial v}L_n(\widehat{\mu}(v), v) = 0$. Or equivalently, assume there exists some $\tau$ such that

$$\tau < c\sqrt{n}/z =: \underline{\tau} \quad \text{and} \quad g(\tau) = 0. \tag{G.3}$$

It is impossible that $c \leq v_0$ because any stationary point $v$ is in $(v_0, V_0)$. Thus $c > v_0$. Let $\Delta = \widehat{\mu}(v) - \mu^*$. Thus on the event $\mathcal{E}_1 \cap \mathcal{E}_2$, using the facts that $\sqrt{x}$ is a concave function and $1/\sqrt{1+y/x}$ is an increasing function of $x$, we have

$$
\begin{aligned}
\frac{1}{n}\sum_{i=1}^{n}\frac{\tau^2}{\sqrt{\tau^2+(\Delta+\varepsilon_i)^2}} &= \frac{1}{n}\sum_{i=1}^{n}\frac{1}{\sqrt{1+(\Delta+\varepsilon_i)^2/\tau^2}} \\
&\leq \frac{1}{n}\sum_{i=1}^{n}\frac{1}{\sqrt{1+(\Delta+\varepsilon_i)^2/\underline{\tau}^2}} \\
&\leq \sqrt{\frac{1}{n}\sum_{i=1}^{n}\frac{1}{1+(\Delta+\varepsilon_i)^2/\underline{\tau}^2}} \\
&\leq \sqrt{\frac{1}{n}\sum_{i=1}^{n}\frac{1}{1+\underline{\tau}^{-2}(\Delta+\varepsilon_i)^2\cdot\mathbb{1}\left((\Delta+\varepsilon_i)^2\leq\underline{\tau}^2\right)}} \\
&\leq \sqrt{1-\frac{1}{n}\cdot\frac{1}{2\underline{\tau}^2}\sum_{i=1}^{n}(\Delta+\varepsilon_i)^2\cdot\mathbb{1}\left((\Delta+\varepsilon_i)^2\leq\underline{\tau}^2\right)}.
\end{aligned}
$$

By the proof from step 1, on the event $\mathcal{E}_1 \cap \mathcal{E}_2$, we have

$$\sup_{v \in [v_0, V_0]} |\widehat{\mu}(v) - \mu^*| \leq r,$$

where $r$ is the same as in step 1. Then

$$
\begin{aligned}
g(\tau) &\leq 1 - \frac{1}{n} \cdot \frac{1}{2\underline{\tau}^2} \sum_{i=1}^{n} (\Delta + \varepsilon_i)^2 \cdot 1\left((\Delta + \varepsilon_i)^2 \leq \underline{\tau}^2\right) - \left(1 - \frac{az^2}{n}\right)^2 \\
&< \frac{2az^2}{n} - \frac{1}{n} \cdot \frac{1}{2\underline{\tau}^2} \sum_{i=1}^{n} (\Delta + \varepsilon_i)^2 \cdot 1\left((\Delta + \varepsilon_i)^2 \leq \underline{\tau}^2\right) \qquad \text{(as long as } az^2/n > 0\text{)} \\
&\leq \frac{2az^2}{n} - \frac{1}{n} \cdot \frac{1}{2\underline{\tau}^2} \sum_{i=1}^{n} \left(\varepsilon_i^2 + 2\Delta\varepsilon_i\right) \cdot 1\left(\varepsilon_i^2 \leq 2^{-1}\underline{\tau}^2 - r^2\right) \\
&\leq \frac{2az^2}{n} - \frac{1}{2\underline{\tau}^2}\left(\frac{1}{n}\sum_{i=1}^{n}\varepsilon_i^2 1\left(\varepsilon_i^2 \leq \frac{\underline{\tau}^2}{2} - r^2\right) - \frac{2}{n}\sum_{i=1}^{n} r|\varepsilon_i| 1\left(\varepsilon_i^2 \leq \frac{\underline{\tau}^2}{2} - r^2\right)\right) \\
&= \frac{2az^2}{n} - \frac{1}{2\underline{\tau}^2}\left(\mathrm{I} - 2r \cdot \mathrm{II}\right).
\end{aligned}
$$

Define the probability event $\mathcal{E}_4$ as

$$\mathcal{E}_4 := \mathcal{E}_{41} \cap \mathcal{E}_{42}$$

where

$$
\mathcal{E}_{41} =: \left\{\frac{1}{n}\sum_{i=1}^{n}\varepsilon_i^2 1\left(\varepsilon_i^2 \leq \frac{\underline{\tau}^2}{2} - r^2\right) \geq \mathbb{E}\varepsilon_i^2 1\left(\varepsilon_i^2 \leq \frac{\underline{\tau}^2}{2} - r^2\right) - \sigma_{\frac{\underline{\tau}^2}{2}}\sqrt{\frac{\underline{\tau}^2 \log(1/\delta)}{n}} - \frac{\underline{\tau}^2 \log(1/\delta)}{6n}\right\},
$$

$$
\mathcal{E}_{42} =: \left\{\frac{1}{n}\sum_{i=1}^{n}|\varepsilon_i| 1\left(\varepsilon_i^2 \leq \frac{\underline{\tau}^2}{2} - r^2\right) \leq \mathbb{E}|\varepsilon_i| 1\left(\varepsilon_i^2 \leq \frac{\underline{\tau}^2}{2} - r^2\right) + \sqrt{\frac{2\sigma_{\underline{\tau}^2/2}^2 \log(1/\delta)}{n}} + \frac{\underline{\tau} \log(1/\delta)}{3\sqrt{2}n}\right\}.
$$

If $n$ is sufficiently large such that

$$
r^2 \leq \epsilon_0 \lesssim \left(\frac{\log n + \log(1/\delta)}{z\sqrt{n}}\right)^2 \leq 1,
$$

$$
\frac{r}{\underline{\tau}^2}\left(\sigma_{\underline{\tau}^2/2}^2 + \sqrt{\frac{2\sigma_{\underline{\tau}^2/2}^2 \log(1/\delta)}{n}} + \frac{\underline{\tau} \log(1/\delta)}{3\sqrt{2}n}\right) \leq \frac{1}{12}\frac{\log(1/\delta)}{n},
$$

then conditional on $\mathcal{E}_4$, we have

$$
\mathrm{I} \geq \mathbb{E}\varepsilon_i^2 1\left(\varepsilon_i^2 \leq \frac{\underline{\tau}^2}{2} - r^2\right) - \sigma_{\frac{\underline{\tau}^2}{2}}\sqrt{\frac{\underline{\tau}^2 \log(1/\delta)}{n}} - \frac{\underline{\tau}^2 \log(1/\delta)}{6n},
$$

$$
\mathrm{II} \leq \mathbb{E}|\varepsilon_i| 1\left(\varepsilon_i^2 \leq \frac{\underline{\tau}^2}{2} - r^2\right) + \sqrt{\frac{2\sigma_{\underline{\tau}^2/2}^2 \log(1/\delta)}{n}} + \frac{\underline{\tau} \log(1/\delta)}{3\sqrt{2}n}.
$$

Thus conditional on $\mathcal{E}_4$, we have

$$g(\tau) < \frac{2az^2}{n} - \frac{1}{2\underline{\tau}^2}\left(\text{I} - 2r \cdot \text{II}\right)$$

$$\leq \frac{2az^2}{n} - \frac{1}{2\underline{\tau}^2}\left(\mathbb{E}\varepsilon_i^2 \mathbf{1}\left(\varepsilon_i^2 \leq \frac{\underline{\tau}^2}{2} - r^2\right) - \sigma_{\underline{\tau}^2/2}\sqrt{\frac{\underline{\tau}^2\log(1/\delta)}{n}} - \frac{\underline{\tau}^2\log(1/\delta)}{6n}\right)$$

$$+ \frac{r}{\underline{\tau}^2}\left(\mathbb{E}|\varepsilon_i|\mathbf{1}\left(\varepsilon_i^2 \leq \frac{\underline{\tau}^2}{2} - r^2\right) + \sqrt{\frac{2\sigma_{\underline{\tau}^2/2}^2\log(1/\delta)}{n}} + \frac{\underline{\tau}\log(1/\delta)}{3\sqrt{2}n}\right)$$

$$\leq \frac{2az^2}{n} - \frac{\sigma_{\underline{\tau}^2/2-\epsilon_0}^2}{2\underline{\tau}^2} + \frac{\sigma_{\underline{\tau}^2/2}\sqrt{\log(1/\delta)}}{2\underline{\tau}\sqrt{n}} + \frac{\log(1/\delta)}{12n} + \frac{r}{\underline{\tau}^2}\left(\sigma_{\underline{\tau}^2/2}^2 + \sqrt{\frac{2\sigma_{\underline{\tau}^2/2}^2\log(1/\delta)}{n}} + \frac{\underline{\tau}\log(1/\delta)}{3\sqrt{2}n}\right)$$

$$\leq \frac{z^2}{n}\left(2a + \frac{\log(1/\delta)}{z^2}\cdot\frac{1}{6}\right) - \frac{\sigma_{\underline{\tau}^2/2-\epsilon_0}^2}{2\underline{\tau}^2} + \frac{\sigma_{\underline{\tau}^2/2}\sqrt{\log(1/\delta)}}{2\underline{\tau}\sqrt{n}}$$

$$= \frac{z^2}{2n}\left(4a + \frac{\log(1/\delta)}{z^2}\cdot\frac{1}{3} - \frac{\sigma_{\underline{\tau}^2/2-\epsilon_0}^2}{c^2} + \frac{\sigma_{\underline{\tau}^2/2}}{c}\cdot\frac{\sqrt{\log(1/\delta)}}{z}\right) \qquad (\underline{\tau} = c\sqrt{n}/z)$$

$$\leq \frac{z^2}{2n}\left(4a + \frac{1}{3} - \frac{\sigma_{\underline{\tau}^2/2-\epsilon_0}^2}{c^2} + \frac{\sigma_{\underline{\tau}^2/2}}{c}\right) \qquad (z^2 \geq \log(1/\delta))$$

$$\leq 0,$$

for any $c$ such that

$$c \leq \frac{\sigma_{\underline{\tau}^2/2}}{2(4a+1/3)}\left(\sqrt{1 + \frac{4(4a+1/3)\sigma_{\underline{\tau}^2/2-\epsilon_0}^2}{\sigma_{\underline{\tau}^2/2}^2}} - 1\right),$$

In other words, conditional no the event $\mathcal{E}_1 \cap \mathcal{E}_2 \cap \mathcal{E}_4$ and taking any $c$ such that it satisfies the above inequality, we have

$$g(\tau) < 0 \text{ for any } \tau < \underline{\tau} = c\sqrt{n}/z.$$

This is a contradiction. Thus, $\widehat{\tau} \geq \underline{\tau} = c\sqrt{n}/z$, or equivalently $\widehat{v} \geq c > v_0$. Using the inequality

$$\sqrt{1+x} - 1 \geq \mathbf{1}(x \geq 3) + \frac{x}{3}\mathbf{1}(0 \leq x < 3) \geq \frac{x}{3} \wedge 1 \quad \forall\, x \geq 0,$$

we have

$$\frac{\sigma_{\underline{\tau}^2/2}}{2(4a+1/3)}\left(\sqrt{1 + \frac{4(4a+1/3)\sigma_{\underline{\tau}^2/2-\epsilon_0}^2}{\sigma_{\underline{\tau}^2/2}^2}} - 1\right)$$

$$= \frac{3\sigma_{\underline{\tau}_{v_0}^2/2}}{14}\left(\sqrt{1 + \frac{28\sigma_{\underline{\tau}^2/2-\epsilon_0}^2}{3\sigma_{\underline{\tau}^2/2}^2}} - 1\right) \qquad (a = 1/2)$$

$$\geq \frac{3\sigma_{\underline{\tau}^2/2}}{14}\left(\frac{28\sigma_{\underline{\tau}^2/2-\epsilon_0}^2}{9\sigma_{\underline{\tau}^2/2}^2} \wedge 1\right)$$

$$= \frac{2\sigma_{\underline{\tau}^2/2-\epsilon_0}^2}{3\sigma_{\underline{\tau}^2/2}} \wedge \frac{3\sigma_{\underline{\tau}^2/2}}{14}$$

$$\geq \frac{1}{5}\left(\frac{\sigma_{\underline{\tau}^2/2-1}}{\sigma_{\underline{\tau}^2/2}} \wedge 1\right)\sigma_{\underline{\tau}^2/2-1}$$

$$\geq \frac{1}{5}\left(\frac{\sigma_{\underline{\tau}_{v_0}^2/2-1}}{\sigma_{\underline{\tau}_{v_0}^2/2}} \wedge 1\right)\sigma_{\underline{\tau}_{v_0}^2/2-1}.$$

Therefore we can take $c = 5^{-1}(\sigma_{\tau_{v_0}^2/2-1}/\sigma_{\tau_{v_0}^2/2} \wedge 1)\sigma_{\tau_{v_0}^2/2-1}$. Thus on the event $\mathcal{E}_1 \cap \mathcal{E}_2 \cap \mathcal{E}_4$, we have

$$\widehat{v} \geq c := c_0 \sigma_{\tau_{v_0}^2/2-1},$$

where $c_0 = 5^{-1}(\sigma_{\tau_{v_0}^2/2-1}/\sigma_{\tau_{v_0}^2/2} \wedge 1)$. This finishes the proof of step 2.

**Proving that the constraint is inactive.** If $\widehat{v} \notin (v_0, V_0)$, then $\widehat{v} \in \{v_0, V_0\}$. Suppose $\widehat{v} = v_0$, then $\widehat{v} = v_0 < c$. Recall that $\tau_{v_0} = v_0\sqrt{n}/z$. Then we must have $f(\tau_{v_0}) \geq 0$, and thus $g(\tau_{v_0}) \geq 0$. However, conditional on the probability event $\mathcal{E}_1 \cap \mathcal{E}_2 \cap \mathcal{E}_4$, repeating the above analysis in step 2 would obtain $g(\tau_{v_0}) < 0$. This is a contradiction. Therefore $\widehat{v} \neq v_0$. Similarly, conditional on probability event $\mathcal{E}_1 \cap \mathcal{E}_2 \cap \mathcal{E}_3$, we can obtain $\widehat{v} \neq V_0$. Therefore, conditional on the probability event $\mathcal{E}_1 \cap \mathcal{E}_2 \cap \mathcal{E}_3 \cap \mathcal{E}_4$, the constraint must be inactive, aka $\widehat{v} \in (v_0, V_0)$.

Using the first result of Lemma F.2 with $\tau^2$ and $\varepsilon_i^2$ replaced by $\tau_{v_0}^2 + 2r^2$ and $2\varepsilon_i^2$ respectively, Lemma G.1, Lemma G.2 with $\tau^2$ and $w_i^2$ replaced by $\bar{\tau}^2$ and $2(r^2 + \varepsilon_i^2)$ respectively, and Lemma G.3, we obtain

$$\mathbb{P}(\mathcal{E}_1) \geq 1 - \delta, \ \mathbb{P}(\mathcal{E}_2) \geq 1 - \delta, \ \mathbb{P}(\mathcal{E}_3) \geq 1 - \delta, \ \mathbb{P}(\mathcal{E}_4) \geq 1 - 2\delta,$$

and thus

$$\mathbb{P}(\mathcal{E}_1 \cap \mathcal{E}_2 \cap \mathcal{E}_3 \cap \mathcal{E}_4) \geq 1 - 5\delta.$$

Putting the above results together, and using Lemmas G.1 and G.3, we obtain with probability at least $1 - 5\delta$ that

$$c_0 \sigma_{\tau_{v_0}^2/2-1} \leq \widehat{v} \leq C_0 \sigma.$$

Using a change of variable $5\delta \to \delta$ completes the proof. $\qquad\square$

## G.2 Proof of Theorem 3.5

*Proof of Theorem 3.5.* On the probability event $\mathcal{E}_1 \cap \mathcal{E}_2 \cap \mathcal{E}_3 \cap \mathcal{E}_4$ where $\mathcal{E}_k$'s are defined the same as in the proof of Theorem 3.4, we have

$$c_0 \sigma_{\tau_{v_0}^2/2-1} \leq \widehat{v} \leq C_0 \sigma.$$

Following the proof of Theorem 3.1, for any fixed $v$ and thus $\tau$, we have

$$\kappa_\ell |\widehat{\mu}(v) - \mu^*| \leq \left| \frac{1}{\sqrt{n}} \sum_{i=1}^n \frac{\varepsilon_i}{z\sqrt{\tau^2 + \varepsilon_i^2}} \right|.$$

For any $v$ such that $c_0 \sigma_{\tau_{v_0}^2/2-1} \leq v \leq C_0 \sigma$ and any $z > 0$, using Lemma G.1 but with $v_0$ and $V_0$ replaced by $c_0 \sigma_{\tau_{v_0}^2/2-1}$ and $C_0 \sigma$ respectively, we obtain with probability at least $1 - \delta$

$$\sup_{v \in [c_0 \sigma_{\tau_{v_0}^2/2-1},\ C_0\sigma]} \kappa_\ell(v) |\widehat{\mu}(v) - \mu^*| \leq \sup_{v \in [c_0 \sigma_{\tau_{v_0}^2/2-1},\ C_0\sigma]} \kappa_\ell(v) |\widehat{\mu}(v) - \mu^*|$$

$$\leq \sup_{v \in [c_0 \sigma_{\tau_{v_0}^2/2-1},\ C_0\sigma]} \left| \frac{1}{\sqrt{n}} \sum_{i=1}^n \frac{\varepsilon_i}{z\sqrt{\tau^2 + \varepsilon_i^2}} \right|$$

$$\leq \frac{\sigma}{c_0 \sigma_{\tau_{v_0}^2/2-1}} \sqrt{\frac{2\log(n/\delta)}{n}} + \frac{1}{z}\frac{\log(n/\delta)}{\sqrt{n}}$$

$$+ \frac{\sigma^2}{2c_0^2 \sigma_{\tau_{v_0}^2/2-1}^2} \frac{z}{\sqrt{n}} + \frac{3(C_0\sigma - c_0\sigma_{\tau_{v_0}^2/2-1})}{\sigma_{\tau_{v_0}^2/2-1}} \frac{1}{z\sqrt{n}},$$

which yields

$$\sup_{v \in [c_0 \sigma_{\tau_{v_0}^2/2-1},\ C_0 \sigma]} |\widehat{\mu}(v) - \mu^*| \le C\sigma \frac{\log(n/\delta) \vee z^2 \vee 1}{z\sqrt{n}},$$

where $C$ is some constant only depending on $\sigma/\sigma_{\tau_{v_0}^2/2-1}$, $c_0$, and $C_0$. Putting the above pieces together and if $\log(1/\delta) \le z^2 \le \log(n/\delta)$, we obtain with probability at least $1 - 6\delta$ that

$$|\widehat{\mu}(\widehat{v}) - \mu^*| \le \sup_{v \in [c_0 \sigma_{\tau_{v_0}^2/2-1},\ C_0 \sigma]} |\widehat{\mu}(v) - \mu^*| \le C \cdot \sigma \frac{\log(n/\delta) \vee 1}{z\sqrt{n}}.$$

Using a change of variable $6\delta \to \delta$ and then setting $z = \log(n/\delta)$ gives

$$|\widehat{\mu}(\widehat{v}) - \mu^*| \le \sup_{v \in [c_0 \sigma_{\tau_{v_0}^2/2-1},\ C_0 \sigma]} |\widehat{\mu}(v) - \mu^*| \le C \cdot \sigma \sqrt{\frac{\log(n/\delta)}{n}}$$

with a lightly different constant $C$, provided that $\log(n/\delta) \ge 1$, aka $n \ge e\delta$. This completes the proof. $\square$

### G.3 Supporting lemmas

We collect supporting lemmas, aka Lemmas G.1, G.2, and G.3, in this subsection.

**Lemma G.1.** Let $0 < \delta < 1$. Suppose $\sigma \lesssim V_0$ and $z \lesssim \sqrt{\log(n/\delta)}$. Then, with probability at least $1 - \delta$, we have

$$\sup_{v \in [v_0, V_0]} \left| \frac{1}{n} \sum_{i=1}^n \frac{\varepsilon_i}{\sqrt{\tau^2 + \varepsilon_i^2}} \right| \le C \cdot \frac{V_0}{v_0} \cdot \frac{\log(n/\delta)}{n}$$

where $C$ is some constant.

*Proof of Lemma G.1.* To prove the uniform bound over $[v_0, V_0]$, we adopt a covering argument. For any $0 < \epsilon \le 1$, there exists an $\epsilon$-cover $\mathcal{N}$ of $[v_0, V_0]$ such that $|\mathcal{N}| \le 3(V_0 - v_0)/\epsilon$. Let $\tau_w = w\sqrt{n}/z$. Then for every $v \in [v_0, V_0]$, there exists a $w \in \mathcal{N} \subset [v_0, V_0]$ such that $|w - \tau| \le \epsilon$ and

$$\left| \frac{1}{\sqrt{n}} \sum_{i=1}^n \frac{\varepsilon_i}{z\sqrt{\tau^2 + \varepsilon_i^2}} \right| \le \left| \frac{1}{\sqrt{n}} \sum_{i=1}^n \frac{\varepsilon_i}{z\sqrt{\tau_w^2 + \varepsilon_i^2}} \right|$$

$$+ \left| \frac{1}{\sqrt{n}} \sum_{i=1}^n \frac{\varepsilon_i}{z\sqrt{\tau_w^2 + \varepsilon_i^2}} - \frac{1}{\sqrt{n}} \sum_{i=1}^n \frac{\varepsilon_i}{z\sqrt{\tau^2 + \varepsilon_i^2}} \right|$$

$$\le \left| \frac{1}{\sqrt{n}} \sum_{i=1}^n \frac{\varepsilon_i}{z\sqrt{\tau_w^2 + \varepsilon_i^2}} - \mathbb{E}\left[ \frac{1}{\sqrt{n}} \sum_{i=1}^n \frac{\varepsilon_i}{z\sqrt{\tau_w^2 + \varepsilon_i^2}} \right] \right|$$

$$+ \left| \mathbb{E}\left[ \frac{1}{\sqrt{n}} \sum_{i=1}^n \frac{\varepsilon_i}{z\sqrt{\tau_w^2 + \varepsilon_i^2}} \right] \right|$$

$$+ \left| \frac{1}{\sqrt{n}} \sum_{i=1}^n \frac{\varepsilon_i}{z\sqrt{\tau_w^2 + \varepsilon_i^2}} - \frac{1}{\sqrt{n}} \sum_{i=1}^n \frac{\varepsilon_i}{z\sqrt{\tau^2 + \varepsilon_i^2}} \right|$$

$$= \mathrm{I} + \mathrm{II} + \mathrm{III}.$$

For II, we have

$$\mathrm{II} \le \frac{\sqrt{n}}{z} \cdot \frac{\sigma^2}{2\tau_w^2} \le \frac{z\sigma^2}{2v_0^2\sqrt{n}}.$$

For III, using the inequality

$$\left| \frac{x}{\sqrt{\tau_w^2 + x^2}} - \frac{x}{\sqrt{\tau^2 + x^2}} \right| \le \frac{|\tau_w - \tau|}{2\,|\tau_w| \wedge |\tau|},$$

we obtain

$$\text{III} \le \frac{\sqrt{n}}{z} \cdot \frac{\epsilon}{2(w \wedge v)} \le \frac{\sqrt{n}}{z} \cdot \frac{\epsilon}{2v_0}.$$

We then bound I. For any fixed $\tau_w$, applying Lemma F.1 with the fact that $\left| \mathbb{E}\left( \tau_w \varepsilon_i / (\tau_w^2 + \varepsilon_i^2)^{1/2} \right) \right| \le \sigma^2/(2\tau_w)$, we obtain with probability at least $1 - 2\delta$

$$\left| \frac{1}{\sqrt{n}} \sum_{i=1}^n \frac{\varepsilon_i}{z\sqrt{\tau_w^2 + \varepsilon_i^2}} - \mathbb{E}\left[ \frac{1}{\sqrt{n}} \sum_{i=1}^n \frac{\varepsilon_i}{z\sqrt{\tau_w^2 + \varepsilon_i^2}} \right] \right| \le \frac{\sqrt{n}}{z\tau_w} \left( \sigma \sqrt{\frac{2\log(1/\delta)}{n}} + \frac{\tau_w \log(1/\delta)}{n} \right)$$

$$\le \frac{\sigma}{z\tau_{v_0}} \sqrt{2\log(1/\delta)} + \frac{1}{z} \frac{\log(1/\delta)}{\sqrt{n}}$$

where $\tau_{v_0} = v_0 \sqrt{n}/z$. Therefore, putting above pieces together and using the union bound, we obtain with probability at least $1 - 6\epsilon^{-1}(V_0 - v_0)\delta$

$$\sup_{v \in [v_0, V_0]} \left| \frac{1}{\sqrt{n}} \sum_{i=1}^n \frac{\varepsilon_i}{z\sqrt{\tau^2 + \varepsilon_i^2}} \right| \le \sup_{w \in \mathcal{N}} \left| \frac{1}{\sqrt{n}} \sum_{i=1}^n \frac{\varepsilon_i}{z\sqrt{\tau_w^2 + \varepsilon_i^2}} - \mathbb{E}\left[ \frac{1}{\sqrt{n}} \sum_{i=1}^n \frac{\varepsilon_i}{z\sqrt{\tau_w^2 + \varepsilon_i^2}} \right] \right|$$

$$+ \frac{z\sigma^2}{2v_0^2 \sqrt{n}} + \frac{\sqrt{n}}{z} \cdot \frac{\epsilon}{2v_0}$$

$$\le \frac{\sigma}{v_0} \sqrt{\frac{2\log(1/\delta)}{n}} + \frac{1}{z} \frac{\log(1/\delta)}{\sqrt{n}} + \frac{\sigma^2}{2v_0^2} \frac{z}{\sqrt{n}} + \frac{\sqrt{n}}{z} \cdot \frac{\epsilon}{2v_0}.$$

Taking $\epsilon = 6(V_0 - v_0)/n$, we obtain with probability at least $1 - n\delta$

$$\sup_{v \in [v_0, V_0]} \left| \frac{1}{\sqrt{n}} \sum_{i=1}^n \frac{\varepsilon_i}{z\sqrt{\tau^2 + \varepsilon_i^2}} \right| \le \frac{\sigma}{v_0} \sqrt{\frac{2\log(1/\delta)}{n}} + \frac{1}{z} \frac{\log(1/\delta)}{\sqrt{n}} + \frac{\sigma^2}{2v_0^2} \frac{z}{\sqrt{n}} + \frac{3(V_0 - v_0)}{v_0} \frac{1}{z\sqrt{n}}.$$

Thus with probability at least $1 - \delta$, we have

$$\sup_{v \in [v_0, V_0]} \left| \frac{1}{\sqrt{n}} \sum_{i=1}^n \frac{\varepsilon_i}{z\sqrt{\tau^2 + \varepsilon_i^2}} \right| \le \frac{\sigma}{v_0} \sqrt{\frac{2\log(n/\delta)}{n}} + \frac{1}{z} \frac{\log(n/\delta)}{\sqrt{n}} + \frac{\sigma^2}{2v_0^2} \frac{z}{\sqrt{n}} + \frac{3(V_0 - v_0)}{v_0} \frac{1}{z\sqrt{n}}$$

$$\le C \cdot \frac{V_0}{v_0} \cdot \frac{\log(n/\delta)}{z\sqrt{n}}$$

provided $z \lesssim \sqrt{\log(n/\delta)}$, where $C$ is a constant only depending on $\sigma^2/(v_0 V_0)$. When $v_0$ and $V_0$ are taken symmetrically around 1, $v_0 V_0$ is close to 1. Multiplying both sides by $z/\sqrt{n}$ finishes the proof. $\qquad\square$

**Lemma G.2.** Let $w_i$ be i.i.d. copies of $w$. For any $0 < \delta < 1$, with probability at least $1 - \delta$

$$\frac{1}{n} \sum_{i=1}^n \frac{\sqrt{\tau^2 + w_i^2} - \tau}{\sqrt{\tau^2 + w_i^2}} - \mathbb{E}\left( \frac{\sqrt{\tau^2 + w_i^2} - \tau}{\sqrt{\tau^2 + w_i^2}} \right) \le \sqrt{\frac{\log(1/\delta)\,\mathbb{E}w_i^2}{n\tau^2}} + \frac{\log(1/\delta)}{3n}.$$

*Proof of Lemma G.2.* The random variables

$$Z_i = Z_i(\tau) := \frac{\sqrt{\tau^2 + w_i^2} - \tau}{\sqrt{\tau^2 + w_i^2}} = \frac{\sqrt{1 + w_i^2/\tau^2} - 1}{\sqrt{1 + w_i^2/\tau^2}}$$

with $\mu_z = \mathbb{E}Z_i$ and $\sigma_z^2 = \text{var}(Z_i)$ are bounded i.i.d. random variables such that

$$0 \le Z_i \le 1 \wedge \frac{w_i^2}{2\tau^2}.$$

Moreover we have

$$\mathbb{E}Z_i^2 \le \frac{\mathbb{E}w_i^2}{2\tau^2}, \ \sigma_z^2 := \text{var}(Z_i) \le \frac{\mathbb{E}w_i^2}{2\tau^2}.$$

For third and higher order absolute moments, we have

$$\mathbb{E}|Z_i|^k \le \frac{\mathbb{E}w_i^2}{2\tau^2} \le \frac{k!}{2} \cdot \frac{\mathbb{E}w_i^2}{2\tau^2} \cdot \left(\frac{1}{3}\right)^{k-2}, \ \text{for all integers } k \ge 3.$$

Therefore, using Lemma J.2 with $v = n\,\mathbb{E}w_i^2/(2\tau^2)$ and $c = 1/3$ acquires that for any $t > 0$

$$\mathbb{P}\left(\sum_{i=1}^n \frac{(1+w_i^2/\tau^2)^{1/2}-1}{(1+w_i^2/\tau^2)^{1/2}} - \sum_{i=1}^n \mathbb{E}\left(\frac{(1+w_i^2/\tau^2)^{1/2}-1}{(1+w_i^2/\tau^2)^{1/2}}\right) \ge -\sqrt{\frac{tn\,\mathbb{E}w_i^2}{\tau^2}} - \frac{t}{3}\right) \le \exp(-t).$$

Taking $t = \log(1/\delta)$ acquires that for any $0 < \delta < 1$

$$\mathbb{P}\left(\frac{1}{n}\sum_{i=1}^n \frac{(1+w_i^2/\tau^2)^{1/2}-1}{(1+w_i^2/\tau^2)^{1/2}} - \mathbb{E}\left(\frac{(1+w_i^2/\tau^2)^{1/2}-1}{(1+w_i^2/\tau^2)^{1/2}}\right) > -\sqrt{\frac{\log(1/\delta)\,\mathbb{E}w_i^2}{n\tau^2}} - \frac{\log(1/\delta)}{3n}\right) > 1-\delta.$$

This finishes the proof.

$\square$

**Lemma G.3.** For any $0 < \delta < 1$, we have with probability at least $1 - \delta$ that

$$\frac{1}{n}\sum_{i=1}^n \varepsilon_i^2 \mathbb{1}\left(\varepsilon_i^2 \le \frac{\tau^2}{2} - r^2\right) \ge \frac{1}{n}\sum_{i=1}^n \mathbb{E}\varepsilon_i^2 \mathbb{1}\left(\varepsilon_i^2 \le \frac{\tau^2}{2} - r^2\right) - \sigma_{\underline{\tau}^2/2}\sqrt{\frac{\tau^2\log(1/\delta)}{n}} - \frac{\tau^2\log(1/\delta)}{6n}.$$

For any $0 < \delta < 1$, we have with probability at least $1 - \delta$ that

$$\frac{1}{n}\sum_{i=1}^n |\varepsilon_i|\mathbb{1}\left(\varepsilon_i^2 \le \frac{\tau^2}{2} - r^2\right) \le \frac{1}{n}\sum_{i=1}^n \mathbb{E}|\varepsilon_i|\mathbb{1}\left(\varepsilon_i^2 \le \frac{\tau^2}{2} - r^2\right) + \sqrt{\frac{2\sigma_{\underline{\tau}^2/2}^2\log(1/\delta)}{n}} + \frac{\tau\log(1/\delta)}{3\sqrt{2}n}.$$

Consequently, we have, with probability at least $1 - 2\delta$, the above two inequalities hold simultaneously.

*Proof of Lemma G.3.* We prove the first two results one by one and the last result directly follows from first two.

**First result.** Let $Z_i = \varepsilon_i^2 \mathbb{1}\left(\varepsilon_i^2 \le \underline{\tau}^2/2 - r^2\right)$. The random variables $Z_i$ with $\mu_z = \mathbb{E}Z_i$ and $\sigma_z^2 = \text{var}(Z_i)$ are bounded i.i.d. random variables such that

$$|Z_i| = \left|\varepsilon_i^2 \mathbb{1}\left(\varepsilon_i^2 \le \underline{\tau}^2/2 - r^2\right)\right| \le \underline{\tau}^2/2,$$
$$|\mu_z| = |\mathbb{E}Z_i| = \left|\mathbb{E}\left(\varepsilon_i^2 \mathbb{1}\left(\varepsilon_i^2 \le \underline{\tau}^2/2 - r^2\right)\right)\right| \le \sigma_{\underline{\tau}^2/2}^2,$$
$$\mathbb{E}Z_i^2 = \mathbb{E}\left(\varepsilon_i^4 \mathbb{1}\left(\varepsilon_i^2 \le \underline{\tau}^2/2 - r^2\right)\right) \le \underline{\tau}^2\sigma_{\underline{\tau}^2/2}^2/2,$$
$$\sigma_z^2 := \text{var}(Z_i) = \mathbb{E}\left(Z_i - \mu_z\right)^2 \le \underline{\tau}^2\sigma_{\underline{\tau}^2/2}^2/2.$$

For third and higher order absolute moments, we have

$$\mathbb{E}|Z_i|^k = \mathbb{E}\left|\varepsilon_i^2\mathbb{1}\left(\varepsilon_i^2 \le \underline{\tau}^2/2 - r^2\right)\right|^k \le \frac{\underline{\tau}^2\sigma_{\underline{\tau}^2/2}^2}{2}\left(\frac{\underline{\tau}^2}{2}\right)^{k-2} \le \frac{k!}{2}\frac{\underline{\tau}^2\sigma_{\underline{\tau}^2/2}^2}{2}\left(\frac{\underline{\tau}^2}{6}\right)^{k-2}, \ \text{for all integers } k \ge 3.$$

Using Lemma J.2 with $v = n\underline{\tau}^2 \sigma_{\underline{\tau}^2/2}^2/2$ and $c = \underline{\tau}^2/6$, we have for any $t > 0$

$$\mathbb{P}\left(\sum_{i=1}^{n}\varepsilon_i^2 1\left(\varepsilon_i^2 \leq \frac{\underline{\tau}^2}{2} - r^2\right) - \sum_{i=1}^{n}\mathbb{E}\varepsilon_i^2 1\left(\varepsilon_i^2 \leq \frac{\underline{\tau}^2}{2} - r^2\right) \leq -\sqrt{n\underline{\tau}^2\sigma_{\underline{\tau}^2/2}^2 t} - \frac{\underline{\tau}^2 t}{6}\right) \leq \exp\left(-t\right).$$

Taking $t = \log(1/\delta)$ acquires the desired result.

**Second result.** With an abuse of notation, let $Z_i = |\varepsilon_i| 1\left(\varepsilon_i^2 \leq \underline{\tau}^2/2 - r^2\right)$. The random variables $Z_i$ with $\mu_z = \mathbb{E}Z_i$ and $\sigma_z^2 = \mathrm{var}(Z_i)$ are bounded i.i.d. random variables such that

$$|Z_i| = \left|\varepsilon_i 1\left(\varepsilon_i^2 \leq \underline{\tau}^2/2 - r^2\right)\right| \leq \underline{\tau}/\sqrt{2},$$
$$|\mu_z| = |\mathbb{E}Z_i| = \left|\mathbb{E}\left(|\varepsilon_i| 1\left(\varepsilon_i^2 \leq \underline{\tau}^2/2 - r^2\right)\right)\right| \leq \sqrt{2}\sigma_{\underline{\tau}^2/2}^2/\underline{\tau},$$
$$\mathbb{E}Z_i^2 = \mathbb{E}\left(\varepsilon_i^2 1\left(\varepsilon_i^2 \leq \underline{\tau}^2/2 - r^2\right)\right) \leq \sigma_{\underline{\tau}^2/2}^2,$$
$$\sigma_z^2 := \mathrm{var}(Z_i) = \mathbb{E}\left(Z_i - \mu_z\right)^2 \leq \sigma_{\underline{\tau}^2/2}^2.$$

For third and higher order absolute moments, we have

$$\mathbb{E}|Z_i|^k = \mathbb{E}\left||\varepsilon_i| 1\left(\varepsilon_i^2 \leq \underline{\tau}^2/2 - r^2\right)\right|^k \leq \sigma_{\underline{\tau}^2/2}^2 \left(\frac{\underline{\tau}}{\sqrt{2}}\right)^{k-2} \leq \frac{k!}{2}\sigma_{\underline{\tau}^2/2}^2 \left(\frac{\underline{\tau}}{3\sqrt{2}}\right)^{k-2}, \quad \text{for all integers } k \geq 3.$$

Using Lemma J.2 with $v = n\sigma_{\underline{\tau}^2/2}^2$ and $c = \underline{\tau}/(3\sqrt{2})$, we have for any $t > 0$

$$\mathbb{P}\left(\sum_{i=1}^{n}|\varepsilon_i| 1\left(\varepsilon_i^2 \leq \frac{\underline{\tau}^2}{2} - r^2\right) - \sum_{i=1}^{n}\mathbb{E}|\varepsilon_i| 1\left(\varepsilon_i^2 \leq \frac{\underline{\tau}^2}{2} - r^2\right) \geq \sqrt{2n\sigma_{\underline{\tau}^2/2}^2 t} + \frac{\underline{\tau} t}{3\sqrt{2}}\right) \leq \exp\left(-t\right).$$

Taking $t = \log(1/\delta)$ acquires the desired result. $\square$

# H  Proofs for Section 4

This section collects proofs for results in Section 4.

## H.1  Proof of Theorem 4.2

*Proof of Theorem 4.2.* First, the MoM estimator $\widehat{\mu}^{\mathrm{MoM}} = M(z_1, \ldots, z_k)$ is equivalent to

$$\operatorname{argmin} \sum_{j=1}^{k} |z_j - \mu|.$$

For any $x \in \mathbb{R}$, let $\ell(x) = |x|$ and define $L(x) = \mathbb{E}\ell'(x + Z)$ where $Z \sim \mathcal{N}(0, 1)$ and

$$\ell'(x) = \begin{cases} 1, & \text{if } x > 0, \\ 0, & \text{if } x = 0, \\ -1, & \text{otherwise.} \end{cases}$$

If the assumptions of Theorem 4 of Minsker (2019) are satisfied, we obtain, after some algebra, that

$$\sqrt{n}\left(\widehat{\mu}^{\mathrm{MoM}} - \mu^*\right) \rightsquigarrow \mathcal{N}\left(0, \frac{\mathbb{E}(\ell'(Z))^2}{(L'(0))^2}\right).$$

Some algebra derives that

$$\frac{\mathbb{E}(\ell'(Z))^2}{(L'(0))^2} = \frac{\pi\sigma^2}{2}.$$

It remains to check the assumptions there. Assumptions (1), (4), and (5) trivially hold. Assumption (2) can be verified by using the following Berry-Esseen bound.

**Fact H.1.** Let $y_1, \ldots, y_m$ be i.i.d. random copies of $y$ with mean $\mu$, variance $\sigma^2$ and $\mathbb{E}|y - \mu|^{2+\iota} < \infty$ for some $\iota \in (0, 1]$. Then there exists an absolute constant $C$ such that

$$\sup_{t \in \mathbb{R}} \left| \mathbb{P}\left( \sqrt{m} \frac{\bar{y} - \mu}{\sigma} \leq t \right) - \Phi(t) \right| \leq C \frac{\mathbb{E}|y - \mu|^{2+\iota}}{\sigma^{2+\iota} m^{\iota/2}}.$$

It remains to check Assumption (3). Because $g(m) \lesssim m^{-\iota/2}$, $\sqrt{k}g(m) \lesssim \sqrt{k}m^{-\iota/2} \to 0$ if $k = o(n^{\iota/(1+\iota)})$ as $n \to \infty$. Thus Assumption (3) holds if $k = o(n^{\iota/(1+\iota)})$ and $k \to \infty$. This completes the proof. □

## H.2 Proof of Theorem 4.3

In this subsection, we state and prove a stronger result of Theorem 4.3, aka Theorem H.2. Theorem 4.3 can then be proved following the same proof under the assumption that $\mathbb{E}|\varepsilon_i|^{2+\iota} < \infty$ for any prefixed $0 < \iota \leq 1$.

**Theorem H.2.** Assume the same assumptions as in Theorem 3.4. Take $z^2 = 2\log(n)$. If $\mathbb{E}\varepsilon_i^4 < \infty$, then

$$\sqrt{n} \begin{bmatrix} \widehat{\mu} - \mu^* \\ \widehat{v} - v_* \end{bmatrix} \rightsquigarrow \mathcal{N}(0, \Sigma), \text{ where } \Sigma = \begin{bmatrix} \sigma^2 & \sigma \mathbb{E}\varepsilon_i^3/2 \\ \sigma \mathbb{E}\varepsilon_i^3/2 & (\sigma^2 \mathbb{E}\varepsilon_i^4 - \sigma^6)/4 \end{bmatrix}.$$

*Proof of Theorem H.2.* Now we are ready to analyze the self-tuned mean estimator $\widehat{\mu} = \widehat{\mu}(\widehat{v})$. For any $0 < \delta < 1$, following the proof of Theorem 3.4, we obtain with probability at least $1 - \delta$ that

$$|\widehat{\mu}(\widehat{v}) - \mu^*| \leq \sup_{v \in [v_0, V_0]} |\widehat{\mu}(v) - \mu^*| \leq 2C \cdot \frac{V_0^2}{v_0} \cdot \frac{\log(n/\delta)}{z\sqrt{n}}.$$

Taking $z^2 \geq \log(n/\delta)$ with $\delta = 1/n$ in the above inequality, we obtain $\widehat{\mu} \to \mu^*$ in probability. Theorem H.3 implies that $\widehat{v} \to \sigma$ in probability. Thus we have $\|\widehat{\theta} - \theta^*\|_2 \to 0$ in probability, where

$$\widehat{\theta} = (\widehat{\mu}, \widehat{v})^\mathrm{T}, \text{ and } \theta^* = (\mu^*, \sigma)^\mathrm{T}.$$

Using the Taylor's theorem for vector-valued functions, we obtain

$$\nabla L_n(\widehat{\theta}) = 0 = \nabla L_n(\theta^*) + H_n(\theta^*)(\widehat{\theta} - \theta^*) + \frac{R_2(\theta)}{2}(\widehat{\theta} - \theta^*)^{\otimes 2},$$

where $\otimes$ indicates the tensor product. Let $\tau_\sigma = \sigma\sqrt{n}/z$. We say that $X_n$ and $Y_n$ are asymptotically equivalent, denoted as $X_n \simeq Y_n$, if both $X_n$ and $Y_n$ converge in distribution to some same random variable/vector $Z$. Rearranging, we obtain

$$\sqrt{n}\left(\widehat{\theta} - \theta^*\right) \simeq [H_n(\theta^*)]^{-1}\left(-\sqrt{n}\nabla L_n(\theta^*)\right)$$

$$= \begin{bmatrix} \frac{\sqrt{n}}{z} \cdot \frac{1}{n}\sum_{i=1}^n \frac{\tau_\sigma^2}{(\tau_\sigma^2+\varepsilon_i^2)^{3/2}} & \frac{n}{z^2} \cdot \frac{1}{n}\sum_{i=1}^n \frac{\tau_\sigma\varepsilon_i}{(\tau_\sigma^2+\varepsilon_i^2)^{3/2}} \\ \frac{n}{z^2} \cdot \frac{1}{n}\sum_{i=1}^n \frac{\tau_\sigma\varepsilon_i}{(\tau_\sigma^2+\varepsilon_i^2)^{3/2}} & \frac{n^{3/2}}{z^3} \cdot \frac{1}{n}\sum_{i=1}^n \frac{\varepsilon_i^3}{(\tau_\sigma^2+\varepsilon_i^2)^{3/2}} \end{bmatrix}^{-1} \begin{bmatrix} \sqrt{n} \cdot \frac{1}{n}\sum_{i=1}^n \frac{\tau_\sigma\varepsilon_i}{\sigma\sqrt{\tau_\sigma^2+\varepsilon_i^2}} \\ \sqrt{n} \cdot \frac{n}{z^2}\frac{1}{n}\sum_{i=1}^n \frac{\sqrt{1+\varepsilon_i^2/\tau_\sigma^2}-1}{\sqrt{1+\varepsilon_i^2/\tau_\sigma^2}} - \sqrt{n} \cdot a \end{bmatrix}$$

$$\simeq \begin{bmatrix} \sigma & 0 \\ 0 & \sigma^3 \end{bmatrix} \begin{bmatrix} \sqrt{n} \cdot \frac{1}{n}\sum_{i=1}^n \frac{\tau_\sigma\varepsilon_i}{\sigma\sqrt{\tau_\sigma^2+\varepsilon_i^2}} \\ \sqrt{n} \cdot \frac{n}{z^2}\frac{1}{n}\sum_{i=1}^n \frac{\sqrt{1+\varepsilon_i^2/\tau_\sigma^2}-1}{\sqrt{1+\varepsilon_i^2/\tau_\sigma^2}} - \sqrt{n} \cdot a \end{bmatrix}$$

$$= \begin{bmatrix} \sigma & 0 \\ 0 & \sigma^3 \end{bmatrix} \begin{bmatrix} \mathrm{I} \\ \mathrm{II} \end{bmatrix},$$

where the second $\simeq$ uses the fact that

$$H_n(\theta^*) \xrightarrow{\text{a.s.}} \begin{bmatrix} \frac{1}{\sigma} & 0 \\ 0 & \frac{1}{\sigma^3} \end{bmatrix}.$$

We proceed to derive the asymptotic property of $(\mathrm{I}, \mathrm{II})^{\mathsf{T}}$. For I, we have

$$\mathrm{I} = \sqrt{n} \cdot \left( \frac{1}{n} \sum_{i=1}^{n} \frac{\tau_\sigma \varepsilon_i}{\sigma \sqrt{\tau_\sigma^2 + \varepsilon_i^2}} - \mathbb{E}\left[ \frac{\tau_\sigma \varepsilon_i}{\sigma \sqrt{\tau_\sigma^2 + \varepsilon_i^2}} \right] \right) + \sqrt{n} \cdot \mathbb{E}\left[ \frac{\tau_\sigma \varepsilon_i}{\sigma \sqrt{\tau_\sigma^2 + \varepsilon_i^2}} \right]$$

$$\rightsquigarrow \mathcal{N}\left( 0, \lim_{n\to\infty} \mathrm{var}\left[ \frac{\tau_\sigma \varepsilon_i}{\sigma \sqrt{\tau_\sigma^2 + \varepsilon_i^2}} \right] \right) + \lim_{n\to\infty} \sqrt{n} \cdot \mathbb{E}\left[ \frac{\tau_\sigma \varepsilon_i}{\sigma \sqrt{\tau_\sigma^2 + \varepsilon_i^2}} \right].$$

It remains to calculate

$$\lim_{n\to\infty} \mathbb{E}\left( \frac{\sqrt{n}\tau_\sigma \varepsilon_i}{\sqrt{\tau_\sigma^2 + \varepsilon_i^2}} \right) \quad \text{and} \quad \lim_{n\to\infty} \mathrm{var}\left[ \frac{\tau \varepsilon_i}{\sqrt{\tau_\sigma^2 + \varepsilon_i^2}} \right].$$

For the former term, if there exists some $0 < \iota \leq 1$ such that $\mathbb{E}|\varepsilon_i|^{2+\iota} < \infty$, using the fact that $\mathbb{E}\varepsilon_i = 0$, we have

$$\left| \mathbb{E}\left( \frac{\sqrt{n}\tau_\sigma \varepsilon_i}{\sqrt{\tau_\sigma^2 + \varepsilon_i^2}} \right) \right| = \sqrt{n}\tau_\sigma \cdot \left| \mathbb{E}\left\{ \frac{-\varepsilon_i/\tau_\sigma}{\sqrt{1 + \varepsilon_i^2/\tau_\sigma^2}} \right\} \right| = \sqrt{n}\tau_\sigma \cdot \left| \mathbb{E}\left\{ \frac{\tau_\sigma^{-1}\varepsilon_i \left( \sqrt{1 + \varepsilon_i^2/\tau_\sigma^2} - 1 \right)}{\sqrt{1 + \varepsilon_i^2/\tau_\sigma^2}} \right\} \right|$$

$$\leq \frac{\sqrt{n}\tau_\sigma}{2} \cdot \mathbb{E}\left| \frac{\varepsilon_i^3/\tau_\sigma^3}{\sqrt{1 + \varepsilon_i^2/\tau_\sigma^2}} \right| \leq \frac{\sqrt{n}\tau_\sigma}{2} \cdot \frac{\mathbb{E}|\varepsilon_i|^{2+\iota}}{\tau_\sigma^{2+\iota}}$$

$$\leq \frac{\sqrt{n}\,\mathbb{E}|\varepsilon_i|^{2+\iota}}{2\tau_\sigma^{1+\iota}} \to 0, \tag{H.1}$$

where the first inequality uses Lemma J.4 (ii) with $r = 1/2$, that is, $\sqrt{1+x} \leq 1 + x/2$ for $x \geq -1$. For the second term, we have

$$\lim_{n\to\infty} \mathrm{var}\left[ \frac{\tau_\sigma \varepsilon_i}{\sqrt{\tau_\sigma^2 + \varepsilon_i^2}} \right] = \lim_{n\to\infty} \mathbb{E}\left[ \frac{\tau_\sigma^2 \varepsilon_i^2}{\tau_\sigma^2 + \varepsilon_i^2} \right] = \sigma^2,$$

by the dominated convergence theorem. Thus

$$\mathrm{I} \rightsquigarrow \mathcal{N}(0, 1).$$

For II, recall $a = 1/2$ and using the facts that

$$\lim_{n\to\infty} \frac{n}{z^2} \cdot \mathbb{E}\left( \frac{\sqrt{1 + \varepsilon_i^2/\tau_\sigma^2} - 1}{\sqrt{1 + \varepsilon_i^2/\tau_\sigma^2}} \right) = \lim_{n\to\infty} \frac{n}{2\tau_\sigma^2 z^2} \cdot \mathbb{E}\left( \frac{1}{\sqrt{1 + \varepsilon_i^2/\tau_\sigma^2}} \cdot \frac{\sqrt{1 + \varepsilon_i^2/\tau_\sigma^2} - 1}{1/(2\tau_\sigma^2)} \right) = \frac{1}{2},$$

$$\lim_{n\to\infty} \sqrt{n} \cdot \left( \frac{n}{z^2} \cdot \mathbb{E}\left( \frac{\sqrt{1 + \varepsilon_i^2/\tau_\sigma^2} - 1}{\sqrt{1 + \varepsilon_i^2/\tau_\sigma^2}} \right) - \frac{1}{2} \right) = 0,$$

we have

$$\mathrm{II} = \sqrt{n} \cdot \frac{n}{z^2} \cdot \frac{1}{n} \sum_{i=1}^{n} \frac{\sqrt{1 + \varepsilon_i^2/\tau_\sigma^2} - 1}{\sqrt{1 + \varepsilon_i^2/\tau_\sigma^2}} - \sqrt{n} \cdot \frac{1}{2}$$

$$\simeq \sqrt{n} \cdot \frac{1}{n} \sum_{i=1}^{n} \left( \frac{n}{z^2} \cdot \frac{\sqrt{1 + \varepsilon_i^2/\tau_\sigma^2} - 1}{\sqrt{1 + \varepsilon_i^2/\tau_\sigma^2}} - \mathbb{E}\left( \frac{n}{z^2} \cdot \frac{\sqrt{1 + \varepsilon_i^2/\tau_\sigma^2} - 1}{\sqrt{1 + \varepsilon_i^2/\tau_\sigma^2}} \right) \right)$$

$$\simeq \mathcal{N}\left( 0, \lim_{n\to\infty} \mathrm{var}\left( \frac{n}{z^2} \cdot \frac{\sqrt{1 + \varepsilon_i^2/\tau_\sigma^2} - 1}{\sqrt{1 + \varepsilon_i^2/\tau_\sigma^2}} \right) \right).$$

If $\mathbb{E}\varepsilon_i^4 < \infty$, then

$$\lim_{n\to\infty} \text{var}\left(\frac{n}{z^2} \cdot \frac{\sqrt{1+\varepsilon_i^2/\tau_\sigma^2}-1}{\sqrt{1+\varepsilon_i^2/\tau_\sigma^2}}\right) = \frac{\mathbb{E}\varepsilon_i^4}{4\sigma^4} - \frac{1}{4},$$

and thus II $\simeq \mathcal{N}\left(0, (\mathbb{E}\varepsilon_i^4/\sigma^4 - 1)/4\right)$. For the cross covariance, we have

$$\lim_{n\to\infty} \text{cov}\left(\frac{\tau_\sigma\varepsilon_i}{\sigma\sqrt{\tau_\sigma^2+\varepsilon_i^2}}, \frac{n}{z^2} \cdot \frac{\sqrt{1+\varepsilon_i^2/\tau_\sigma^2}-1}{\sqrt{1+\varepsilon_i^2/\tau_\sigma^2}}\right)$$

$$= \lim_{n\to\infty} \mathbb{E}\left(\frac{\tau_\sigma\varepsilon_i}{\sigma\sqrt{\tau_\sigma^2+\varepsilon_i^2}} \cdot \frac{n}{z^2} \cdot \frac{\sqrt{1+\varepsilon_i^2/\tau_\sigma^2}-1}{\sqrt{1+\varepsilon_i^2/\tau_\sigma^2}}\right)$$

$$= \frac{\mathbb{E}\varepsilon_i^3}{2\sigma^3}.$$

Thus

$$\sqrt{n}\,(\widehat{\theta}-\theta^*) \rightsquigarrow \mathcal{N}(0, \Sigma),$$

where

$$\Sigma = \begin{bmatrix} \sigma & 0 \\ 0 & \sigma^3 \end{bmatrix}\begin{bmatrix} 1 & \mathbb{E}\varepsilon_i^3/(2\sigma^3) \\ \mathbb{E}\varepsilon_i^3/(2\sigma^3) & (\mathbb{E}\varepsilon_i^4/\sigma^4 - 1)/4 \end{bmatrix}\begin{bmatrix} \sigma & 0 \\ 0 & \sigma^3 \end{bmatrix} = \begin{bmatrix} \sigma^2 & \sigma\mathbb{E}\varepsilon_i^3/2 \\ \sigma\mathbb{E}\varepsilon_i^3/2 & (\sigma^2\mathbb{E}\varepsilon_i^4 - \sigma^6)/4 \end{bmatrix}.$$

Therefore, for $\widehat{\mu}$ only, we have

$$\sqrt{n}\,(\widehat{\mu}-\mu^*) \rightsquigarrow \mathcal{N}(0, \sigma^2).$$

$\square$

## H.3   Consistency of $\widehat{v}$

This subsection proves that $\widehat{v}$ is a consistent estimator of $\sigma$ when the $(2+\iota)$-th moment exists. Recall that

$$\nabla_v L_n(\mu, v) = \frac{n}{z^2} \cdot \frac{1}{n}\sum_{i=1}^n \left(\frac{\tau}{\sqrt{\tau^2+(y_i-\mu)^2}} - 1\right) + a$$

where $a = 1/2$. We emphasize that the following proof only needs the second moment assumption $\sigma^2 = \mathbb{E}\varepsilon_i^2 < \infty$.

**Theorem H.3** (Consistency of $\widehat{v}$)**.** Assume the same assumptions as in Theorem 3.4. Take $z^2 \geq \log(n)$. Then

$$\widehat{v} \longrightarrow \sigma \quad \text{in probability.}$$

*Proof of Theorem H.3.* By the proof of Theorem 3.4, we obtain with probability at least $1-\delta$ that the following two results hold simultaneously:

$$\sup_{v\in[v_0, V_0]} |\widehat{\mu}(v) - \mu^*| \leq 2C \cdot \frac{V_0^2}{v_0} \cdot \frac{\log(n/\delta)}{z\sqrt{n}} =: r, \tag{H.2}$$

$$v_0 < c_0\sigma_{\tau_{v_0}^2-1} \leq \widehat{v} \leq C_0\sigma < V_0, \tag{H.3}$$

provided that $z^2 \geq \log(5/\delta)$ and $n$ is large enough. Therefore, the constraint in the optimization problem (3.1) is not active, and thus

$$\nabla_v L_n(\widehat{\mu}, \widehat{v}) = 0.$$

Using Lemma H.4 together with the equality above, we obtain with probability at least $1 - \delta$ that

$$\frac{c_0}{V_0^3}|\widehat{v} - \sigma|^2 \leq \frac{c_0}{\widehat{v}^3 \vee \sigma^3}|\widehat{v} - \sigma|^2 \leq \rho_\ell|\widehat{v} - \sigma|^2$$

$$\leq \langle \nabla_v L_n(\widehat{\mu}, \widehat{v}) - \nabla_v L_n(\widehat{\mu}, \sigma), \widehat{v} - \sigma \rangle$$

$$\leq |\nabla_v L_n(\widehat{\mu}, \sigma)| \, |\widehat{v} - \sigma|$$

$$\leq \left| \frac{n}{z^2} \cdot \frac{1}{n} \sum_{i=1}^{n} \left( \frac{\tau_\sigma}{\sqrt{\tau_\sigma^2 + (y_i - \widehat{\mu})^2}} - 1 \right) + a \right| \, |\widehat{v} - \sigma|.$$

Plugging (H.2) into the above inequality and canceling $|\widehat{v} - \sigma|$ on both sides, we obtain with probability at least $1 - 2\delta$ that

$$\frac{c_0}{V_0^3}|\widehat{v} - \sigma| \leq \left| \frac{n}{z^2} \cdot \frac{1}{n} \sum_{i=1}^{n} \left( \frac{\tau_\sigma}{\sqrt{\tau_\sigma^2 + (y_i - \widehat{\mu})^2}} - 1 \right) + a \right|$$

$$\leq \sup_{\mu \in \mathbb{B}_r(\mu^*)} \left| \frac{n}{z^2} \cdot \frac{1}{n} \sum_{i=1}^{n} \left( \frac{\tau_\sigma}{\sqrt{\tau_\sigma^2 + (y_i - \mu)^2}} - 1 \right) + a \right|$$

$$= \frac{n}{z^2} \cdot \sup_{\mu \in \mathbb{B}_r(\mu^*)} \left| \frac{1}{n} \sum_{i=1}^{n} \left( \frac{\tau_\sigma}{\sqrt{\tau_\sigma^2 + (y_i - \mu)^2}} - 1 \right) + \frac{az^2}{n} \right|$$

$$\leq \frac{n}{z^2} \cdot \sup_{\mu \in \mathbb{B}_r(\mu^*)} \left| \frac{1}{n} \sum_{i=1}^{n} \left( 1 - \frac{\tau_\sigma}{\sqrt{\tau_\sigma^2 + (y_i - \mu)^2}} \right) - \mathbb{E}\left( 1 - \frac{\tau_\sigma}{\sqrt{\tau_\sigma^2 + (y_i - \mu)^2}} \right) \right|$$

$$+ \frac{n}{z^2} \cdot \sup_{\mu \in \mathbb{B}_r(\mu^*)} \left| \mathbb{E}\left( 1 - \frac{\tau_\sigma}{\sqrt{\tau_\sigma^2 + (y_i - \mu)^2}} \right) - \frac{az^2}{n} \right|$$

$$=: \mathrm{I} + \mathrm{II}.$$

It remains to bound terms I and II. We start with term II. Let $r_i^2 = (y_i - \mu)^2$. We have

$$\mathrm{II} = \frac{n}{z^2} \cdot \sup_{\mu \in \mathbb{B}_r(\mu^*)} \left| \mathbb{E}\left( 1 - \frac{\tau_\sigma}{\sqrt{\tau_\sigma^2 + (y_i - \mu)^2}} \right) - \frac{az^2}{n} \right|$$

$$= \max\left\{ \sup_{\mu \in \mathbb{B}_r(\mu^*)} \left( \frac{n}{z^2} \cdot \mathbb{E}\frac{\sqrt{1 + r_i^2/\tau_\sigma^2} - 1}{\sqrt{1 + r_i^2/\tau_\sigma^2}} - a \right), \sup_{\mu \in \mathbb{B}_r(\mu^*)} \left( a - \frac{n}{z^2} + \mathbb{E}\frac{1}{\sqrt{1 + r_i^2/\tau_\sigma^2}} \right) \right\}$$

$$=: \mathrm{II}_1 \vee \mathrm{II}_2.$$

In order to bound II, we bound $\mathrm{II}_1$ and $\mathrm{II}_2$ respectively. For term $\mathrm{II}_1$, using Lemma J.4 (ii), aka $(1 + x)^r \leq 1 + rx$ for $x \geq -1$ and $r \in (0, 1)$, and $a = 1/2$, we have

$$\mathrm{II}_1 = \sup_{\mu \in \mathbb{B}_r(\mu^*)} \left( \frac{n}{z^2} \cdot \mathbb{E}\frac{\sqrt{1 + r_i^2/\tau_\sigma^2} - 1}{\sqrt{1 + r_i^2/\tau_\sigma^2}} - a \right)$$

$$\leq \sup_{\mu \in \mathbb{B}_r(\mu^*)} \left\{ \frac{n}{z^2} \cdot \left( 1 + \mathbb{E}\frac{r_i^2}{2\tau_\sigma^2} - 1 \right) - a \right\}$$

$$\leq \frac{n}{z^2} \cdot \mathbb{E}\frac{\varepsilon_i^2 + 2r|\varepsilon_i| + r^2|}{2\tau_\sigma^2} - \frac{1}{2} \qquad (a = 1/2)$$

$$\leq \frac{r}{\sigma}\left( 1 + \frac{r}{2\sigma} \right)$$

$$\leq \frac{2r}{\sigma}$$

if $n$ is large enough such that $r \le 2\sigma$. To bound $\text{II}_2$, we need Lemma E.1. Specifically, for any $0 \le \gamma < 1$, we have

$$(1+x)^{-1} \le 1 - (1-\gamma)x, \text{ for any } 0 \le x \le \frac{\gamma}{1-\gamma}.$$

Using this result, we obtain

$$
\mathbb{E}\frac{1}{\sqrt{1 + r_i^2/\tau_\sigma^2}} \le \sqrt{\mathbb{E}\frac{1}{1 + r_i^2/\tau_\sigma^2}} \qquad \text{(concavity of } \sqrt{x}\text{)}
$$

$$
\le \sqrt{\mathbb{E}\left\{\left(1 - \frac{(1-\gamma)r_i^2}{\tau_\sigma^2}\right)1\left(\frac{r_i^2}{\tau_\sigma^2} \le \frac{\gamma}{1-\gamma}\right) + \frac{1}{1 + r_i^2/\tau_\sigma^2}1\left(\frac{r_i^2}{\tau_\sigma^2} > \frac{\gamma}{1-\gamma}\right)\right\}}
$$

$$
\le \sqrt{1 - (1-\gamma)\,\mathbb{E}\left(\frac{r_i^2}{\tau_\sigma^2}1\left(\frac{r_i^2}{\tau_\sigma^2} \le \frac{\gamma}{1-\gamma}\right)\right)} \qquad \text{(Lemma E.1)}
$$

$$
\le \sqrt{1 - (1-\gamma)\,\mathbb{E}\left(\frac{r_i^2}{\tau_\sigma^2}1\left(\frac{r_i^2}{\tau_\sigma^2} \le \frac{\gamma}{1-\gamma}\right)\right)}
$$

$$
\le \sqrt{1 - (1-\gamma)\,\mathbb{E}\left(\frac{\varepsilon_i^2 - 2r|\varepsilon_i| + r^2}{\tau_\sigma^2}1\left(\frac{2(\varepsilon_i^2 + r^2)}{\tau_\sigma^2} \le \frac{\gamma}{1-\gamma}\right)\right)} \qquad (\forall\, \mu \in \mathbb{B}_r(\mu^*))
$$

$$
\le 1 - \frac{1-\gamma}{2}\,\mathbb{E}\left(\frac{\varepsilon_i^2 - 2r|\varepsilon_i| + r^2}{\tau_\sigma^2}1\left(\frac{2(\varepsilon_i^2 + r^2)}{\tau_\sigma^2} \le \frac{\gamma}{1-\gamma}\right)\right),
$$

where the first inequality uses the concavity of $\sqrt{x}$, the third inequality uses Lemma E.1, and the last inequality uses the inequality that $(1+x)^{-1} \le 1 - x/2$ for $x \in [0,1]$, aka Lemma J.4 (iii) with $r = -1$, provided that

$$
(1-\gamma)\,\mathbb{E}\left(\frac{\varepsilon_i^2 - 2r|\varepsilon_i| - r^2}{\tau_\sigma^2}1\left(\frac{2(\varepsilon_i^2 + r^2)}{\tau_\sigma^2} \le \frac{\gamma}{1-\gamma}\right)\right) \le (1-\gamma)\frac{\sigma^2 - 2r\sigma - r^2}{\tau_\sigma^2} \le 1.
$$

Thus term $\text{II}_2$ can be bounded as

$$
\text{II}_2 = \sup_{\mu \in \mathbb{B}_r(\mu^*)}\left(a - \frac{n}{z^2} + \frac{n}{z^2}\cdot\mathbb{E}\frac{1}{\sqrt{1 + r_i^2/\tau_\sigma^2}}\right)
$$

$$
\le a - \frac{n}{z^2} + \frac{n}{z^2}\cdot\left\{1 - \frac{1-\gamma}{2}\,\mathbb{E}\left(\frac{\varepsilon_i^2 - 2r|\varepsilon_i| + r^2}{\tau_\sigma^2}1\left(\frac{2(\varepsilon_i^2 + r^2)}{\tau_\sigma^2} \le \frac{\gamma}{1-\gamma}\right)\right)\right\}
$$

$$
\le a - \frac{1-\gamma}{2\sigma^2}\cdot\mathbb{E}\varepsilon_i^2 + \frac{1-\gamma}{2\sigma^2}\cdot 2r\cdot\mathbb{E}(|\varepsilon_i|)
$$

$$
\le a - \frac{1-\gamma}{2} + \frac{r(1-\gamma)}{\sigma}
$$

$$
= \frac{\gamma}{2} + \frac{r(1-\gamma)}{\sigma}. \qquad (a = 1/2)
$$

Combining the upper bound for $\text{II}_1$ and $\text{II}_2$ and using the fact that, we obtain

$$
\text{II} \le \max\{\text{II}_1, \text{II}_2\} \le \frac{\gamma}{2} + \frac{2r}{\sigma} \to 0,
$$

if $\gamma = \gamma(n) \to 0$.

We proceed to bound I. Recall that

$$
\text{I} = \frac{n}{z^2}\cdot\sup_{\mu \in \mathbb{B}_r(\mu^*)}\left|\frac{1}{n}\sum_{i=1}^{n}\left(1 - \frac{\tau_\sigma}{\sqrt{\tau_\sigma^2 + (y_i - \mu)^2}}\right) - \mathbb{E}\left(1 - \frac{\tau_\sigma}{\sqrt{\tau_\sigma^2 + (y_i - \mu)^2}}\right)\right|.
$$

For any $0 < \epsilon \le 2r$, there exists an $\epsilon$-cover $\mathcal{N} \subseteq \mathbb{B}_r(\mu^*)$ of $\mathbb{B}_r(\mu^*)$ such that $|\mathcal{N}| \le 6r/\epsilon$. Then for any $\mu \in \mathbb{B}_r(\mu^*)$ there exists a $\omega \in \mathcal{N}$ such that $|\omega - \mu| \le \gamma$, and

$$\left| \frac{1}{n} \sum_{i=1}^{n} \left( 1 - \frac{\tau_\sigma}{\sqrt{\tau_\sigma^2 + (y_i - \mu)^2}} \right) - \mathbb{E}\left( 1 - \frac{\tau_\sigma}{\sqrt{\tau_\sigma^2 + (y_i - \mu)^2}} \right) \right|$$

$$= \left| \frac{1}{n} \sum_{i=1}^{n} \frac{\sqrt{1 + (y_i - \mu)^2/\tau_\sigma^2} - 1}{\sqrt{1 + (y_i - \mu)^2/\tau_\sigma^2}} - \mathbb{E}\frac{\sqrt{1 + (y_i - \mu)^2/\tau_\sigma^2} - 1}{\sqrt{1 + (y_i - \mu)^2/\tau_\sigma^2}} \right|$$

$$\le \left| \frac{1}{n} \sum_{i=1}^{n} \frac{\sqrt{1 + (y_i - \omega)^2/\tau_\sigma^2} - 1}{\sqrt{1 + (y_i - \omega)^2/\tau_\sigma^2}} - \mathbb{E}\frac{\sqrt{1 + (y_i - \omega)^2/\tau_\sigma^2} - 1}{\sqrt{1 + (y_i - \omega)^2/\tau_\sigma^2}} \right|$$

$$+ \left| \frac{1}{n} \sum_{i=1}^{n} \frac{\sqrt{1 + (y_i - \mu)^2/\tau_\sigma^2} - 1}{\sqrt{1 + (y_i - \mu)^2/\tau_\sigma^2}} - \frac{1}{n} \sum_{i=1}^{n} \frac{\sqrt{1 + (y_i - \omega)^2/\tau_\sigma^2} - 1}{\sqrt{1 + (y_i - \omega)^2/\tau_\sigma^2}} \right|$$

$$+ \left| \mathbb{E}\frac{\sqrt{1 + (y_i - \mu)^2/\tau_\sigma^2} - 1}{\sqrt{1 + (y_i - \mu)^2/\tau_\sigma^2}} - \mathbb{E}\frac{\sqrt{1 + (y_i - \omega)^2/\tau_\sigma^2} - 1}{\sqrt{1 + (y_i - \omega)^2/\tau_\sigma^2}} \right|$$

$$= I_1 + I_2 + I_3.$$

For $I_1$, using Lemma G.2 acquires with probability at least $1 - 2\delta$ that

$$I_1 \le \sqrt{\frac{\mathbb{E}(y_i - \omega)^2 \log(1/\delta)}{n\tau_\sigma^2}} + \frac{\log(1/\delta)}{3n}$$

$$\le \sqrt{\frac{2(\sigma^2 + r^2)\log(1/\delta)}{n\tau_\sigma^2}} + \frac{\log(1/\delta)}{3n}$$

$$\le \frac{2z\sqrt{\log(1/\delta)}}{n} + \frac{\log(1/\delta)}{3n}$$

provided $r^2 \le \sigma^2$. Let

$$g(x) = -\frac{1}{n} \sum_{i=1}^{n} \frac{\tau}{\sqrt{\tau^2 + (x + \varepsilon_i)^2}}.$$

Using the mean value theorem and the inequality that $|x/(1 + x^2)^{3/2}| \le 1/2$, we obtain

$$|g(x) - g(y)| = \left| \frac{1}{n} \sum_{i=1}^{n} \frac{(\widetilde{x} + \varepsilon_i)/\tau_\sigma}{(1 + (\widetilde{x} + \varepsilon_i)^2/\tau_\sigma^2)^{3/2}} \cdot \frac{x - y}{\tau_\sigma} \right| \le \frac{|x - y|}{2\tau_\sigma},$$

where $\widetilde{x}$ is some convex combination of $x$ and $y$. Then we have

$$I_2 = \left| \frac{1}{n} \sum_{i=1}^{n} \frac{(\widetilde{\Delta} + \varepsilon_i)/\tau_\sigma}{(1 + (\widetilde{\Delta} + \varepsilon_i)^2/\tau_\sigma^2)^{3/2}} \cdot \frac{\Delta_\mu - \Delta_\omega}{\tau_\sigma} \right| \le \frac{\epsilon}{2\tau_\sigma}$$

where $\widetilde{\Delta}$ is some convex combination of $\Delta_w = \mu^* - w$ and $\Delta_\mu = \mu^* - \mu$. For $II_3$, a similar argument for bounding $II_2$ yields

$$I_3 = \left| \mathbb{E}\left( \frac{(\widetilde{\Delta} + \varepsilon_i)/\tau_\sigma}{(1 + (\widetilde{\Delta} + \varepsilon_i)^2/\tau_\sigma^2)^{3/2}} \right) \cdot \frac{\Delta_\mu - \Delta_\omega}{\tau_\sigma} \right|$$

$$\le \mathbb{E}|\widetilde{\Delta} + \varepsilon_i| \cdot \frac{\epsilon}{\tau_\sigma^2}$$

$$\le \frac{\epsilon\sqrt{2(r^2 + \sigma^2)}}{\tau_\sigma^2},$$

where the last inequality uses Jensen's inequality, i.e. $\mathbb{E}|\widetilde{\Delta} + \varepsilon_i| \leq \sqrt{\mathbb{E}(\widetilde{\Delta} + \varepsilon_i^2)} \leq \sqrt{2(r^2 + \sigma^2)}$. Putting the above pieces together and using the union bound, we obtain with probability at least $1 - 12\epsilon^{-1}r\delta$

$$
\begin{aligned}
\mathrm{I} &\leq \frac{n}{z^2} \cdot \sup_{\omega \in \mathcal{N}} \left| \frac{1}{n} \sum_{i=1}^{n} \frac{\sqrt{1 + (y_i - \omega)^2/\tau_\sigma^2} - 1}{\sqrt{1 + (y_i - \omega)^2/\tau_\sigma^2}} - \mathbb{E} \frac{\sqrt{1 + (y_i - \omega)^2/\tau_\sigma^2} - 1}{\sqrt{1 + (y_i - \omega)^2/\tau_\sigma^2}} \right| \\
&\quad + \frac{n}{z^2} \cdot \frac{\epsilon}{2\tau_\sigma} \left( 1 + \frac{2\sqrt{2(r^2 + \sigma^2)}}{\tau_\sigma} \right) \\
&\leq \frac{2\sqrt{\log(1/\delta)}}{z} + \frac{\log(1/\delta)}{3z^2} + \frac{\epsilon\sqrt{n}}{\sigma z},
\end{aligned}
$$

provided that

$$
2\sqrt{2(r^2 + \sigma^2)} \leq \tau_\sigma.
$$

Putting above results together, we obtain with probability at least $1 - (12r/\epsilon + 2)\delta$ that

$$
\begin{aligned}
|\widehat{v} - \sigma| &\lesssim \mathrm{I} + \mathrm{II} \\
&\leq \frac{2\sqrt{\log(1/\delta)}}{z} + \frac{\log(1/\delta)}{3z^2} + \frac{\epsilon\sqrt{n}}{\sigma z} + \frac{\gamma}{2} + \frac{2r}{\sigma}.
\end{aligned}
$$

Let $C' = 24CV_0^2/v_0$. Therefore, taking $\epsilon = 1/\sqrt{n}$, $\delta = 1/\log n$, and $z^2 \geq \log(n)$, we obtain with probability at least

$$
1 - \frac{C'\left(\sqrt{\log n} + \log\log n/\sqrt{\log n}\right) + 2}{\log n}
$$

that

$$
|\widehat{v} - \sigma| \lesssim \sqrt{\frac{\log\log n}{\log n}} + \frac{\log\log n}{\log n} + \frac{1}{\sqrt{\log n}} + \gamma + r \to 0.
$$

Therefore $\widehat{v} \to \sigma$ in probability. This finishes the proof.

$\square$

## H.4 Local strong convexity in $v$

In this section, we first present the local strong convexity of the empirical loss function with respect to $v$ uniformly over a neighborhood of $\mu^*$.

**Lemma H.4** (Local strong convexity in $v$). Let $\mathbb{B}_r(\mu^*) = \{\mu : |\mu - \mu^*| \leq r\}$. Assume $r = r(n) = o(1)$. Let $0 < \delta < 1$ and $n$ is sufficiently large. Take $\varpi$ such that $\max\{\varpi r\sqrt{n}, \varpi\} \to 0$ and $\varpi\sqrt{n} \to \infty$. Then, with probability at least $1 - \delta$, we have

$$
\inf_{\mu \in \mathbb{B}_r(\mu^*)} \frac{\langle \nabla_v L_n(\mu, v) - \nabla_v L_n(\mu, v_*), v - \sigma \rangle}{|v - \sigma|^2} \geq \rho_\ell = \frac{\sigma_{c\varpi^2 n/(4z^2)}^2}{2(v^3 \vee \sigma^3)} \geq \frac{c_0}{v^3 \vee \sigma^3},
$$

where $c$ and $c_0$ are some constants.

*Proof of Lemma H.4.* Recall $\tau = v\sqrt{n}/z$. For notational simplicity, write $\tau_\sigma = \sigma\sqrt{n}/z$, $\tau_{v_0} = v_0\sqrt{n}/z$, $\tau_\varpi = \varpi\sqrt{n}/z$, and $\Delta = \mu^* - \mu$. It follows that

$$
\begin{aligned}
\langle \nabla_v L_n(\mu, v) - \nabla_v L_n(\mu, \sigma), v - \sigma \rangle &= \frac{n}{z^2} \left\langle \frac{1}{n}\sum_{i=1}^n \frac{\tau}{\sqrt{\tau^2 + (y_i - \mu)^2}} - \frac{1}{n}\sum_{i=1}^n \frac{\tau_\sigma}{\sqrt{\tau_\sigma^2 + (y_i - \mu)^2}}, v - \sigma \right\rangle \\
&= \frac{n^{3/2}}{z^3} \cdot \frac{1}{n}\sum_{i=1}^n \frac{(y_i - \mu)^2}{(\widetilde{\tau}^2 + (y_i - \mu)^2)^{3/2}} |v - \sigma|^2 \\
&\geq \frac{n^{3/2}}{z^3} \cdot \frac{1}{n}\sum_{i=1}^n \frac{(y_i - \mu)^2}{((\tau \vee \tau_\sigma)^2 + (y_i - \mu)^2)^{3/2}} |v - \sigma|^2
\end{aligned}
$$

where $\widetilde{\tau}$ is some convex combination of $\tau$ and $\tau_\sigma$, that is $\widetilde{\tau} = (1 - \lambda)\tau_\sigma + \lambda\tau$ for some $\lambda \in [0, 1]$. Because $\tau^3 x^2/(\tau^2 + x^2)^{3/2}$ is an increasing function of $\tau$, if $\tau_\varpi \leq \tau \vee \tau_\sigma$, we have

$$
\begin{aligned}
\frac{\langle \nabla_v L_n(\mu, v) - \nabla_v L_n(\mu, \sigma), v - v_* \rangle}{|v - \sigma|^2} &\geq \frac{n^{3/2}}{z^3(\tau \vee \tau_\sigma)^3} \cdot \frac{1}{n}\sum_{i=1}^n \frac{(\tau \vee \tau_\sigma)^3(y_i - \mu)^2}{(\tau^2 \vee \tau_\sigma^2 + (y_i - \mu)^2)^{3/2}} \\
&\geq \frac{n^{3/2}}{z^3(\tau \vee \tau_\sigma)^3} \cdot \frac{1}{n}\sum_{i=1}^n \frac{\tau_\varpi^3(y_i - \mu)^2}{(\tau_\varpi^2 + (y_i - \mu)^2)^{3/2}}.
\end{aligned}
$$

Thus

$$
\begin{aligned}
&\inf_{\mu \in \mathbb{B}_r(\mu^*)} \frac{\langle \nabla_v L_n(\mu, v) - \nabla_v L_n(\mu, \sigma), v - v_* \rangle}{|v - \sigma|^2} \\
&\geq \frac{n^{3/2}}{z^3(\tau \vee \tau_*)^3} \cdot \inf_{\mu \in \mathbb{B}_r(\mu^*)} \frac{1}{n}\sum_{i=1}^n \frac{\tau_\varpi^3(y_i - \mu)^2}{(\tau_\varpi^2 + (y_i - \mu)^2)^{3/2}} \\
&= \frac{n^{3/2}}{z^3(\tau \vee \tau_\sigma)^3} \cdot \left( \inf_{\mu \in \mathbb{B}_r(\mu^*)} \left( \mathbb{E}\frac{\tau_\varpi^3(y_i - \mu)^2}{(\tau_\varpi^2 + (y_i - \mu)^2)^{3/2}} \right) \right. \\
&\qquad\qquad \left. - \sup_{\mu \in \mathbb{B}_r(\mu^*)} \left| \frac{1}{n}\sum_{i=1}^n \frac{\tau_\varpi^3(y_i - \mu)^2}{(\tau_\varpi^2 + (y_i - \mu)^2)^{3/2}} - \mathbb{E}\frac{\tau_\varpi^3(y_i - \mu)^2}{(\tau_\varpi^2 + (y_i - \mu)^2)^{3/2}} \right| \right) \\
&= \frac{n^{3/2}}{z^3(\tau \vee \tau_\sigma)^3} \cdot (\mathrm{I} - \mathrm{II}).
\end{aligned}
$$

It remains to lower bound I and upper bound II. We start with I. Let $f(x) = x/(1 + x)^{3/2}$ which satisfies

$$
f(x) \geq \begin{cases} \epsilon x & x \leq c_\epsilon \\ 0 & x > c_\epsilon, \end{cases}
$$

and $Z = (y - \mu)^2/\tau_\varpi^2$ in which $y \sim y_i$. Suppose $r^2 \leq c_\epsilon \tau_\varpi^2/4$, then we have

$$
\begin{aligned}
\inf_{\mu \in \mathbb{B}_r(\mu^*)} \left( \mathbb{E}\frac{\tau_\varpi^3(y_i - \mu)^2}{(\tau_\varpi^2 + (y_i - \mu)^2)^{3/2}} \right) &= \inf_{\mu \in \mathbb{B}_r(\mu^*)} \mathbb{E}\left( \frac{\tau_\varpi^2 Z}{(1 + Z)^{3/2}} \right) \\
&\geq \epsilon \cdot \inf_{\mu \in \mathbb{B}_r(\mu^*)} \mathbb{E}\left[ (y - \mu)^2 1((y - \mu)^2 \leq c_\epsilon \tau_\varpi^2) \right] \\
&\geq \epsilon \cdot \inf_{\mu \in \mathbb{B}_r(\mu^*)} \mathbb{E}\left[ (y - \mu)^2 1(\varepsilon^2 \leq c_\epsilon \tau_\varpi^2/2 - r^2) \right] \\
&\geq \epsilon \cdot \inf_{\mu \in \mathbb{B}_r(\mu^*)} \left( \mathbb{E}\left[ (\Delta^2 + \varepsilon^2)1\left( \varepsilon^2 \leq \frac{c_\epsilon \tau_\varpi^2}{4} \right) \right] - \frac{8\Delta\sigma^2}{c_\epsilon \tau_\varpi^2} \right) \\
&\geq \epsilon \cdot \left( \mathbb{E}\left[ \varepsilon^2 1\left( \varepsilon^2 \leq \frac{c_\epsilon \tau_\varpi^2}{4} \right) \right] - \frac{8r\sigma^2}{c_\epsilon \tau_\varpi^2} \right).
\end{aligned}
$$

We then proceed with II. For any $0 < \gamma \leq 2r$, there exists an $\gamma$-cover $\mathcal{N}$ of $\mathbb{B}_r(\mu^*)$ such that $|\mathcal{N}| \leq 6r/\gamma$. Then for any $\mu \in \mathbb{B}_r(\mu^*)$ there exists an $\omega \in \mathcal{N}$ such that $|\omega - \mu| \leq \gamma$, and thus by Lemma H.5 we have

$$
\left| \frac{1}{n} \sum_{i=1}^{n} \frac{\tau_\varpi^3 (y_i - \mu)^2}{(\tau_\varpi^2 + (y_i - \mu)^2)^{3/2}} - \mathbb{E} \frac{\tau_\varpi^3 (y_i - \mu)^2}{(\tau_\varpi^2 + (y_i - \mu)^2)^{3/2}} \right|
$$

$$
\leq \left| \frac{1}{n} \sum_{i=1}^{n} \frac{\tau_\varpi^3 (y_i - \omega)^2}{(\tau_\varpi^2 + (y_i - \omega)^2)^{3/2}} - \mathbb{E} \frac{\tau_\varpi^3 (y_i - \omega)^2}{(\tau_\varpi^2 + (y_i - \omega)^2)^{3/2}} \right|
$$

$$
+ \left| \frac{1}{n} \sum_{i=1}^{n} \frac{\tau_\varpi^3 (y_i - \omega)^2}{(\tau_\varpi^2 + (y_i - \omega)^2)^{3/2}} - \frac{1}{n} \sum_{i=1}^{n} \frac{\tau_\varpi^3 (y_i - \mu)^2}{(\tau_\varpi^2 + (y_i - \mu)^2)^{3/2}} \right|
$$

$$
+ \left| \mathbb{E} \frac{\tau_\varpi^3 (y_i - \omega)^2}{(\tau_\varpi^2 + (y_i - \omega)^2)^{3/2}} - \mathbb{E} \frac{\tau_\varpi^3 (y_i - \mu)^2}{(\tau_\varpi^2 + (y_i - \mu)^2)^{3/2}} \right|
$$

$$
= \mathrm{II}_1 + \mathrm{II}_2 + \mathrm{II}_3.
$$

For $\mathrm{II}_1$, Lemma H.5 implies with probability at least $1 - 2\delta$

$$
\mathrm{II}_1 \leq \sqrt{\frac{2\tau_\varpi^2 \mathbb{E}(y_i - \omega)^2 \log(1/\delta)}{3n}} + \frac{\tau_\varpi^2 \log(1/\delta)}{3\sqrt{3}n} \leq \sqrt{\frac{2\tau_\varpi^2 (\sigma^2 + r^2) \log(1/\delta)}{3n}} + \frac{\tau_\varpi^2 \log(1/\delta)}{3\sqrt{3}n}.
$$

Let

$$
g(x) = \frac{1}{n} \sum_{i=1}^{n} \frac{\tau^3 (x + \varepsilon_i)^2}{(\tau^2 + (x + \varepsilon_i)^2)^{3/2}}.
$$

Using the mean value theorem and the inequality that $|\tau^2 x/(\tau^2 + x^2)^{3/2}| \leq 1/\sqrt{3}$, we obtain

$$
|g(x) - g(y)| = \left| \frac{1}{n} \sum_{i=1}^{n} \frac{\tau^3 (\widetilde{x} + \varepsilon_i)\left(\tau^2 - (\widetilde{x} + \varepsilon_i)^2\right)}{(\tau^2 + (\widetilde{x} + \varepsilon_i)^2)^{5/2}} (x - y) \right| \leq \frac{\tau}{\sqrt{3}} |x - y|.
$$

Then we have

$$
\mathrm{II}_2 = \left| \frac{1}{n} \sum_{i=1}^{n} \frac{\tau_\varpi^3 (\widetilde{\Delta} + \varepsilon_i)\left(\tau_\varpi^2 - (\widetilde{\Delta} + \varepsilon_i)^2\right)}{(\tau_\varpi^2 + (\widetilde{\Delta} + \varepsilon_i)^2)^{5/2}} (\Delta_w - \Delta_\mu) \right| \leq \frac{\tau_\varpi \gamma}{\sqrt{3}}
$$

where $\widetilde{\Delta}$ is some convex combination of $\Delta_w = \mu^* - w$ and $\Delta_\mu = \mu^* - \mu$. For $\mathrm{II}_3$, we have

$$
\mathrm{II}_3 = \left| \mathbb{E} \left( \frac{\tau_\varpi^3 (\widetilde{\Delta} + \varepsilon_i)\left(\tau_\varpi^2 - (\widetilde{\Delta} + \varepsilon_i)^2\right)}{(\tau_\varpi^2 + (\widetilde{\Delta} + \varepsilon_i)^2)^{5/2}} \right) (\Delta_w - \Delta_\mu) \right| \leq \gamma \mathbb{E}|\widetilde{\Delta} + \varepsilon_i| \leq \gamma \sqrt{\mathbb{E}\left(\widetilde{\Delta} + \varepsilon_i\right)^2},
$$

where the last inequality uses Jensen's inequality. Putting the above pieces together and using the union bound, we obtain with probability at least $1 - 12\gamma^{-1}r\delta$

$$
\mathrm{II} \leq \sup_{\omega \in \mathcal{N}} \left| \frac{1}{n} \sum_{i=1}^{n} \frac{\tau_\varpi^3 (y_i - \omega)^2}{(\tau_\varpi^2 + (y_i - \omega)^2)^{3/2}} - \mathbb{E} \frac{\tau_\varpi^3 (y_i - \omega)^2}{(\tau_\varpi^2 + (y_i - \omega)^2)^{3/2}} \right| + \frac{\tau_\varpi \gamma}{\sqrt{3}} + \gamma \sqrt{r^2 + \sigma^2}
$$

$$
\leq \sqrt{\frac{2\tau_\varpi^2 (r^2 + \sigma^2) \log(1/\delta)}{3n}} + \frac{\tau_\varpi^2 \log(1/\delta)}{3\sqrt{3}n} + \frac{\tau_\varpi \gamma}{\sqrt{3}} + \gamma \sqrt{r^2 + \sigma^2}
$$

$$
= \sqrt{r^2 + \sigma^2} \left( \sqrt{\frac{2\varpi^2 \log(1/\delta)}{3z^2}} + \gamma \right) + \frac{\varpi^2 \log(1/\delta)}{3\sqrt{3}z^2} + \frac{\varpi \gamma \sqrt{n}}{\sqrt{3}}.
$$

Combining the bounds for I and II yields with probability at least $1 - \delta$

$$
\inf_{\mu \in \mathbb{B}_r(\mu^*)} \frac{\langle \nabla_v L_n(\mu, v) - \nabla_v L_n(\mu, \sigma), v - \sigma \rangle}{|v - \sigma|^2}
$$

$$
\geq \frac{n^{3/2}}{z^3 (\tau \vee \tau_\sigma)^3} \left\{ \epsilon \left( \mathbb{E} \left[ \varepsilon^2 1 \left( \varepsilon^2 \leq \frac{c_\epsilon \tau_\varpi^2}{4} \right) \right] - \frac{8r\sigma^2}{c_\epsilon \tau_\varpi^2} \right) \right.
$$

$$
\left. - \sqrt{r^2 + \sigma^2} \left( \sqrt{\frac{2\varpi^2 \log(1/\delta)}{3z^2}} + \gamma \right) - \frac{\varpi^2 \log(1/\delta)}{3\sqrt{3}z^2} - \frac{\varpi\gamma\sqrt{n}}{\sqrt{3}} \right\}
$$

$$
\geq \frac{1}{2(v \vee \sigma)^3} \mathbb{E} \left[ \varepsilon^2 1 \left( \varepsilon^2 \leq \frac{c_\epsilon \tau_\varpi^2}{4} \right) \right]
$$

where $\epsilon, \varpi, \gamma, n$ are picked such that $\epsilon = 3/4$, $\gamma = 12r$, and

$$
\epsilon \left( \mathbb{E} \left[ \varepsilon^2 1 \left( \varepsilon^2 \leq \frac{c_\epsilon \tau_\varpi^2}{4} \right) \right] - \frac{8r\sigma^2 z^2}{c_\epsilon \varpi^2 n} \right) - \sqrt{r^2 + \sigma^2} \left( \sqrt{\frac{2\varpi^2 \log(1/\delta)}{3z^2}} + \gamma \right) - \frac{\varpi^2 \log(1/\delta)}{3\sqrt{3}z^2} - \frac{\varpi\gamma\sqrt{n}}{\sqrt{3}}
$$

$$
\geq \frac{1}{2} \mathbb{E} \left[ \varepsilon^2 1 \left( \varepsilon^2 \leq \frac{c_\epsilon \tau_\varpi^2}{4} \right) \right] \geq \frac{1}{4}\sigma.
$$

For example, we can pick $\varpi$ such that

$$
\max\{\varpi r \sqrt{n}, \varpi\} \to 0 \text{ and } \varpi\sqrt{n} \to \infty
$$

as $n \to \infty$. This completes the proof.

$\square$

## H.5 Supporting lemmas

This subsection proves a supporting lemma that is used prove Lemma H.4.

**Lemma H.5.** Let $w_i$ be i.i.d. copies of $w$. For any $0 < \delta < 1$, we have

$$
\frac{1}{n} \sum_{i=1}^n \frac{\tau^3 w_i^2}{(\tau^2 + w_i^2)^{3/2}} - \mathbb{E} \frac{\tau^3 w_i^2}{(\tau^2 + w_i^2)^{3/2}} \geq -\sqrt{\frac{2\tau^2 \mathbb{E} w_i^2 \log(1/\delta)}{3n}} - \frac{\tau^2 \log(1/\delta)}{3\sqrt{3}n}, \text{ with prob. } 1 - \delta,
$$

$$
\left| \frac{1}{n} \sum_{i=1}^n \frac{\tau^3 w_i^2}{(\tau^2 + w_i^2)^{3/2}} - \mathbb{E} \frac{\tau^3 w_i^2}{(\tau^2 + w_i^2)^{3/2}} \right| \leq \sqrt{\frac{2\tau^2 \mathbb{E} w_i^2 \log(1/\delta)}{3n}} + \frac{\tau^2 \log(1/\delta)}{3\sqrt{3}n}, \text{ with prob. } 1 - 2\delta.
$$

*Proof of Lemma H.5.* We only prove the first result and the second result follows similarly. The random variables $Z_i = Z_i(\tau) := \tau^3 w_i^2/(\tau^2 + w_i^2)^{3/2}$ with $\mu_z = \mathbb{E} Z_i$ and $\sigma_z^2 = \text{var}(Z_i)$ are bounded i.i.d. random variables such that

$$
0 \leq Z_i = \tau^3 w_i^2/(\tau^2 + w_i^2)^{3/2} \leq w_i^2 \wedge \frac{\tau^2}{\sqrt{3}} \wedge \frac{\tau|w_i|}{\sqrt{3}}.
$$

Moreover we have

$$
\mathbb{E} Z_i^2 = \mathbb{E} \left( \frac{\tau^6 w_i^4}{(\tau^2 + \varepsilon_i^2)^3} \right) \leq \frac{\tau^2 \mathbb{E} w_i^2}{3}, \quad \sigma_z^2 := \text{var}(Z_i) \leq \frac{\tau^2 \mathbb{E} w_i^2}{3}.
$$

For third and higher order absolute moments, we have

$$
\mathbb{E}|Z_i|^k = \mathbb{E} \left| \frac{\tau^3 w_i^2}{(\tau^2 + \varepsilon_i^2)^{3/2}} \right|^k \leq \frac{\tau^2 \mathbb{E} w_i^2}{3} \cdot \left( \frac{\tau^2}{\sqrt{3}} \right)^{k-2} \leq \frac{k!}{2} \cdot \frac{\tau^2 \mathbb{E} w_i^2}{3} \cdot \left( \frac{\tau^2}{3\sqrt{3}} \right)^{k-2}, \text{ for all integers } k \geq 3.
$$

Therefore, using Lemma J.2 with $v = n\tau^2 \mathbb{E}w_i^2/3$ and $c = \tau^2/(3\sqrt{3})$ acquires that for any $t \geq 0$

$$\mathbb{P}\left(\sum_{i=1}^n \frac{\tau^3 w_i^2}{(\tau^2 + \varepsilon_i^2)^{3/2}} - \sum_{i=1}^n \mathbb{E}\left(\frac{\tau^3 w_i^2}{(\tau^2 + \varepsilon_i^2)^{3/2}}\right) \geq -\sqrt{\frac{2n\tau^2\mathbb{E}w_i^2 t}{3}} - \frac{\tau^2 t}{3\sqrt{3}}\right) \leq \exp(-t).$$

Taking $t = \log(1/\delta)$ acquires that for any $0 < \delta < 1$

$$\mathbb{P}\left(\frac{1}{n}\sum_{i=1}^n \frac{\tau^3 w_i^2}{(\tau^2 + w_i^2)^{3/2}} - \frac{1}{n}\sum_{i=1}^n \mathbb{E}\left(\frac{\tau^3 w_i^2}{(\tau^2 + \varepsilon_i^2)^{3/2}}\right) > -\sqrt{\frac{2\tau^2\mathbb{E}w_i^2 \log(1/\delta)}{3n}} - \frac{\tau^2 \log(1/\delta)}{3\sqrt{3}n}\right) > 1 - \delta.$$

This finishes the proof. $\qquad\square$

# I   Proofs for Section 6

We first prove Proposition 6.1.

*Proof of Proposition 6.1.* The proof directly follows from Theorem 3.5 and the union bound. $\qquad\square$

Next, we prove Proposition 6.2.

*Proof of Proposition 6.2.* We only sketch the proof, as most of the proof follows from that of Theorem H.2. By Proposition 6.1 and taking $z^2 = 2\log n$, we obtain

$$\|\widehat{\mu} - \mu^*\|_2 \to 0 \ \text{ in probability.}$$

Similarly, following the proof of Theorem H.3, we obtain

$$\|\widehat{v} - \sigma\|_2 \to 0 \ \text{ in probability,}$$

where $\widehat{v} = (\widehat{v}_1, \ldots, \widehat{v}_d)^{\mathsf{T}}$ and $\sigma = (\sigma_{11}, \ldots, \sigma_{dd})^{\mathsf{T}}$.

With a slight overload of notation, let $L_n(\mu) = L_n(\mu, \sigma)$. Let $\tau_{\sigma_k} = \sigma_{kk}\sqrt{n}/z$. Then following the proof of Theorem H.2, we obtain

$$\sqrt{n}\left(\widehat{\mu} - \mu^*\right) \simeq [H_n(\mu^*)]^{-1}\left(-\sqrt{n}\,\nabla L_n(\mu^*)\right)$$

$$= \begin{bmatrix} \frac{\sqrt{n}}{z} \cdot \frac{1}{n}\sum_{i=1}^n \frac{\tau_{\sigma_{11}}^2}{(\tau_{\sigma_{11}}^2+\varepsilon_{i1}^2)^{3/2}} & 0 & \cdots \\ & \vdots & \ddots \\ 0 & & \frac{\sqrt{n}}{z} \cdot \frac{1}{n}\sum_{i=1}^n \frac{\tau_{\sigma_{dd}}^2}{(\tau_{\sigma_{dd}}^2+\varepsilon_{id}^2)^{3/2}} \end{bmatrix}^{-1} \begin{bmatrix} \sqrt{n} \cdot \frac{1}{n}\sum_{i=1}^n \frac{\tau_{\sigma_{11}}\varepsilon_{i1}}{\sigma_{11}\sqrt{\tau_{\sigma_{11}}^2+\varepsilon_{i1}^2}} \\ \vdots \\ \sqrt{n} \cdot \frac{1}{n}\sum_{i=1}^n \frac{\tau_{\sigma_{dd}}\varepsilon_{id}}{\sigma_{dd}\sqrt{\tau_{\sigma_{dd}}^2+\varepsilon_{id}^2}} \end{bmatrix}$$

$$\simeq \begin{bmatrix} \sigma_{11} & 0 & \cdots \\ \vdots & \ddots & \\ 0 & & \sigma_{dd} \end{bmatrix} \begin{bmatrix} \sqrt{n} \cdot \frac{1}{n}\sum_{i=1}^n \frac{\tau_{\sigma_{11}}\varepsilon_{i1}}{\sigma_{11}\sqrt{\tau_{\sigma_{11}}^2+\varepsilon_{i1}^2}} \\ \vdots \\ \sqrt{n} \cdot \frac{1}{n}\sum_{i=1}^n \frac{\tau_{\sigma_{dd}}\varepsilon_{id}}{\sigma_{dd}\sqrt{\tau_{\sigma_{dd}}^2+\varepsilon_{id}^2}} \end{bmatrix}$$

$$=: \Lambda\,\mathrm{I},$$

where $\Lambda = \mathrm{diag}(\sigma_{11}, \ldots, \sigma_{dd})$.

We only to derive the asymptotic distribution of the term I:

$$\mathrm{I} = \sqrt{n} \cdot \left(\begin{bmatrix} \frac{1}{n}\sum_{i=1}^n \frac{\tau_{\sigma_{11}}^2\varepsilon_{i1}}{\sigma\sqrt{\tau_{\sigma_{11}}^2+\varepsilon_{i1}^2}} \\ \vdots \\ \frac{1}{n}\sum_{i=1}^n \frac{\tau_{\sigma_{dd}}^2\varepsilon_{id}}{\sigma\sqrt{\tau_{\sigma_{dd}}^2+\varepsilon_{id}^2}} \end{bmatrix} - \mathbb{E}\begin{bmatrix} \frac{1}{n}\sum_{i=1}^n \frac{\tau_{\sigma_{11}}\varepsilon_{i1}}{\sigma_{11}\sqrt{\tau_{\sigma_{11}}^2+\varepsilon_{i1}^2}} \\ \vdots \\ \frac{1}{n}\sum_{i=1}^n \frac{\tau_{\sigma_{dd}}\varepsilon_{id}}{\sigma_{dd}\sqrt{\tau_{\sigma_{dd}}^2+\varepsilon_{id}^2}} \end{bmatrix}\right) + \sqrt{n} \cdot \mathbb{E}\begin{bmatrix} \frac{1}{n}\sum_{i=1}^n \frac{\tau_{\sigma_{11}}\varepsilon_{i1}}{\sigma_{11}\sqrt{\tau_{\sigma_{11}}^2+\varepsilon_{i1}^2}} \\ \vdots \\ \frac{1}{n}\sum_{i=1}^n \frac{\tau_{\sigma_{dd}}\varepsilon_{id}}{\sigma_{dd}\sqrt{\tau_{\sigma_{dd}}^2+\varepsilon_{id}^2}} \end{bmatrix}$$

$$= \mathrm{I}_1 + \mathrm{I}_2.$$

Again, following the proof of Theorem H.2, the $\ell_2$ norm of the second term goes to 0. For the first term I, we have

$$
\mathrm{I}_1 \rightsquigarrow \mathcal{N}\left(0, \lim_{n\to\infty} \mathrm{cov}\left(\begin{bmatrix} \frac{\tau^2_{\sigma_{11}}\varepsilon_{i1}}{\sigma\sqrt{\tau^2_{\sigma_{11}}+\varepsilon^2_{i1}}} \\ \vdots \\ \frac{\tau^2_{\sigma_{dd}}\varepsilon_{id}}{\sigma\sqrt{\tau^2_{\sigma_{dd}}+\varepsilon^2_{id}}} \end{bmatrix}\right)\right)
$$
$$
= \mathcal{N}\left(0, \Lambda^{-1}\Sigma\Lambda^{-1}\right).
$$

Thus we have

$$
\sqrt{n}\left(\widehat{\mu}-\mu^*\right) \rightsquigarrow \mathcal{N}(0,\Sigma).
$$

This finishes the proof.

$\square$

## J  Preliminary lemmas

This section collects preliminary lemmas that are frequently used in the proofs for the main results and supporting lemmas. We first collect the Hoeffding's inequality and then present a form of Bernstein's inequality. We omit their proofs and refer interested readers to Boucheron et al. (2013).

**Lemma J.1** (Hoeffding's inequality). Let $Z_1,\ldots,Z_n$ be independent real-valued random variables such that $a \leq Z_i \leq b$ almost surely. Let $S_n = \sum_{i=1}^n (Z_i - \mathbb{E}Z_i)$ and $v = n(b-a)^2$. Then for all $t \geq 0$,

$$
\mathbb{P}\left(S_n \geq \sqrt{vt/2}\right) \leq e^{-t},\ \mathbb{P}\left(S_n \leq -\sqrt{vt/2}\right) \leq e^{-t},\ \mathbb{P}\left(|S_n| \geq \sqrt{vt/2}\right) \leq 2e^{-t}.
$$

**Lemma J.2** (Bernstein's inequality). Let $Z_1,\ldots,Z_n$ be independent real-valued random variables such that

$$
\sum_{i=1}^n \mathbb{E}Z_i^2 \leq v,\ \sum_{i=1}^n \mathbb{E}|Z_i|^k \leq \frac{k!}{2}vc^{k-2}\text{ for all } k \geq 3.
$$

If $S_n = \sum_{i=1}^n (Z_i - \mathbb{E}Z_i)$, then for all $t \geq 0$,

$$
\mathbb{P}\left(S_n \geq \sqrt{2vt}+ct\right) \leq e^{-t},\ \mathbb{P}\left(S_n \leq -(\sqrt{2vt}+ct)\right) \leq e^{-t},\ \mathbb{P}\left(|S_n| \geq \sqrt{2vt}+ct\right) \leq 2e^{-t}.
$$

*Proof of Lemma J.2.* This lemma involves a two-sided extension of Theorem 2.10 by Boucheron et al. (2013). The proof follows from a similar argument used in the proof of Theorem 2.10, and thus is omitted. $\square$

Our third lemma concerns the localized Bregman divergence for convex functions. It was first established in Fan et al. (2018). For any loss function $L$, define the Bregman divergence and the symmetric Bregman divergence as

$$
D_L(\beta_1,\beta_2) = L(\beta_1) - L(\beta_2) - \langle \nabla L(\beta_2), \beta_1 - \beta_2 \rangle,
$$
$$
D_L^s(\beta_1,\beta_2) = D_L(\beta_1,\beta_2) + D_L(\beta_2,\beta_1).
$$

**Lemma J.3.** For any $\beta_\eta = \beta^* + \eta(\beta - \beta^*)$ with $\eta \in (0,1]$ and any convex loss function $L$, we have

$$
D_L^s(\beta_\eta,\beta^*) \leq \eta D_L^s(\beta,\beta^*).
$$

Our forth lemma in this section concerns three basic inequalities that are frequently used in the proofs.

**Lemma J.4.** The following inequalities hold:

(i) $(1+x)^r \geq 1 + rx$ for $x \geq -1$ and $r \in \mathbb{R} \setminus (0,1)$;

(ii) $(1+x)^r \leq 1 + rx$ for $x \geq -1$ and $r \in (0,1)$;

(iii) $(1+x)^r \leq 1 + (2^r - 1)x$ for $x \in [0,1]$ and $r \in \mathbb{R} \setminus (0,1)$.

