# OpenReview forum: "Do we need to estimate the variance in robust mean estimation?"
_TMLR — Rejected by TMLR_

### Review · Reviewer_xmsE · 2023-09-12

**Summary Of Contributions:**

This paper proposes an algorithm for robust mean estimation. The proposed method can automatically estimate the variance, and theoretical justification on its advantage over median of means (MoM) is provided. Numerical results also demonstrate its advantage.

**Audience:**

No

**Claims And Evidence:**

Yes

**Requested Changes:**

Please provide some more results in multi-variate asymptotic distribution.

**Strengths And Weaknesses:**

Pros:

This paper provides rigorous theoretical justification on the performance of the proposed method (Theorem 4.2) and its advantage over MoM (Theorem 4.1). The mathematical statements are clear and easy to understand.

Cons:

My two major concerns are about the contribution and the topic of this paper:

*    This paper studies a robust mean estimation and provides a lot of theoretical justification. However, in my point of view, this paper is more suitable for a statistics journal rather than a ML journal for the lack of real-data analysis. Mean estimation is a very traditional statistical problem.
*    The theory main considers a univariate case and does not consider multi-variate case. A theory in a multi-variate scenario may provide more insights on the relationship between the mean estimation task performance and the covariance structure of the distribution, i.e., for the asymptotic normality results in Theorem 4.1 and 4.2, how are the asymptotic distributions related to the covariance structure?

Besides the two main concerns, the following are some minor comments in writing:
*    I would suggest mentioning "heavy-tailed distribution" in either the abstract or the title.
*    The first paragraph of the paper is a little bit misleading. It mentions about some applications in high-dimensional statistics. However, starting from the second paragraph, it talks about a mean estimation problem for a univariate random variable.
*    In the first sentence of the abstract, "...self-tuned robust estimators for estimating the mean of distributions with only finite variances", does "with only finite variances" refer to the robust estimator, or refer to the distribution?
*    In "the resulting estimator for the robustification parameter can adapt to the unknown variance automatically", what does the "variance" refer to?

---

> ### Author Response · Authors · 2023-10-03
> **Response to reviewer xmsE**
>
> We would like to thank this reviewer for his comments which have helped the authors to dig deeper about the subject. We address the comments in a one-to-one manner below.
>
> **Major comments:**
>
> >
> > 1. However, in my point of view, this paper is more suitable for a statistics journal rather than a ML journal for the lack of real-data analysis. Mean estimation is a very traditional statistical problem.
> >
>
> As also pointed out by other reviewers, we argue that the mean estimation problem of skewed or heavy tailed distributions is fundamental and of wide interest to statistics as well as machine learning community. Thus, an efficient method that is not overly complicated could have a broad impact. Our method also serves as the foundation for tackling more general problems.
>
> >
> > 2. The theory main considers a univariate case and does not consider multi-variate case. A theory in a multi-variate scenario may provide more insights on the relationship between the mean estimation task performance and the covariance structure of the distribution.
> >
>
> Following the suggestion of reviewer 5VdC, we have applied our estimator coordinate-wise to obtain a multivariate estimator and the corresponding upper bound for the multivariate case. See the discussion section for details (Proposition 6.1). The result follows from Theorem 3.5 and the union bound. Specifically the upper bound is at the order of $\sqrt{\mathrm{tr}(\Sigma)\log(nd/\delta)/n}$, which is only optimal up to logarithmic factors.
>
> We emphasize that extending the robust estimator to the multivariate case with optimal statistical guarantees (without the logarithmic factor in the leading term) is generally a challenging problem. For example, extending the MoM estimator to the multivariate case has motivated many works; see Section 3 by Lugosi and Mendelson (2019a) for a recent review. The difficulty is due to the fact that there is no standard notion of a median for multivariate data (Small, 1990).  Most of them are computationally expensive or even NP-hard. It is challenging to develop their asymptotic properties. For example, due to combinatoric nature of the median-of-means tournament (Lugosi and Mendelson, 2019b), it is very hard to write it in an $M$ estimation framework. We will leave this to future work.
>
> **Minor comments:**
>
> >
> > 1&2: I would suggest mentioning "heavy-tailed distribution" in either the abstract or the title. In the first sentence of the abstract, "...self-tuned robust estimators for estimating the mean of distributions with only finite variances", does "with only finite variances" refer to the robust estimator, or refer to the distribution?
> >
>
> Thanks for this comment. We have rewritten the first sentence of the abstract to  ``In this paper, we propose self-tuned robust estimators for estimating the mean of heavy-tailed distributions, where heavy-tailed distributions refer to distributions with only finite variances."
>
> >
> > 3. In "the resulting estimator for the robustification parameter can adapt to the unknown variance automatically", what does the "variance" refer to?
> >
>
> The variance referes to the unknown data variance. We have rewritten this sentence to “……can automatically adapt to the unknown data variance and can achieve near-optimal finite-sample performance.”
>
> **Reference:**
>
>  Lugosi and Mendelson (2019a), Mean estimation and regression under heavy-tailed distributions — A survey.
>
> Lugosi and Mendelson (2019b), Sub-Gaussian estimators of the mean of a random vector.
>
> Small (1990), A Survey of Multidimensional Medians.

---

> > ### Comment · Reviewer_xmsE · 2023-10-04
> >
> > Thanks for the authors replying to my comments.
> >
> > For the new Proposition 6.1, I understand that it is an in-probability bound so the log term is necessary, but I'm wondering whether it is possible to derive the asymptotic distribution in a format similar to Theorem 4.1 and 4.2? If so, what is the asymptotic variance?
> >
> > For my major concern 1 about whether this paper is suitable for ML journal or not, I would leave this up to AC for the decision.
> >
> > The overall quality of this paper looks good to me. Since (1) existing minimax lower bound analysis mainly focuses on the convergence rate rather than a exact multiplicative constant, and (2) sample mean usually achieves minimax optimal, I think showing the asymptotic normality with a clear smaller variance is already sufficient.

---

> ### Author Response · Authors · 2023-10-05
> **Response to reviewer xmsE**
>
> We have further proved the asymptotic result, Proposition 6.2, for the proposed multivariate estimator; see our revised manuscript. To our surprise (at least at first glance), it also achieves full asymptotic efficiency. Specifically, under certain conditions, we have:
> $$
> \hat\mu(\hat v) - \mu^* \rightsquigarrow \mathcal{N}(0, \Sigma).
> $$
>
> In hindsight, the above result is natural, as the simple coordinate-wise multivariate mean estimator also approaches to the sample mean estimator in the asymptotic limit, and thus is asymptotically efficient. This highlights another advantage of our procedure: It can easily be extended and yet still achieves full asymptotic efficiency.

---

> > ### Comment · Reviewer_xmsE · 2023-10-05
> >
> > Thanks for your response and the additional result.

---

> > > ### Author Response · Authors · 2023-10-08
> > > **Thanks**
> > >
> > > Thanks very much for your review which has helped to improve the paper significantly! If you have any other questions/confusion, please let us know.

---

### Review · Reviewer_5VdC · 2023-09-15

**Summary Of Contributions:**

This paper addresses the question of computationally efficient, heavy-tailed, one-dimensional mean estimation for distributions with finite but unknown covariance. A new estimator is proposed for this task, which adapts to the unknown variance and achieves near optimal finite-sample performance. Their estimator is shown to be asymptotically efficient (achieving the CR lower bound), in comparison to existing approaches for the problem, and can be computed efficiently. Numerical experiments demonstrating the superiority of the proposed approach compared to other methods are also provided.

The idea behind the construction is the following: the Huber loss can be used for this task with known variance (i.e., penalize mean estimate by expected squared loss for small loss values and mean absolute deviation for large loss values). Further, it is possible to smooth/convexify the Huber loss, which yields the pseudo-Huber loss. Both depend on some proxy for the variance, which is unknown. The authors then add in an extra optimization parameter for this variance proxy and augment the pseudo-Huber loss so that the expected risk is minimized when this parameter is a decent proxy of the true variance.

**Audience:**

Yes

**Broader Impact Concerns:**

No concerns.

**Claims And Evidence:**

Yes

**Requested Changes:**

Beyond addressing the weaknesses above, here are a few more secondary comments:

- I suggest setting $\alpha=0.5$ from the get go and perhaps comment on why this is the right choice. The current approach (which leaves it as a free parameter and then arrives at this choice at the end of the paragraph following Theorem 2.3) does not contribute to clarity nor generality.

- In regard to Theorems. 4.2-4.3, do the authors have intuition about what is the distinctive characteristic of the proposed approach that enables its asymptotic efficiency, compared to the MoM estimator? Adding such a discussion would be a welcome addition. Another small point: I suggest being consistent with how $\iota\in(0,1]$ (vs. $0<\iota\leq 1$) is written in those theorems.

- $v_0$ should be fixed at the start of Lemma 3.2. Currently, it appears out of nowhere.

- In Eq. (3.1), $\hat{v}$ should be just $v$.

- Can the authors provide an interpretation for the variance ration assumption from Theorem 3.4? Perhaps identifying primitive sufficient conditions or providing an example that verifies it would be helpful.

**Strengths And Weaknesses:**

Strengths:

- The self-tuning estimator and the fact that it overcomes the unknown variance issue is interesting to me, and I wonder if this can be characterized as a special case of a more general approach. The idea is novel (I, at least, have not seen this approach before) and constitutes a good contribution.

- The paper is well written. The technical writing is solid, the exposition/motivation/outlook are to a good level as well. I enjoyed reading it.

- The efficiency result provides a nice theoretical distinction between the proposed estimator and the MoM method. The argument is reminiscent of that for the classical sample mean vs. median estimator for the expectation (it seems appropriate to mention this and provide a brief discussion). This result provides a theoretical explanation of the improved performance observed empirically.

- Speaking of the experiments, the practical performance of the estimator is solid, beating others consistently (though see second weakness, not clear that they couldn’t be improved)

Weaknesses:

-The conditions for the finite-sample guarantees from Theorems 3.4 and 3.5 seem quite subtle and are hard to interpret. Could these conditions be relaxed to more natural and primitive assumptions?

- The MoM estimator is dead simple, more computationally efficient, and solves the formulated problem. The authors admit this but emphasize the asymptotic performance as a key benefit of their approach. However, the guarantees of the self-tuning approach depend on some subtle assumptions, and the finite-sample performance depends on unknown constants. This means that the sample complexity cannot be computed in advance for fixed error. This should be mentioned in the limitation section.

- A glaring limitation of the approach is the fact that it only treats the one-dimensional case. I find it odd that the authors do not even comment on the extension to $d>1$. For example, they can apply the current method coordinate-wise to obtain an L^2 error of $\sqrt{d}$ times the individual error, perhaps with some $\log d$ factor for a union bound. I encourage the authors to examine this case and, if possible, provide an analysis. The attained risk may not be optimal but still worth mentioning. Either way, the limitations bit in the Conclusion section should discuss the extension to higher dimensions and the roadblocks towards it.

- It is not clear that the MoM and trimmed mean estimators have been optimized for asymptotic efficiency. I wonder if the constant of 8 which appears in the MoM case could be reduced for large n in a way that would make it more efficient. This idea is implemented in the proposed approach and leads to its superiority. It seems appropriate to try check where existing approaches can be adapted or prove a formal limitation result.

- My sense is that this approach is unlikely to scale to high-dimensions (in general, M-estimation like Huber loss usually does not work for high-dimensional robust statistics). Did the authors try looking at the $d>1$ case. The authors motivate their work by mentioning `high-dimensional data analysis' in the first sentence of the intro. It would be appropriate to the multivariate case or at least adequately discuss it and explain the roadblocks towards the extension. This too should be added to the limitations section.


Overall:

Despite the above weaknesses, the paper is interesting and appears correct. If the authors could get finite-sample performance guarantees with more interpretable assumptions, and/or argue convincingly that MoM or similar approaches could not be easily improved, I think it will qualify for acceptance.

---

> ### Author Response · Authors · 2023-10-03
> **Response to reviewer 5VdC, part 1**
>
> We would like to thank this reviewer for many helpful comments which have helped improved the presentation of the paper. We address the comments in a one-to-one manner below.
>
> **Major comments:**
>
> >
> > Major comment 1 and minor comment 5. The conditions for the finite-sample guarantees from Theorems 3.4 and 3.5 seem quite subtle and are hard to interpret. Could these conditions be relaxed to more natural and primitive assumptions?
> >
>
> Major comment 1 and minor comment 5 are similar, and we address them here. Thanks for pointing this out. We have restated the theorem. Specifically, we have removed this variance ratio condition. Instead, the lower bound in Theorem 3.4 becomes
> $$
> \hat v \geq c_0 \sigma_{\tau_{v_0}^2 -\epsilon_0} \geq c_0 \sigma_{\tau_{v_0}^2 -1}.
> $$
> See our revised manuscript for details. Theorem 3.5 has been revised accordingly.
>
> >
> > 2. However, the guarantees of the self-tuning approach depend on some subtle assumptions, and the finite-sample performance depends on unknown constants. This means that the sample complexity cannot be computed in advance for fixed error. This should be mentioned in the limitation section.
> >
>
> Thank you for this suggestion.  We have addressed in our limitation section.
>
> >
> > 3&5.  The multi-dimensional case $d>1$.
> >
>
> The 3&5 major comments are on the study of multi-dimensional case $d>1$, so we address both of them here in this thread. Following the suggestion of this reviewer, we have applied our estimator coordinate-wise and obtained a result for the multivariate case. See the discussion section for details.
>
> >
> > 4. It is not clear that the MoM and trimmed mean estimators have been optimized for asymptotic efficiency. I wonder if the constant of 8 which appears in the MoM case could be reduced for large n in a way that would make it more efficient. This idea is implemented in the proposed approach and leads to its superiority. It seems appropriate to try check where existing approaches can be adapted or prove a formal limitation result.
> >
>
> We extremely thank this reviewer for asking this inspiring question! This question makes us to deep dive the working mechanisms of our estimator and the MoM estimator, as stated below.
>
> We start with explaining intuitively why our self-tuned estimator can achieve (near) optimal performance in both the finite-sample regime and the asymptotic regime. Because the self-tuned estimator in (3.1) is a self-tuned version of the pseudo-Huber estimator in (2.2), we focus on the pseudo-Huber estimator $\hat\mu(\tau)$. Theorem 2.1 suggests that taking $\tau = \sigma \sqrt{n/\log(1/\delta)}$ guarantees the sub-Gaussian performance of $\hat \mu (\tau)$ for finite samples. Meanwhile, as $n\rightarrow \infty$, we have $\tau= \sigma \sqrt{n/\log(1/\delta)} \rightarrow \infty$. Thus the pseudo-Huber loss approaches to the least square loss which corresponds to the negative log maximum likelihood of Gaussian distributions, which leads to the asymptotically efficient mean estimator.
>
> For MoM estimators, the situation differs. On one hand, to attain robustness in the finite-sample regime, the number of blocks $k$ should be greater than or equal to $\lceil 8 \log(1/\delta)\rceil$, as demonstrated in the proof of Theorem 4.1 by Lugosi and Mendelson (2019). On the other hand, to approach the sample mean estimator and achieve asymptotic efficiency in the large sample limit, the number of blocks should diminish to 1 as the sample size $n$ grows. Consequently, optimal finite-sample and asymptotic properties represent two  contrasting characteristics for MoM estimators. In other words, the MoM estimator can not simultaneously adapt to both regimes. This contrast seems to arise from the discontinuous nature of the MoM estimator which cannot smoothly transition from requiring at least $k=3$ blocks (for defining the median) to functioning as an empirical mean estimator.
>
> **References**:
>
> Lugosi and Mendelson (2019), Mean estimation and regression under heavy-tailed distributions — A survey.

---

> ### Author Response · Authors · 2023-10-03
> **Response part 2**
>
> **Minor comments:**
>
> >
> > 1. I suggest setting $\alpha =0.5$ from the get go and perhaps comment on why this is the right choice.
> >
>
> Thanks for pointing this out. We thought about this. But with that change, it is hard to discuss the optimality of $a=1/2$ and it is even harder to connect to Huber’s concomitant estimator discussed in the end of Section 2. After some reflection, we prefer to keep as it is in Section 2. For all discussions after Section 2, we set $a=1/2$.
>
> >
> > 2. In regard to Theorems. 4.2-4.3, do the authors have intuition about what is the distinctive characteristic of the proposed approach that enables its asymptotic efficiency, compared to the MoM estimator? Adding such a discussion would be a welcome addition. Another small point: I suggest being consistent with how $\iota \in (0,1]$ (vs. $0<\iota\leq1$) is written in those theorems.
> >
>
> For the intuition, please see our comment to major comment 4. We have also changed all $\iota \in (0,1]$ to  $0<\iota\leq1$ to make it consistent.
>
> >
> > 3.$v_0$ should be fixed at the start of Lemma 3.2.
> >
>
> Thank you, we have fixed this.
>
> >
> > 4. In Eq. (3.1), $\hat v$ should be just $v$.
> >
>
> Thank you, we have fixed this.
>
> >
> > minor comment 5
> >
>
> This is addressed in response to major comment 1.

---

> ### Comment · Reviewer_5VdC · 2023-10-10
>
> Thank you for the response to the above points my comments and questions. Below, please find some additional thoughts:
>
> * In the authors' repones to my original "Major comment 1 and minor comment 5", I don't see how this modification addresses the original concern. Perhaps the authors can provide more context?
>
> * I like the explanation for the asymptotic efficiency and believe that including this discussion has improved clarity and interpretability of the results. Thank you for adding that.
>
> * The sample complexity results leave much to be desired. Having bounds that *explicitly* depend on the parameters of the problem would significantly strengthen them. Perhaps the c_0 and C_0 constants can be bounded within certain ranges? Is it clear that such constants exist for any parameter values? I don't view a full account of the above as mandatory for acceptance; in my opinion, the paper has merit and presents innovative ideas even with this limitation. Still, if more can be said, that would be welcome. (unrelated: I also suggest changing the phrasing in Thms. 3.4 and 3.5 from "Let c0 and C0 be some constants, and suppose ..." to "Suppose that c0 and C0 are constants with [or: such that]...". )
>
> * I am pleased with the remaining responses. Thank you.

---

> ### Author Response · Authors · 2023-10-10
> **Further response to reviewer 5VdC**
>
> Thanks for your positive responses and follow-up questions! We address two of your questions below. We start with comment 3.
>
> >
> > 3. The sample complexity results leave much to be desired. Having bounds that *explicitly* depend on the parameters of the problem would significantly strengthen them. Perhaps the c_0 and C_0 constants can be bounded within certain ranges?
> >
>
> This is a good suggestion. $c_0$ and $C_0$ here have simple dependencies on problem-dependent quantities. But there are also other constants that depends on other constants and problem-dependent quantities in a complicated way. For readability concerns, we hide all the dependencies. In hindsight, we agree that having bounds that explicitly depend on the parameters might be useful. To balance the readability and explicitness of the results, in the follow-up revision, we plan to provide results with explicit constants in the appendix. We will also revise the phrasing in the results as you suggested.
>
> >
> > 1. In the authors' repones to my original "Major comment 1 and minor comment 5", I don't see how this modification addresses the original concern. Perhaps the authors can provide more context?
> >
>
> We apologize if we misunderstood your comment.  In our original submission, our statement of Theorem 3.4 reads as “Assume that $n$ is sufficiently large and there exists an $\epsilon_0= \tilde O(1/n)$ such that $\sigma_{\tau_{v_0}^2/2−\epsilon_0}/\sigma_{\tau_{v_0}^2/2} ≥ 1/3$….”
>
> We agree that requiring this variance ratio assumption is pretty strong especially with the constant $1/3$ there. We modified the proof, and removed this assumption. But now the lower bound depends on $\sigma_{\tau_{v_0}^2/2−1}/\sigma_{\tau_{v_0}^2/2} $. We clarify that this is not a very strong assumption as $\tau_{v_0}^2\geq \tau_{v_0}^2/2-1 \rightarrow \infty$, and thus, by the dominated convergence theorem,  we have both $\sigma_{\tau_{v_0}^2/2−1}$ and $\sigma_{\tau_{v_0}^2/2}$ approach to $\sigma$, and thus their ratio approaches to $1$.
>
> We give an example. Suppose the third absolute moment exists. Then
> \begin{align*}
> \frac{\sigma^2_{\tau_{v_0}^2/2−1}}{\sigma^2_{\tau_{v_0}^2/2}}
> &= 1  -  \frac{\sigma^2_{\tau_{v_0}^2/2−1}}{\sigma^2_{\tau_{v_0}^2/2}}
> \geq 1 -\frac{\mathbb{E} |\epsilon|^3}{\sigma^2_{\tau^2_{v_0}/2}} \times \frac{1}{\sqrt{\tau_{v_0}^2/2-1}}
> = 1- O\left(\frac{1}{\sqrt{n}}\right).
> \end{align*}
> We will follow up with a revision that has these added.

---

> > ### Comment · Reviewer_5VdC · 2023-10-11
> >
> > Thank you. The planned revisions sound good to me.

---

> > > ### Author Response · Authors · 2023-10-12
> > > **Thank you**
> > >
> > > Thank you very much for your review! We believe your comments have helped to improve the paper significantly! If you have any other questions/confusion, please let us know.

---

### Review · Reviewer_p9aB · 2023-09-25

**Summary Of Contributions:**

This work studies the problem of how to estimate the mean of a distribution with finite variance given i.i.d. samples and proposes a self-tuned estimator which does not require cross-validation or Lepski's method to tune hyper-parameter. The estimator is based on jointly minimizing a newly defined penalized pseudo-Huber loss function over both the estimation variable $\mu$ and robustness variable $\nu$. For finite-sample theory, an estimation error bound in the order of $O(\sqrt{\log(n)/n})$ is established, where $n$ is the number of samples.To compare with the existing estimators median-of-means (MoM) and trimmed mean estimator, the asymptotic efficiency of the proposed estimator is studied. In numerical experiments, the proposed method achieves lower estimation errors on datasets generated from skewed generalized t distributions, compared with sample mean, MoM, trimmed mean, cross validation, and Lepski's method, and it is also more computationally efficient than the last two methods.

**Audience:**

Yes

**Broader Impact Concerns:**

I have no concern on the ethical implications of the work.

**Claims And Evidence:**

Yes

**Requested Changes:**

The points in the Weaknesses/questions are my suggested adjustments. In my view, (1) (3) (4) are critical, while the others would strengthen the work.

Besides, there are some minor issues.
- Should the $\hat{\mu}(\tau)$ in (2.2) be $\tilde{\mu}(\tau)$?
- Is the $\hat{\mu}(\tau)$ in Theorem 2.1 the solution of the pseudo-Huber loss or the penalized pseudo-Huber loss?
- In the last paragraph of section 2, what is the loss function $\ell$? Should it be $\rho$ instead?

**Strengths And Weaknesses:**

Strengths

(1) The problem of mean estimation of skewed or heavy tailed distributions is fundamental and of wide interest. Thus, an efficient method that is not overly complicated could have a broad impact.

(2) The main idea of the proposed self-tuned estimator, jointly minimizing over estimated mean $\mu$ and robustness parameter $\nu$, is clearly explained. Theorem 2.3 further justifies minimization over $\nu$ by proving that, given ground truth $\mu_{\star}$, the optimal $\nu_{\star}$ can converge to the distribution variance $\sigma$ as $n$ tends to infinity.

(3) Conclusions on finite-sample estimation error bound and asymptotic error bound provide theoretical guarantees on the performance of the estimators. These theoretical results also justify the advantage of the proposed estimator over existing methods.

Weaknesses/questions

(1) The motivation of defining the penalized pseudo-Huber function is to avoid trivial solutions $0$ and $\infty$. Then why is the constraint $\nu_0 \leq \nu \leq V_0$ still needed to guard $\hat{\nu}$ from $0$ and $\infty$ in (3.1)? Since problem (3.1) is the actual optimization problem to be solved, why not just use the pseudo-Huber function? How do we choose $\nu_0$ and $V_0$ in practice? Is cross-validation still needed?

(2) The theorems can be stated more clearly.
- In Theorem 3.1, both $z$ and $r$ depend on $\delta$, but after the definitions of $z$ and $r$ it says that for any $\delta$ the error bound holds with probably $1-\delta$. Are they the same $\delta$? If so, it'd be better to put the phrase ``for any $\delta$'' in the front, before defining $z$ and $r$. The same issue is also in Lemma 3.2.
- In Theorem 3.1, the error bound actually equals to the radius $r$. It'd be more clear to use the same notation and state $|\hat{\mu} - \mu^\star| < r$. Also, is there any intuitive explanation on why the error bound equals to the radius in the assumption?
- In Lemma 3.2 and Corollary 3.3, one assumption is $n \geq C\max(z^2(\sigma^2 + r^2)/v_0^2, \log (1/\delta))$, but $r$ also depends on $n$, so what is the requirement on $n$ for this assumption to hold? It would be better to plug $r$ into this inequality to draw a condition on $n$.

(3) In Theorem 3.4 and Theorem 3.5, one assumption is $c_0 \sigma_{v_0^2 n/z^2} \leq C_0 \sigma$. What is the requirement on $n$ for this assumption to hold? It'd be better to discuss this issue at least for some commonly encountered distributions. The key question here is whether or not this assumption imposes a more strict condition on $n$.

(4) In the numerical section, what is the method/solver actually used to solve problem (3.5)? How does this method/solver scale as the number of samples $n$ increase? This would determine the runtime efficiency of the proposed estimator.

(5) In the numerical section, is there a reason not to compare all methods in the same plot? For example, can Figure 1 and Figure 3 be merged together? It'd be better to also report the runtimes of sample mean, MoM, and trimmed mean.

---

> ### Author Response · Authors · 2023-10-03
> **Response to reviewer p9aB, part 1**
>
> We would like to thank this reviewer for many helpful comments which have helped improved the presentation of the paper. We address the comments in a one-to-one manner below.
>
>
> **Major comments**
> >
> > 1. Why is the constraint $v_0\leq v \leq V_0$ still needed? Since problem (3.1) is the actual problem to be solved, why not just use pseudo-Huber function? How do we choose $v_0$ and $V_0$ in practice? Is cross-validation still needed?
> >
>
> We first apologize for the typo in Eq. (3.1). We suspect that most of these questions are due to this typo. As pointed out by reviewer 5Vd3, the $\hat v$ in the constraint in Eq. (3.1) should be $v$.  Thus it is NOT equivalent to the pseudo-Huber loss. The $v_0$ and $V_0$ in practice are taken as small and large constants. In our experiments, we take $v_0$ and $V_0$ as $10^{-4}$ and $10^4$ respectively. The pseudo-Huber loss, when treated as a loss function for $v$, has trivial minimizers $v=0$ and $v = \infty$. The penalized pseudo-Huber avoids these two trivial minimizers, aka when starting an initialization, an algorithm such as gradient descent will not converge to these trivial minimizers.  Yet, we still need the constraint that $v_0\leq v\leq V_0$. The reason is that, to run an algorithm such as the alternative gradient descent (see Appendix C), we need to initialize $v$ at some initialization $v_{\mathrm{init}}$ bounded away from $0$ and $\infty$, because $v=0$ and $\infty$ correspond to a nonsmooth loss function (in $\mu$) and a trivial loss function respectively. And smoothness is needed for establishing the strong convexity of the loss function. Technically speaking, we need $v_0\leq  v \leq V_0$ to bound the tail probability of $\mathcal{E}_2$; see Lemma G.1 and proof of Theorem 3.4.
>
> >
> > 2.  (a). It is better to put the phrase “for any $\delta$” in the front before defining $z$ and $r$ in Theorem 3.1 and Lemma 3.2. (b). Theorem 3.1. The error bound actually equals to the radius $r$. Why is the error equal to $r$? (c). In Lemma 3.2 and Corollary 3.3, the assumption involves $r$. Can you make this explicit?
> >
>
> We address these questions as follows.
>
> (a) We have fixed this.
>
> (b). We apologize for this confusion. We have revised the statement of theorem to be “Assume Assumption 1 holds with any $r {\geq} r_0:={(\kappa_\ell)}^{-1}\left(\sigma/({\sqrt{2}v})+1\right)^2 \sqrt{\log(2 /\delta)/{n}}$ …. ” Indeed we need Assumption 1 to hold with an $r$ that is larger than the upper bound, so that we can utilize the strong convexity to convert the loss error bound to the parameter estimation error bound; see the proof of Theorem 3.1. Thus $r$ has be larger than that error bound. Taking $r$ to be as small as possible, i.e., $r=r_0$, results in Assumption 1 being at its weakest.
>
> (c). Because the penalized pseudo-Huber loss function transitions from a quadratic to a linear function when $|x|$ is approximately $\sqrt{n}$, thus Assumption 1 holds with any $r$ such that $\sqrt{n} \gtrsim r \geq r_0$.  Thus we can take $r$ to be a constant, and this will not make the sample complexity condition worse. For example, to make the assumption independent of $r$, we can simply take $r=\sigma$ which, by $r\geq \textit{error~bound}$ implies that the following sample complexity condition
> $$
> r {\geq} {(\kappa_\ell)}^{-1}\left(\sigma/({\sqrt{2}v})+1\right)^2\sqrt{\log(2 /\delta)/{n}},
> $$
> which is implied by
> $$
> n\geq \kappa_\ell^{-2} (1/v_0 + 1/\sigma)^2 \log(2/\delta).
> $$
>
> This, together with $n\geq C\max\\{2z^2\sigma^2/v_0^2,\\ \log(1/\delta)\\}$, gives the sample complexity condition.  Directly plugging $r^2=r_0^2$ into the sample complexity leads to a quadratic inequality in $n$ and solving it gives complicated and hard-to-interpret sample complexity condition.  We prefer the current condition because it is intepretable. Specifically, the first sample complexity condition that $n\geq C z^2(\sigma^2+{r^2})/v_0^2$ comes from requirement that $\tau_{v_0}^2:=v_0^2 n/z^2 \geq C(\sigma^2 + r^2)$ in the proof of Lemma 3.2. Recall that the robustification parameter $\tau_{v_0}^2:= v_0^2 n/z^2$ determines the size of the quadratic region. Thus, intuitively, this requirement is minimal in the sense that Assumption  can only hold when $\tau_{v_0}^2$ is larger than $r^2$ plus the noise variance $\sigma^2$ (due to stochasticity).
>
> >
> > 3. In Theorem 3.4 and Theorem 3.5. One assumption is $c_0 \sigma_{v_0^2n/z^2}\leq C_0 \sigma < V_0$.
> >
>
> Because
> $
> \sigma_{v_0^2n/z^2}= \sqrt{\mathbb{E}[\varepsilon^21(\varepsilon^2\leq v_0^2 n/z^2)]}
> $
> is the truncated standard deviation, we have $\sigma_{v_0^2n/z^2}\leq \sigma$, and thus $c_0 \sigma_{v_0^2n/z^2}\leq C_0 \sigma < V_0$ automatically holds with $c_0\leq C_0$. Following the suggestions of other reviewers, we have rewritten the statements of the results. Please see the revised paper for details.

---

> ### Author Response · Authors · 2023-10-03
> **Response part 2**
>
> >
> > 4. In the numerical section, what is the method/solver actually used to solve problem (3.5)? How does this method/solver scale as the number of samples increase? This would determine the runtime efficiency of the proposed estimator.
> >
>
> We use alternating gradient descent with the Barzilai and Borwein method and backtracking line search (in python); see Appendix C for our meta algorithm. For the same setting as in Table 1, for $n=100, 1000, 10000, 100000$ (repeated 1000 times), the run time is $1.54$, $1.58$, $3.02$, $25.04$ seconds, respectively:
>
> |$n$| $100$| $1000$| $10,000$ | $100,000$|
> |------| -------- | ------- |------- |------- |
> |time| 1.54| 1.58| 3.02 | 25.04|
>
> We will release the code on github publicly.
>
>
> >
> > 5. In the numerical section, is there a reason not to compare all methods in the same plot? For example, can Figure 1 and Figure 3 be merged together? It'd be better to also report the runtimes of sample mean, MoM, and trimmed mean.
> >
>
> We provide the runtimes of sample mean, MoM, and trimmed mean. Specifically. When combining Figure 1 and Figure 3 into one figure, we found that the figure got every crowded with 4 lines overlapping together and it is very hard to distinguish different lines; see the lefts panels of Figure 1 and Figure 3. Moreover, Figure 3 is all for minimizing the penalized pseudo-Huber loss with different methods for tuning the robustification parameter. Thus we prefer not to combine Figure 1 and Figure 3. Table 1 is mainly for comparing the run time for different tuning methods. We report the runtime of sample mean, MoM, and trimmed mean here and in the last second paragraph on page 10. Specifically, for the same setting as in Table 1, the run time for sample mean, MoM, and trimmed mean  is $0.018$, $0.111$, and $0.057$, respectively:
>
> |method| sample mean| MoM| trimmed mean |
> |------| -------- | ------- |------- |
> |time| 0.018| 0.111| 0.057 |
>
>
> **Minor comments:**
> >
> > 1. Should the $\hat \mu(\tau)$ in (2.2) be $\tilde\mu(\tau)$?
> >
>
> We have fixed this. The earlier $\tilde\mu(\tau)$ should be $\hat \mu(\tau)$.
> >
> > 2.  Is the $\hat \mu(\tau)$ in Theorem 2.1 the solution of the pseudo-Huber loss or the penalized pseudo-Huber loss?
> >
>
> We have fixed the previous one and thus this one. It should be the solution to the  pseudo-Huber loss.
>
> >
> > 3. In the last paragraph of section 2, what is the loss function $\ell$? Should it be $\rho$ instead?
> >
>
> Yes, we have fixed this. Thanks for pointing this out.

---

> > ### Comment · Reviewer_p9aB · 2023-10-23
> > **Thank you**
> >
> > Thank you for responding to my comments in great details.

---

### Author Response · Authors · 2024-01-11
**The latest version**

The latest version of this paper can be found on arXiv: https://arxiv.org/abs/2107.00118. In this version,
1. we proof-read the manuscript and the appendix to minimize typos and minor errors.
2. we de-colored all texts.
3. following the suggestion of one reviewer, we made the constants in Theorem 3.4 and Theorem 3.5 universal.
4. we also made some slight modification to the figures.
5. we made the source code available at Github: https://github.com/statsle/automean.
6. we added a new figure - Figure 1 - in the main text, and added the following discussion: In summary, our self-tuned estimator can achieve optimal performance in both finite-sample and large-sample regimes. We point out that the large-sample regime is used to approximate the regime when the sample size is relatively large instead of describing the case of \ n = \infty \). We will refer to this ability as adaptivity to both finite-sample and large-sample regimes, or simply adaptivity. The MoM estimator does not naturally possess this adaptivity due to its discontinuous nature. Figure 1 provides a comparison between our self-tuned estimator and the MoM estimator in terms of adaptivity.

The earlier complete open reviews of this paper can be found at https://github.com/qsunstats/self-tuned-reports/blob/main/OpenReview.pdf and a later private review relating to Lee and Valiant can be found at https://github.com/qsunstats/self-tuned-reports/blob/main/TMLR%201503%20review.pdf. In support of open reviews, the authors of this paper shall soon repost these comments here and respond to them.

---

### Comment · Editors_In_Chief · 2024-01-15

In the process of reviewing this paper, the editorial board received one additional review, volunteered by email, rather than in the OpenReview system. This review was shared with the authors, who provided some responses to the comments in the review. After deliberation, the paper was rejected in the end. For transparency and additional context for any interested readers, we share a version of the emailed review, as well as the reviewer responses to the author responses. The authors, as well as anyone else, are welcome to comment on this review and discussion.

---

This paper proposes an ERM-style mean estimator for the class of univariate distributions with finite variance, with the goal of beating the standard median-of-means estimator.
Theoretically, the estimator achieves an almost-sub-Gaussian rate, up to a $\sqrt{\log n}$ factor (though, see important caveats below).
The authors also show some numerical experiments to justify their construction.

While this submission improves on Catoni's estimator in the sense of not having to know the variance exactly, the strongest known result for 1-d mean estimation is in fact by Lee and Valiant (FOCS 2021, Optimal Sub-Gaussian Mean Estimation in $\mathbb{R}$), where they achieve the optimal estimation error $(1+o(1)) \sigma\sqrt{2\log\frac{1}{\delta}/n}$, with the optimal multiplicative constant of $\sqrt{2}$, without needing to know the variance or any information about it other than the fact that it exists.
The multiplicative $1+o(1)$ slack is also independent of the data distribution, meaning there is uniform convergence over the class of finite-variance distributions.
These guarantees imply that Lee and Valiant's estimator is completely "self-tuned" in the sense of this submission.

As such, Lee and Valiant already solved the problem set out by this submission, with provable statistical optimality, while the results of this paper are much weaker, being loose to a $\sqrt{\log n}$ factor, and in fact additionally requires coarse knowledge of the variance which makes it not fully self-tuned (see more below).
Yet, the Lee and Valiant result was neither discussed nor cited in this paper.

By contrast, the estimator construction and results of this submission are far from statistically optimal, and more importantly, the estimator is not actually completely self-tuning.
The estimator still needs some rough knowledge on the variance of the distribution (the $v_0$ and $V_0$ quantities in Section 3.2).
Assuming only the existence of the second moment, it is impossible to estimate the variance itself with distribution-independent non-asymptotic bounds.
Along similar lines, the authors also acknowledge that the "constant" $C$ in Theorem 3.5 cannot converge to any good universal constant at a uniform rate over the class of finite-variance distributions, due to its dependence on the truncated variance.
In fact, from the proof of Theorem 3.5, this $C$ can be arbitrarily large, for badly behaved distributions in the small-sample regime, when the truncated variance is extremely small (or even 0) compared to the true variance.
Thus, strictly speaking, even if we had coarse knowledge of the variance, the non-asymptotic result of this paper (Theorem 3.5) is much weaker than the Catoni bounds, because the “constant” $C$ is rather non-trivially dependent on the unknown data distribution.
The distribution-dependence of $C$ also makes the theorem unusable in practice (e.g. for calculating the number of samples to get to within $\epsilon$ number of standard deviations from the true mean).
On the other hand, both Catoni and Lee-Valiant give distribution-independent rates and do not suffer from this important shortcoming, fully demonstrating their robustness to heavy tails.


In terms of techniques, the conceptual contribution of this submission is at a high-level also somewhat similar to the Lee-Valiant estimator yet this paper gets rather sub-optimal guarantees.
The key idea in this work is to introduce another parameter in the loss function that represents the scale, and to optimize for it directly from the data.
As Lee and Valiant discuss explicitly in their work (e.g. Section III.D in FOCS 2021 paper), their estimator already is also expressible as a 2-parameter $\psi$-estimator, where they add an $\alpha$ parameter in addition to the mean estimate $\hat{\mu}$ to handle the same scale-tuning issue.
The Lee and Valiant construction and analysis yields a tight statistical guarantee and an easy-to-implement and fast-to-run estimator, while the 2-parameter M-estimation approach in this paper is far from optimal and further requires running ERM.
Consequently, the authors have not made a convincing case for how their construction and result add to the existing literature.

(cont.)

---

> ### Comment · Editors_In_Chief · 2024-01-15
>
> We also note that, relative to Lee and Valiant first posting their result on arXiv, the original version of this submission was made publicly available more than 7 months after, and the current submission was sent to TMLR more than 2 years and 9 months after.
> Due to the large time gaps, it would be inappropriate to consider this submission as work concurrent with (Lee and Valiant).
>
> ---
>
> Below, we address some of the points raised by the authors earlier in the review process.
>
> 1. Initialization. The authors of this submission pointed out that the Lee and Valiant result uses median-of-means as an initialization, while the estimator in this paper is "initialization free". The authors further claimed that the extra $\sqrt{\log n}$ factor is probably unavoidable for initialization-free estimators.
>
> We are confused as to why using an initialization would be an issue.
> Moreover, the authors' technical claim is incorrect.
> Lee and Valiant's estimator can be interpreted as a one-step iteration in an iterative fixpoint-finding computation (Section III.D in FOCS 2021 paper): running the estimator multiple times, replacing the "initialization" with the previous estimate, and keep running that in order to find the fixpoint of a $\psi$-estimator. It just so happens that the convergence is very fast, that even using median-of-means as an initialization is good enough to only do a single iteration. The authors' technical claim that the spurious $\sqrt{\log n}$ factor cannot be removed if initialization-free is therefore incorrect, if we use the $\psi$-estimator version of the Lee and Valiant construction (Definition 4 in FOCS 2021 paper). But again, it is unclear in the first place why it is an issue to use an initialization.
>
> 2. Asymptotic results. The authors pointed out that Lee and Valiant does not prove any asymptotic results for their estimator, whereas this paper establishes asymptotic normality and efficiency under the slightly stronger assumption of having a $2+\iota$ moment for any $\iota > 0$ (Theorem 4.3).
>
> However, getting an asymptotically normal and efficient estimator is trivial: the sample mean is already asymptotically optimal. The key challenge and the point of Catoni's and Lee-Valiant's contributions is robustness to heavy tails, which was demonstrated through tight non-asymptotic guarantees with optimal constants, and importantly uniform convergence over the class of finite-variance distributions. As the review above already pointed out, the non-asymptotic bounds in this submission are much weaker, and lack uniform convergence.
>
> Additionally, while Lee and Valiant did not explicitly prove asymptotic normality and efficiency, showing these results is a straightforward exercise by the construction of their estimator, under the same $2+\iota$ moment assumption made in this submission in Theorem 4.3.
> As they described in their work, Lee and Valiant's estimator (fractionally) trims a total of $\Theta(\log 1/\delta)$ many samples (which is a constant-in-$n$ number of samples) before returning the mean of the remaining samples.
> From this, it is a pretty simple exercise to show that the difference between the sample mean and their estimator converges to 0 in probability, meaning that their estimator is also asymptotically normal and efficient.
>
> 3. Multivariate extension. The authors pointed out that they showed a multivariate extension of their estimator while Lee and Valiant did not.
>
> Even ignoring the same uniform convergence issues and coarse-knowledge-of-(co)variance issues in their 1-d construction, the submission's high-d non-asymptotic bounds are even more loose than their 1-d bounds.
> Compared to the optimal sub-Gaussian rate, this submission loses not only a $\sqrt{\log n}$ factor, but is as bad as *multiplicative* $\operatorname{Tr}(\Sigma)$ *times* $\log(n/\delta)$ instead of *additive* in the sub-Gaussian rate.
> In fact, the shown guarantees are worse than the standard coordinatewise median-of-means estimator, which is well-known to be highly sub-optimal.
> See for example the discussion in the Lugosi-Mendelson survey (also cited in this submission) on known results and the development and focus of the field.

---

> ### Author Response · Authors · 2024-01-15
> **Comments by the author**
>
> Thanks for the comments. This is not the full version of the reviewer’s comments. For the original comments, please see the link posted below, which we will also repost here in comment. Because the reviewers refuse to openly review on the platform of open review (as pointed out by Gautam Kamath), the authors are not aware of Part II of this post. Part II seems due to a misread of the authors points. The authors will post the original comments, our points, and version of  the email history, and related things, after travel. Stay tuned!